# Different estimation methods of the modified Kies Topp-Leone model with applications and quantile regression

Safar M. Alghamdi[1], Olayan Albalawi[2], Sanaa Mohammed Almarzouki[3], Vasili B. V. Nagarjuna[4], Suleman Nasiru[5], Mohammed Elgarhy[6,7]*

1 Department of Mathematics and Statistics, College of Science, Taif University, Taif, Saudi Arabia, 2 Department of Statistics, Faculty of Science, University of Tabuk, Tabuk, Saudi Arabia, 3 Statistics Department, Faculty of Science, King Abdul Aziz University, Jeddah, Kingdom of Saudi Arabia, 4 Department of Mathematics Vellore Institute of Technology Andhra Pradesh, Amaravati, India, 5 Department of Statistics and Actuarial Science, School of Mathematical Sciences, C. K. Tedam University of Technology and Applied Sciences, Ghana, 6 Mathematics and Computer Science Department, Faculty of Science, Beni-Suef University, Beni-Suef, Egypt, 7 Department of Basic Sciences, Higher Institute of Administrative Sciences, Belbeis, AlSharkia, Egypt

* m_elgarhy85@sva.edu.eg

**Data Availability Statement:** All data are reported in data analysis Section.

**Funding:** This research was funded by Taif University, Saudi Arabia, Project No. (TU-DSSP-

## Abstract

This paper introduces the modified Kies Topp-Leone (MKTL) distribution for modeling data on the (0, 1) or [0, 1] interval. The shapes of the density and hazard rate functions manifest desirable shapes, making the MKTL distribution suitable for modeling data with different characteristics at the unit interval. Twelve different estimation methods are utilized to estimate the distribution parameters, and Monte Carlo simulation experiments are executed to assess the performance of the methods. The simulation results suggest that the maximum likelihood method is the superior method. The usefulness of the new distribution is illustrated by utilizing three data sets, and its performance is juxtaposed with that of other competing models. The findings affirm the superiority of the MKTL distribution over the other candidate models. Applying the developed quantile regression model using the new distribution disclosed that it offers a competitive fit over other existing regression models.

## 1 Introduction

Distributional assumptions are innate in ascertainment of an apt parametric model for analysis. As a consequence, deciding on a germane parametric model has an interconnection with the concealed distribution governing the data generating process. Therefore exploring to identify the appropriate distribution before fitting any parametric model to any data is not only a requirement but cardinal in making right inferences. Notwithstanding the fact that innumerable distributions exist for one to hand-pick from for any analysis, providing the best fit with almost zero or minimal loss of information is essential. This has called for the appendages of existing distributions by researchers with the primary goal of ameliorating their performances.

The Topp-Leone (TL) distribution (see [1]) is one of the oldest distributions that have been recently modified by researchers to enhance its suitability in modeling data. The TL

2024-296). The funders had role in study design, software, numerical analysis, and preparation of the final version of the manuscript.

**Competing interests:** The authors have declared that no competing interests exist.

distribution with shape parameter $\eta > 0$ has cumulative distribution function (CDF) and probability density function (PDF) define respectively as

$$G(x; \eta) = x^\eta (2 - x)^\eta = (1 - (1 - x)^2)^\eta, 0 < x < 1 \tag{1}$$

and

$$g(x; \eta) = 2\eta x^{\eta-1}(1 - x)(2 - x)^{\eta-1} 0 < x < 1. \tag{2}$$

Some of the extensions of the TL distribution in literature are: new extended TL distribution by [2], cosine TL Weibull distribution by [3], tangent TL Weibull distribution by [4], modified Kies inverted TL distribution by [5], Fréchet TL Kumaraswamy distribution by [6], TL Weibull distribution by [7], type II power TL normal distribution by [8], sine TL inverse Lomax distribution by [9], Weibull TL generated generalized half-normal distribution by [10], inverted TL distribution by [11], TL Gompertz distribution by [12], new power TL-G family by [13], Type II generalized TL-G family by [14], Type I half-logistic TL distribution by [15], Type II TL-G family by [16] and for more information see [17–24]. In this study, a new add-on of the TL distribution called modified Kies TL (MKTL) distribution is developed utilizing the modified Kies (MK) family of distributions proposed by [25]. The CDF and PDF of the MK family of distributions are respectively given by

$$F(x; \mathbf{\Psi}) = 1 - e^{-\left[\frac{G(x;\mathbf{\Psi})}{1-G(x;\mathbf{\Psi})}\right]^\beta}, x \in \mathbb{R}, \beta > 0 \tag{3}$$

and

$$f(x; \mathbf{\Psi}) = \frac{\beta g(x; \mathbf{\Psi}) G(x; \mathbf{\Psi})^{\beta-1}}{[1 - G(x; \mathbf{\Psi})]^{\beta+1}} e^{-\left[\frac{G(x;\mathbf{\Psi})}{1-G(x;\mathbf{\Psi})}\right]^\beta}, x \in \mathbb{R}, \beta > 0, \tag{4}$$

where $g(x; \mathbf{\Psi})$ and $G(x; \mathbf{\Psi})$ are the parent PDF and CDF for the baseline distribution with a set of parameters $\Psi$ and $\beta$ is the shape parameter of the family. Furthermore, [25] employed the binomial and exponential series to reformulate the PDF as a linear amalgamation of the exponentiated family as follows:

$$f(x; \mathbf{\Psi}) = g(x; \mathbf{\Psi}) \sum_{i,j=0}^{\infty} \vartheta_{i,j} [G(x; \mathbf{\Psi})]^{\beta(i+1)+j-1}, \tag{5}$$

where $\vartheta_{i,j} = \frac{(-1)^i \beta}{i!} \binom{\zeta(i+1)+j}{j}$.

Our motivations for the formulation of the MKTL distribution are:

- Propose a new distribution capable of fitting data on the unit interval with various traits and offer optimal fit with least loss of information.

- Study the traits of the estimates of the parameters of the MKTL distribution using twelve different estimation procedures in order to pinpoint the most desired estimation method for estimating the parameters.

- Examine the utility of the MKTL distribution utilizing data sets with different characteristics and compare its performances with other competitive distributions.

- Formulate new quantile regression for modeling the relationship between an endogenous variable define on the (0, 1) interval and a set of exogenous variables.

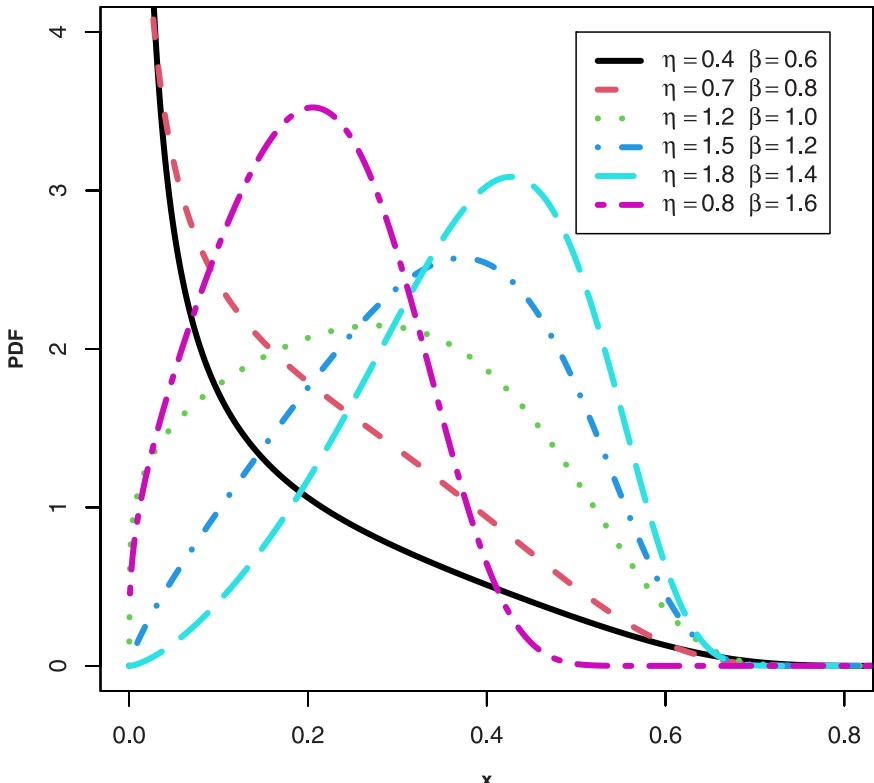

**Fig 1. Plots of PDF for the MKTL distribution.**

The succeeding sections of the work are put together in the following form: The formulation of the MKTL distribution is detailed in section 2. The statistical properties of the MKTL distribution are presented in Section 3. The estimation procedures utilized to estimate the parameters of the model are described in Section 4. The simulation experiments performed to examine how well the estimation methods estimates the parameters are conferred in Section 5. The usefulness of the MKTL distribution with respect to fitting data is given in Section 6. The MKTL quantile regression model and its applications are dispensed in Section 7. The Concluding remarks are finally given in Section 8.

## 2 Formulation of the MKTL distribution

In this section, we construct the Modified Kies Topp Leone (MKTL) distribution by inserting (3) and (4) in (1) and (2) and it has the following CDF, PDF, reliability function (RF) and HRF of the MKTL of distribution are

$$F(x; \eta, \beta) = 1 - e^{-[(1-(1-x)^2)^{-\eta}-1]^{-\beta}}, \quad 0 < x < 1, \; \eta, \beta > 0, \quad (6)$$

$$f(x; \eta, \beta) = \frac{2\beta\eta x^{\eta-1}(1-x)(2-x)^{\eta-1}(1-(1-x)^2)^{\eta\beta-\eta}}{[1-(1-(1-x)^2)^{\eta}]^{\beta+1}} e^{-[(1-(1-x)^2)^{-\eta}-1]^{-\beta}}, \; 0 < x < 1, \; \eta, \beta > 0. \quad (7)$$

$$R(x; \beta, \eta) = e^{-[(1-(1-x)^2)^{-\eta}-1]^{-\beta}},$$

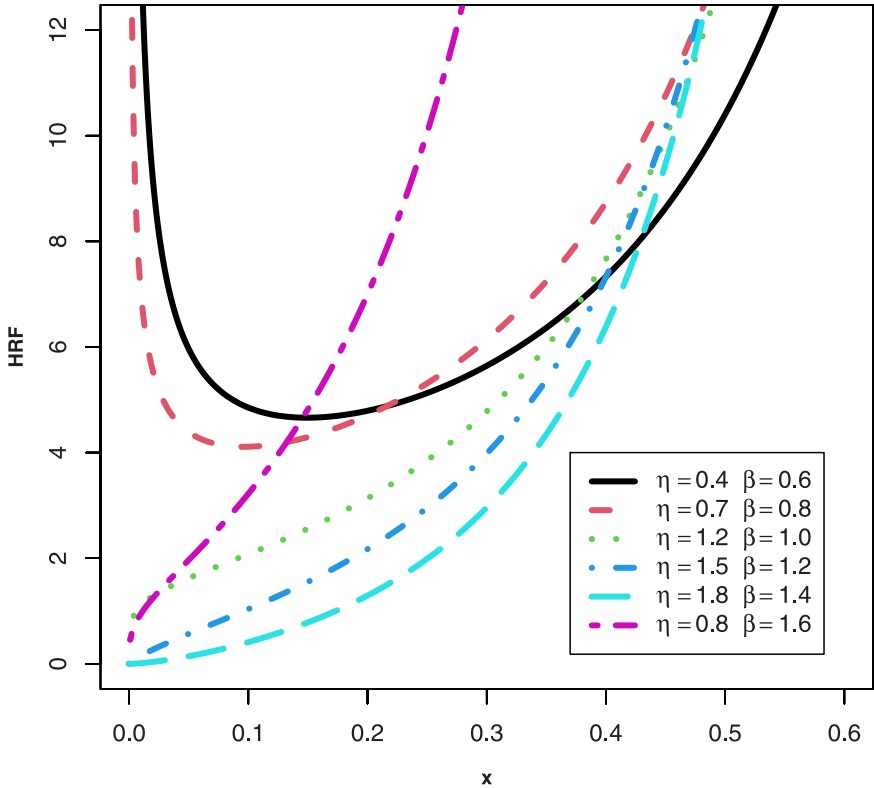

**Fig 2. Plots of HRF for the MKTL distribution.**

and

$$h(x; \beta, \eta) = \frac{2\beta\eta x^{\eta-1}(1-x)(2-x)^{\eta-1}(1-(1-x)^2)^{\eta\beta-\eta}}{[1-(1-(1-x)^2)^{\eta}]^{\beta+1}}.$$

The reversed HRF (RHRF), cumulative HRF (CHRF), odd ratio (OR), failure rate average (FRA) and Mills ratio (MR) of the MKTL of distribution are

$$\tau(x; \beta, \eta) = \frac{2\beta\eta x^{\eta-1}(1-x)(2-x)^{\eta-1}(1-(1-x)^2)^{\eta\beta-\eta}}{[1-e^{-[(1-(1-x)^2)^{-\eta}-1]^{-\beta}}][1-(1-(1-x)^2)^{\eta}]^{\beta+1}} e^{-[(1-(1-x)^2)^{-\eta}-1]^{-\beta}},$$

$$H(x; \beta, \eta) = \frac{-1}{[(1-(1-x)^2)^{-\eta}-1]^{-\beta}},$$

$$OR(x; \beta, \eta) = e^{[(1-(1-x)^2)^{-\eta}-1]^{-\beta}} - 1,$$

$$FRA(x; \beta, \eta) = \frac{-1}{x[(1-(1-x)^2)^{-\eta}-1]^{-\beta}},$$

**Table 1. Results of *Q1, Q2, Q3, BSK* and *MKUR* associated with the MKTL distribution.**

| η | β | Q1 | Q2 | Q3 | BSK | MKUR |
|---|---|---|---|---|---|---|
| 1.5 | 2.0 | 0.23096 | 0.29338 | 0.34869 | -0.06044 | 1.20953 |
| | 2.5 | 0.25929 | 0.31200 | 0.35714 | -0.07731 | 1.22794 |
| | 3.0 | 0.27939 | 0.32475 | 0.36283 | -0.08721 | 1.23970 |
| | 3.5 | 0.29430 | 0.33400 | 0.36691 | -0.09357 | 1.24758 |
| | 4.0 | 0.30576 | 0.34102 | 0.36998 | -0.09795 | 1.25311 |
| 2.0 | 2.0 | 0.30074 | 0.36377 | 0.41776 | -0.07733 | 1.22279 |
| | 2.5 | 0.32967 | 0.38212 | 0.42589 | -0.09017 | 1.23891 |
| | 3.0 | 0.34986 | 0.39457 | 0.43134 | -0.09755 | 1.24888 |
| | 3.5 | 0.36468 | 0.40357 | 0.43524 | -0.10221 | 1.25542 |
| | 4.0 | 0.37600 | 0.41036 | 0.43817 | -0.10535 | 1.25991 |
| 2.5 | 2.0 | 0.35538 | 0.41716 | 0.46902 | -0.08731 | 1.23138 |
| | 2.5 | 0.38392 | 0.43488 | 0.47676 | -0.09778 | 1.24585 |
| | 3.0 | 0.40365 | 0.44685 | 0.48193 | -0.10367 | 1.25462 |
| | 3.5 | 0.41804 | 0.45546 | 0.48563 | -0.10732 | 1.26026 |
| | 4.0 | 0.42898 | 0.46196 | 0.48841 | -0.10973 | 1.26409 |
| 3.0 | 2.0 | 0.39933 | 0.45925 | 0.50889 | -0.09390 | 1.23736 |
| | 2.5 | 0.42713 | 0.47627 | 0.51625 | -0.10280 | 1.25061 |
| | 3.0 | 0.44623 | 0.48773 | 0.52117 | -0.10772 | 1.25852 |
| | 3.5 | 0.46010 | 0.49597 | 0.52468 | -0.11070 | 1.26355 |
| | 4.0 | 0.47061 | 0.50216 | 0.52732 | -0.11263 | 1.26692 |
| 3.5 | 2.0 | 0.43555 | 0.49348 | 0.54101 | -0.09858 | 1.24175 |
| | 2.5 | 0.46251 | 0.50982 | 0.54803 | -0.10637 | 1.25408 |
| | 3.0 | 0.48094 | 0.52080 | 0.55271 | -0.11059 | 1.26135 |
| | 3.5 | 0.49429 | 0.52867 | 0.55606 | -0.11310 | 1.26593 |
| | 4.0 | 0.50439 | 0.53459 | 0.55857 | -0.11469 | 1.26896 |
| 4.0 | 2.0 | 0.46602 | 0.52199 | 0.56758 | -0.10207 | 1.24511 |
| | 2.5 | 0.49213 | 0.53769 | 0.57430 | -0.10903 | 1.25672 |
| | 3.0 | 0.50991 | 0.54823 | 0.57878 | -0.11274 | 1.26350 |
| | 3.5 | 0.52277 | 0.55577 | 0.58197 | -0.11489 | 1.26772 |
| | 4.0 | 0.53248 | 0.56144 | 0.58437 | -0.11623 | 1.27050 |

and

$$MR(x; \beta, \eta) = \frac{[1 - (1 - (1 - x)^2)^\eta]^{\beta+1}}{2\beta\eta x^{\eta-1}(1 - x)(2 - x)^{\eta-1}(1 - (1 - x)^2)^{\eta\beta-\eta}}.$$

We can notice some observations from Figs 1 and 2 such as: the PDF for the MKTL distribution can be declining, right-skewed, left-skewed and unimodal shapes, but the HRF can be bathtub, increasing and J-shaped for the MKTL distribution.

## 3 Statistical properties

The essential mathematical characteristics of the MKTL distribution are addressed in this section of the article. The quantile function, the median, moments, and incomplete and conditional moments are computed.

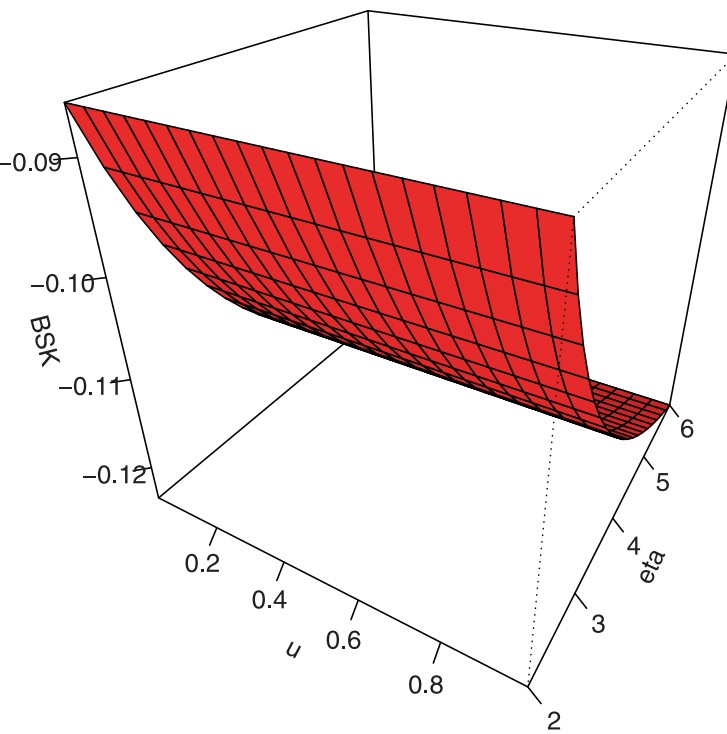

**Fig 3. 3D Plots of BSK at $\beta$ = 1.5 for the MKTL distribution.**

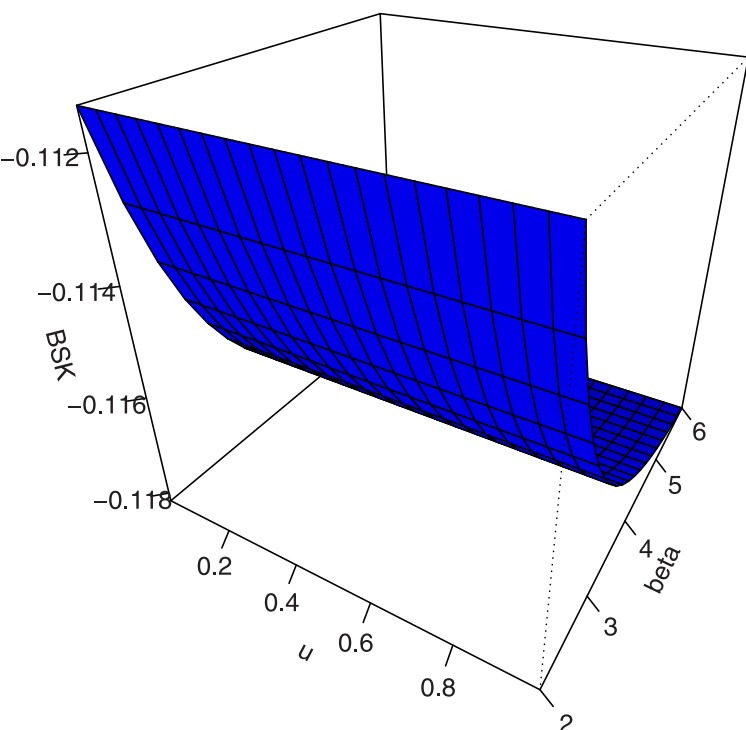

**Fig 4. 3D Plots of BSK at $\eta$ = 3.0 for the MKTL distribution.**

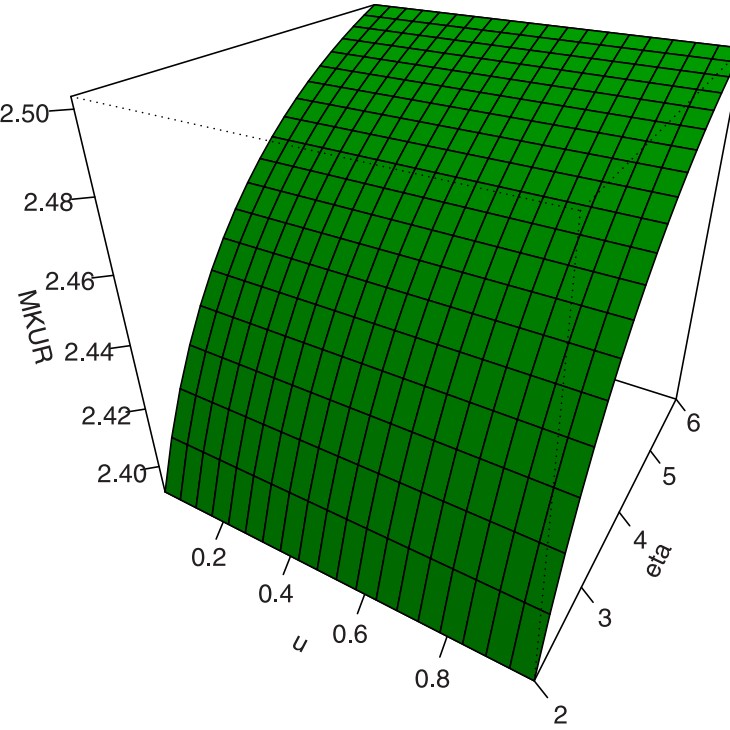

**Fig 5. 3D Plots of MKUR at $\beta$ = 1.5 for the MKTL distribution.**

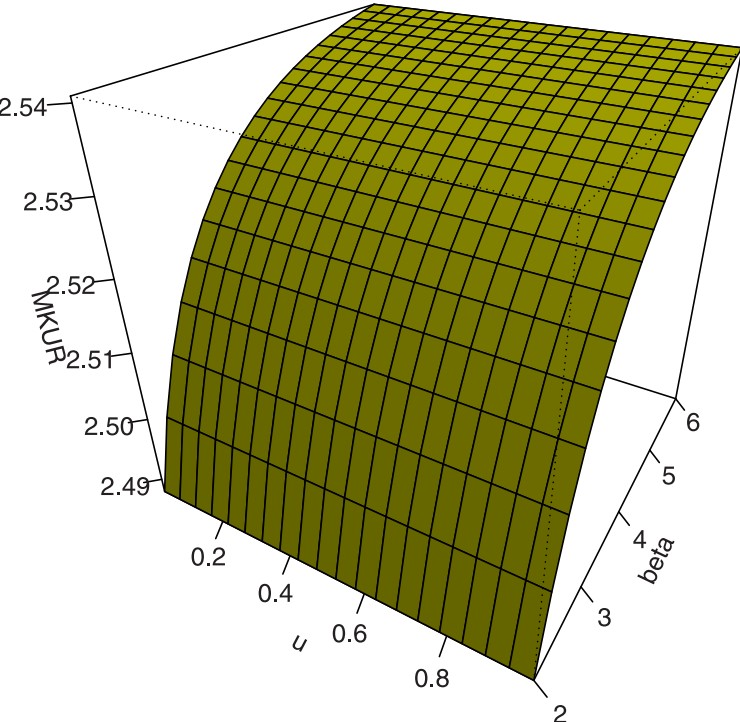

**Fig 6. 3D Plots of MKUR at $\eta$ = 3.0 for the MKTL distribution.**

**Table 2. Results of $\mu'_1$, $\mu'_2$, $\mu'_3$, $\mu'_4$, $\sigma^2$, $\sigma$, CS, CK, and CV associated with the MKTL distribution.**

| $\eta$ | $\beta$ | $\mu'_1$ | $\mu'_2$ | $\mu'_3$ | $\mu'_4$ | $\sigma^2$ | $\sigma$ | CS | CK | CV |
|---|---|---|---|---|---|---|---|---|---|---|
| 0.3 | 0.2 | 0.19490 | 0.13294 | 0.10194 | 0.08254 | 0.09495 | 0.30814 | 1.33334 | 3.22136 | 1.58106 |
| | 0.4 | 0.11592 | 0.04934 | 0.02611 | 0.01540 | 0.03590 | 0.18947 | 1.77465 | 5.21677 | 1.63445 |
| | 0.5 | 0.09569 | 0.03265 | 0.01436 | 0.00719 | 0.02350 | 0.15330 | 1.87184 | 5.85345 | 1.60197 |
| | 0.7 | 0.07248 | 0.01681 | 0.00525 | 0.00192 | 0.01156 | 0.10752 | 1.89869 | 6.34648 | 1.48325 |
| | 0.9 | 0.06067 | 0.01033 | 0.00241 | 0.00067 | 0.00665 | 0.08155 | 1.79743 | 6.13874 | 1.34425 |
| 0.5 | 0.2 | 0.22959 | 0.15970 | 0.12456 | 0.10234 | 0.10699 | 0.32709 | 1.10789 | 2.63131 | 1.42471 |
| | 0.4 | 0.16540 | 0.07697 | 0.04365 | 0.02728 | 0.04961 | 0.22273 | 1.31319 | 3.57015 | 1.34665 |
| | 0.5 | 0.14885 | 0.05819 | 0.02840 | 0.01551 | 0.03603 | 0.18982 | 1.31740 | 3.74787 | 1.27524 |
| | 0.7 | 0.13015 | 0.03832 | 0.01438 | 0.00615 | 0.02138 | 0.14622 | 1.22450 | 3.71887 | 1.12351 |
| | 0.9 | 0.12117 | 0.02899 | 0.00882 | 0.00308 | 0.01431 | 0.11962 | 1.07516 | 3.46499 | 0.98726 |
| 0.7 | 0.2 | 0.25511 | 0.17914 | 0.14108 | 0.11691 | 0.11406 | 0.33773 | 0.96538 | 2.32043 | 1.32385 |
| | 0.4 | 0.20457 | 0.10024 | 0.05919 | 0.03827 | 0.05840 | 0.24166 | 1.04810 | 2.86142 | 1.18128 |
| | 0.5 | 0.19229 | 0.08138 | 0.04219 | 0.02425 | 0.04441 | 0.21074 | 1.01137 | 2.91820 | 1.09595 |
| | 0.7 | 0.17958 | 0.06067 | 0.02529 | 0.01183 | 0.02842 | 0.16858 | 0.87336 | 2.82566 | 0.93875 |
| | 0.9 | 0.17467 | 0.05057 | 0.01786 | 0.00708 | 0.02006 | 0.14163 | 0.71101 | 2.65726 | 0.81091 |
| 0.9 | 0.2 | 0.27589 | 0.19471 | 0.15431 | 0.12861 | 0.11859 | 0.34437 | 0.86079 | 2.12359 | 1.24825 |
| | 0.4 | 0.23735 | 0.12052 | 0.07315 | 0.04842 | 0.06419 | 0.25336 | 0.86541 | 2.47444 | 1.06744 |
| | 0.5 | 0.22899 | 0.10242 | 0.05534 | 0.03295 | 0.04998 | 0.22356 | 0.80552 | 2.49367 | 0.97632 |
| | 0.7 | 0.22183 | 0.08235 | 0.03686 | 0.01831 | 0.03314 | 0.18204 | 0.64472 | 2.41910 | 0.82059 |
| | 0.9 | 0.22062 | 0.07258 | 0.02833 | 0.01224 | 0.02391 | 0.15463 | 0.47772 | 2.33034 | 0.70082 |
| 1.1 | 0.2 | 0.29371 | 0.20785 | 0.16545 | 0.13849 | 0.12158 | 0.34868 | 0.77791 | 1.98681 | 1.18717 |
| | 0.4 | 0.26566 | 0.13859 | 0.08586 | 0.05785 | 0.06802 | 0.26081 | 0.72752 | 2.23783 | 0.98169 |
| | 0.5 | 0.26072 | 0.12162 | 0.06780 | 0.04143 | 0.05365 | 0.23162 | 0.65277 | 2.24885 | 0.88842 |
| | 0.7 | 0.25832 | 0.10292 | 0.04858 | 0.02524 | 0.03619 | 0.19024 | 0.47889 | 2.21164 | 0.73640 |
| | 0.9 | 0.26019 | 0.09403 | 0.03950 | 0.01818 | 0.02633 | 0.16227 | 0.31056 | 2.18880 | 0.62369 |
| 1.3 | 0.2 | 0.30950 | 0.21932 | 0.17513 | 0.14707 | 0.12353 | 0.35147 | 0.70910 | 1.88639 | 1.13562 |
| | 0.4 | 0.29062 | 0.15496 | 0.09758 | 0.06666 | 0.07050 | 0.26552 | 0.61758 | 2.08382 | 0.91363 |
| | 0.5 | 0.28861 | 0.13929 | 0.07958 | 0.04966 | 0.05600 | 0.23664 | 0.53277 | 2.09893 | 0.81992 |
| | 0.7 | 0.29020 | 0.12229 | 0.06019 | 0.03240 | 0.03807 | 0.19512 | 0.35087 | 2.10251 | 0.67237 |
| | 0.9 | 0.29455 | 0.11453 | 0.05094 | 0.02463 | 0.02777 | 0.16664 | 0.18270 | 2.13399 | 0.56581 |
| 1.5 | 0.2 | 0.32376 | 0.22956 | 0.18374 | 0.15470 | 0.12474 | 0.35319 | 0.65020 | 1.81001 | 1.09088 |
| | 0.4 | 0.31295 | 0.16997 | 0.10846 | 0.07494 | 0.07203 | 0.26838 | 0.52680 | 1.98007 | 0.85759 |
| | 0.5 | 0.31344 | 0.15565 | 0.09076 | 0.05760 | 0.05741 | 0.23960 | 0.43479 | 2.00465 | 0.76444 |
| | 0.7 | 0.31834 | 0.14048 | 0.07156 | 0.03965 | 0.03915 | 0.19786 | 0.24788 | 2.04755 | 0.62153 |
| | 0.9 | 0.32467 | 0.13395 | 0.06241 | 0.03140 | 0.02854 | 0.16894 | 0.08068 | 2.12406 | 0.52035 |
| 1.7 | 0.2 | 0.33684 | 0.23887 | 0.19153 | 0.16160 | 0.12540 | 0.35412 | 0.59866 | 1.75052 | 1.05130 |
| | 0.4 | 0.33313 | 0.18384 | 0.11864 | 0.08276 | 0.07287 | 0.26994 | 0.44988 | 1.90899 | 0.81030 |
| | 0.5 | 0.33575 | 0.17089 | 0.10137 | 0.06526 | 0.05816 | 0.24116 | 0.35261 | 1.94535 | 0.71826 |
| | 0.7 | 0.34340 | 0.15758 | 0.08263 | 0.04691 | 0.03965 | 0.19912 | 0.16260 | 2.02461 | 0.57989 |
| | 0.9 | 0.35133 | 0.15228 | 0.07376 | 0.03836 | 0.02885 | 0.16985 | 0.00318 | 2.13888 | 0.48345 |

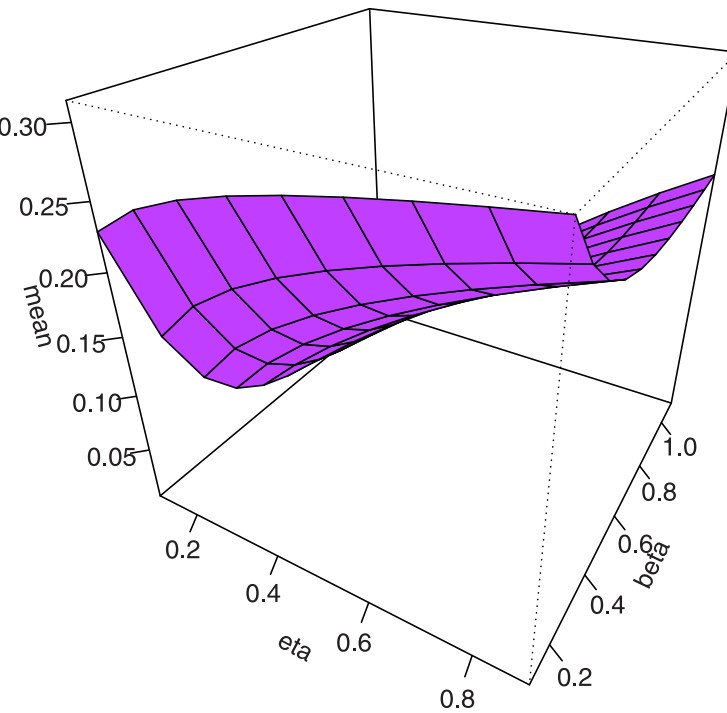

**Fig 7. 3D plots of mean for the MKTL distribution.**

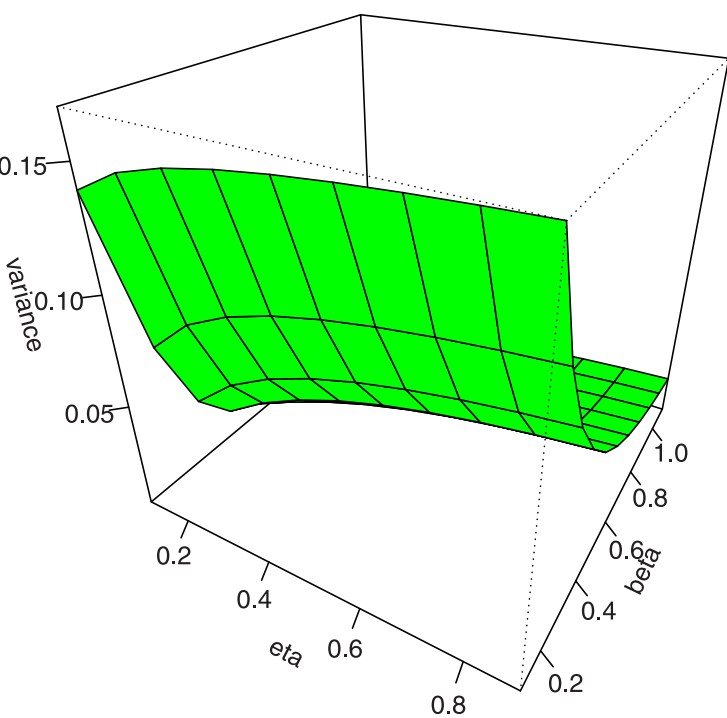

**Fig 8. 3D plots of variance for the MKTL distribution.**

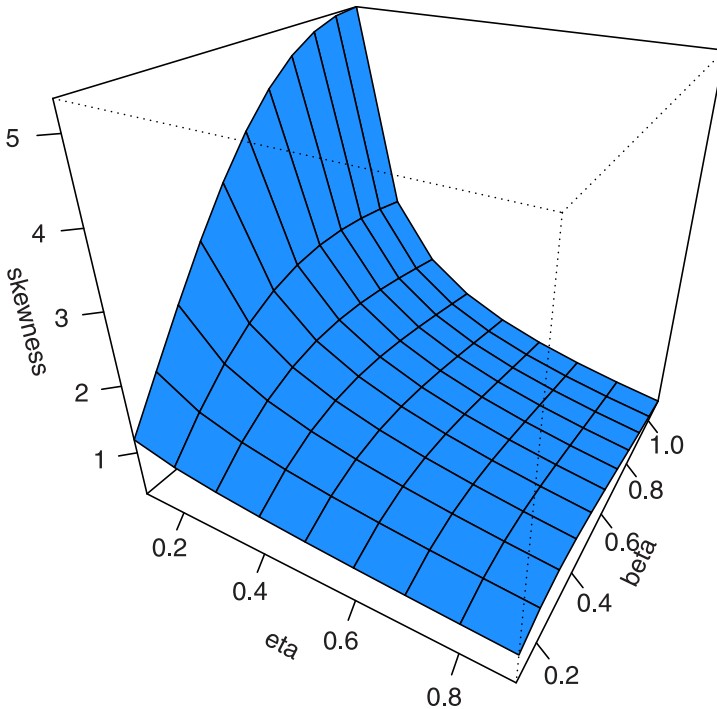

**Fig 9. 3D plots of CS for the MKTL distribution.**

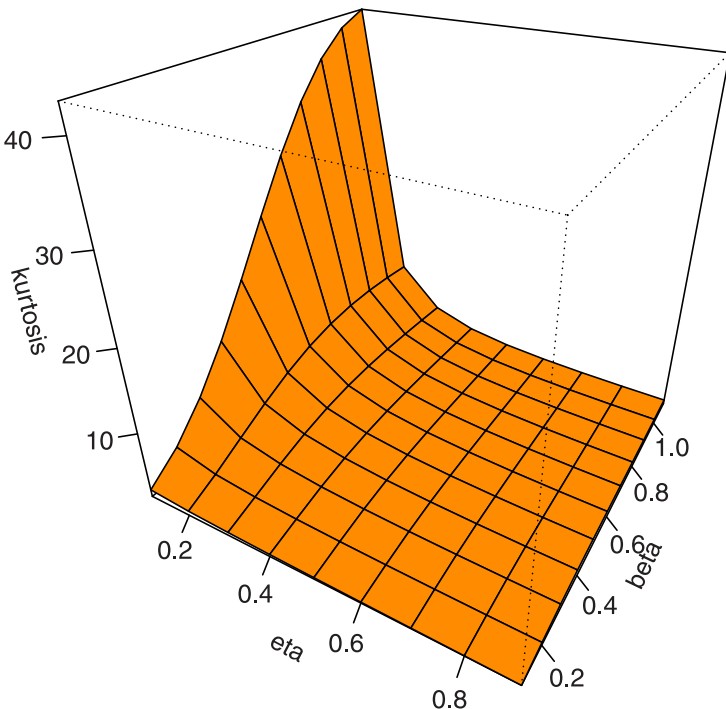

**Fig 10. 3D plots of CK for the MKTL distribution.**

**Table 3. Initial values of the parameters $\eta$ and $\beta$.**

| Table | 4 | 5 | 6 | 7 | 8 | 9 | 10 |
|---|---|---|---|---|---|---|---|
| $\eta$ | 0.4 | 0.4 | 0.9 | 0.9 | 0.9 | 1.4 | 1.9 |
| $\beta$ | 0.4 | 0.9 | 0.4 | 0.9 | 1.4 | 0.9 | 1.4 |

## 3.1 Quantile function

The $u^{th}$ quantile symbolized by $x_u$ of the MKTL distribution is obtained from the subsequent formula

$$Q(u) = x_u = 1 - \left(1 - \left(1 + \left[\log\left(\frac{1}{1-u}\right)\right]^{\frac{-1}{\beta}}\right)^{\frac{-1}{\eta}}\right)^{\frac{1}{2}}. \tag{8}$$

In order to determine the median of the MKTL distribution, we substitute $u = 0.5$ into Eq (8) as shown:

$$m = x_{0.5} = 1 - \left(1 - \left(1 + [\log(2)]^{\frac{-1}{\beta}}\right)^{\frac{-1}{\eta}}\right)^{\frac{1}{2}}.$$

Also, by substituting $u = 0.25, 0.5, 0.75$ into Eq (8) we get the first ($Q1$), second ($Q2$) and third ($Q3$) quantiles. Table 1 shows some numerical values of quantiles for the MKTL distribution. Figs 3–6 show 3D Plots of BSK and MKUR at $\beta = 1.5$ and at $\eta = 3.0$ for the MKTL distribution.

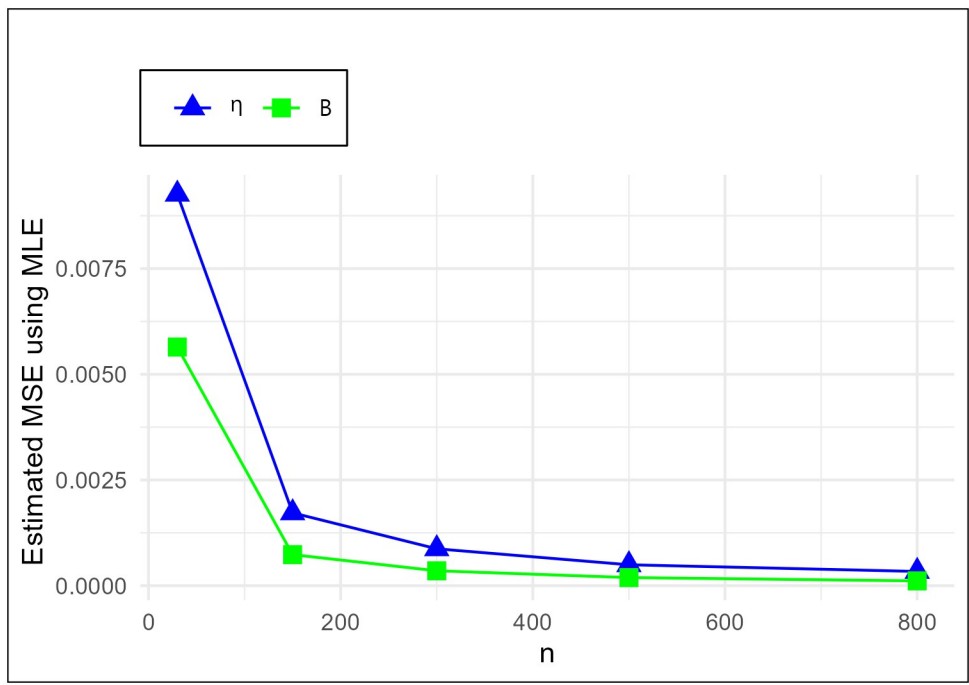

**Fig 11. MSE for MLE schemes in Table 4.**

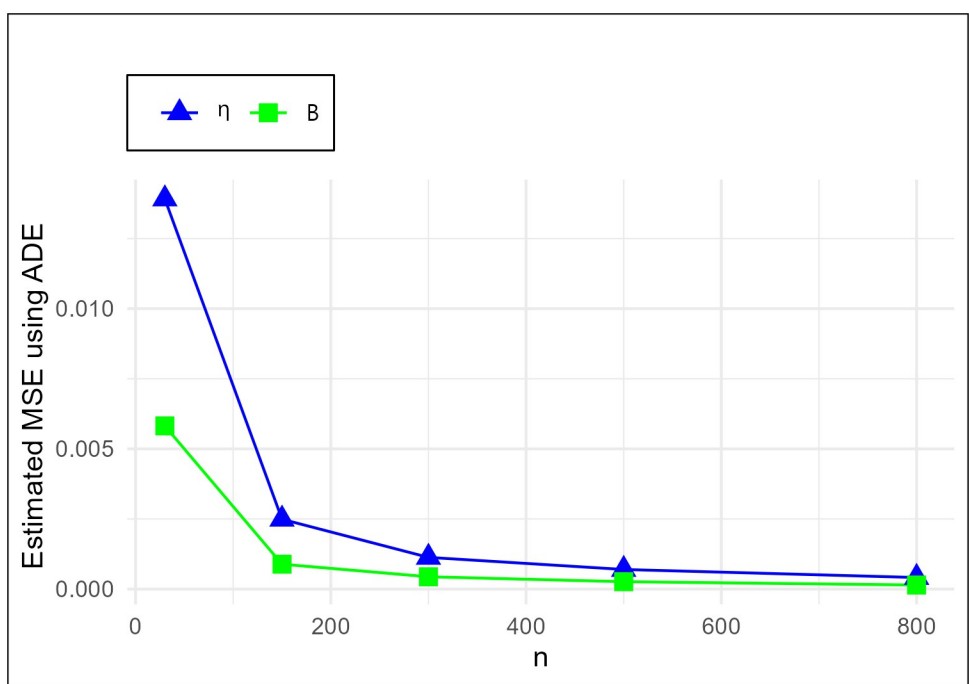

**Fig 12. MSE for ADE schemes in Table 4.**

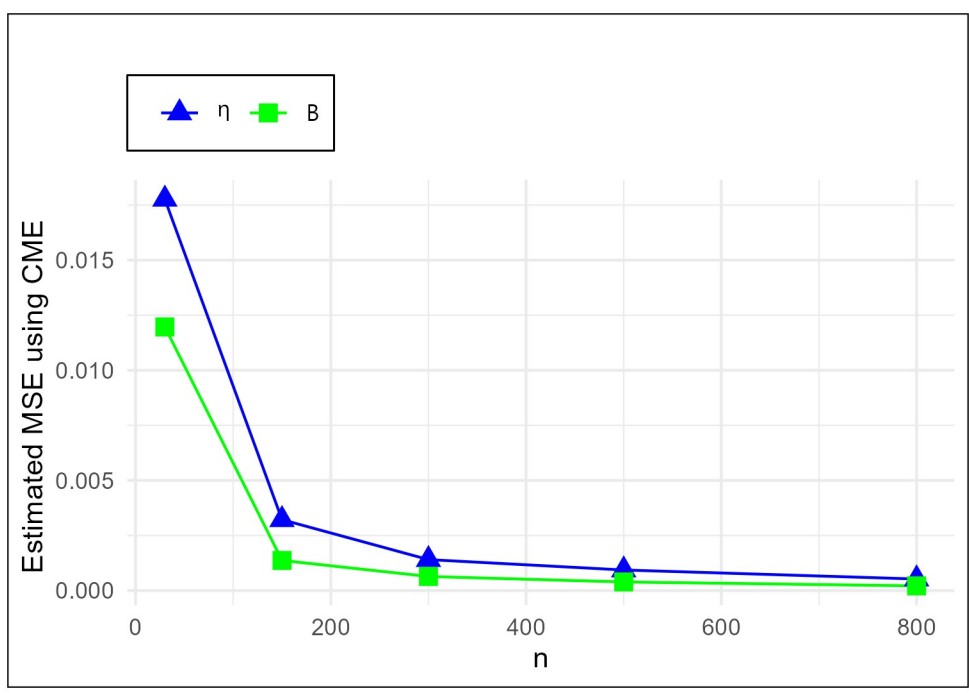

**Fig 13. MSE for CME schemes in Table 4.**

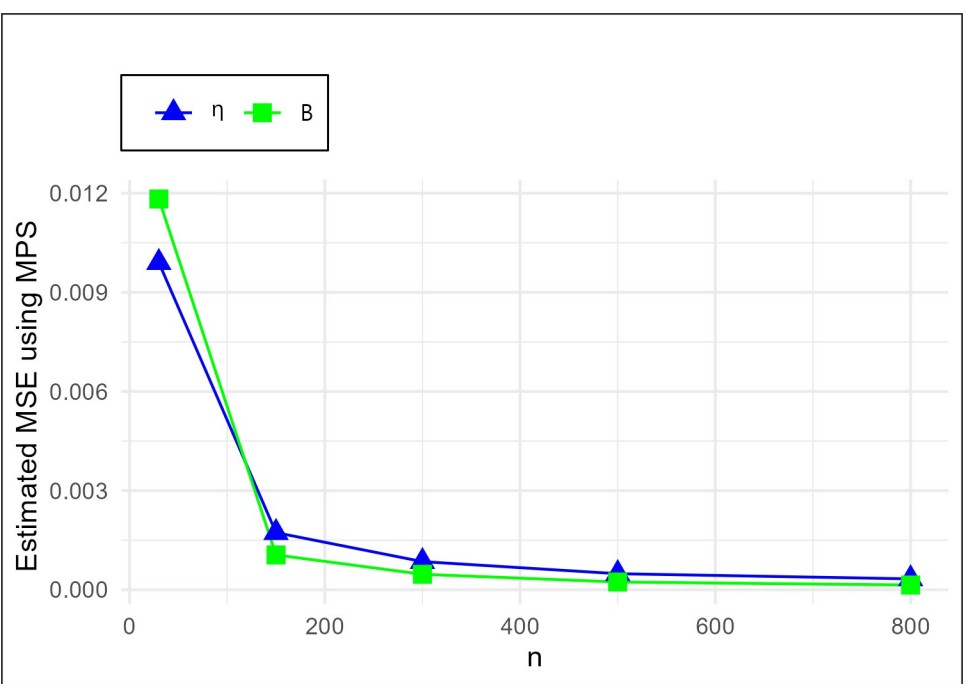

**Fig 14. MSE for MPS schemes in Table 4.**

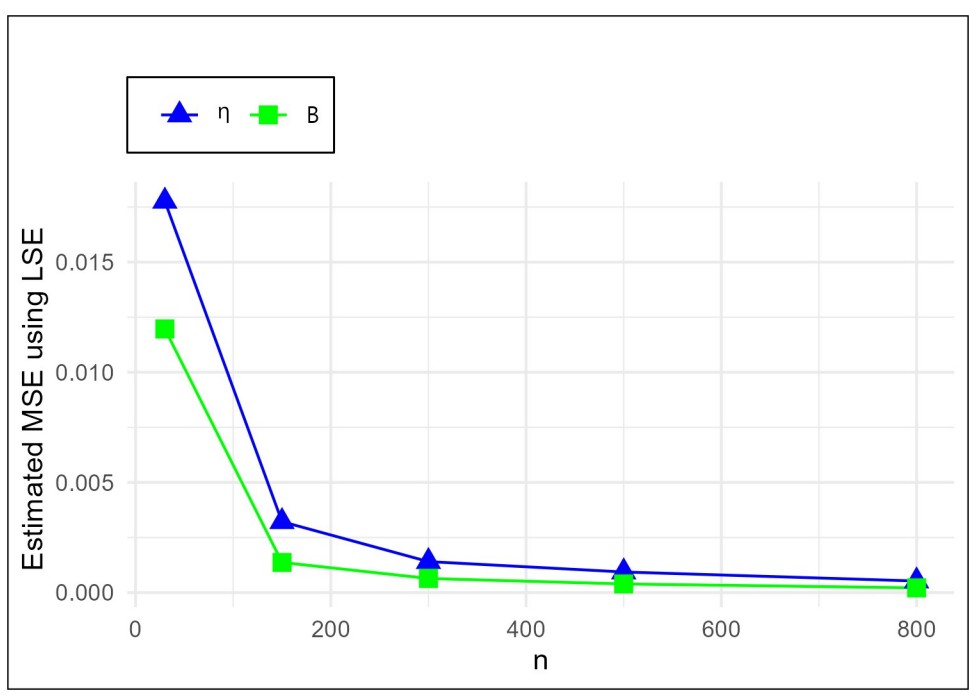

**Fig 15. MSE for LSE schemes in Table 4.**

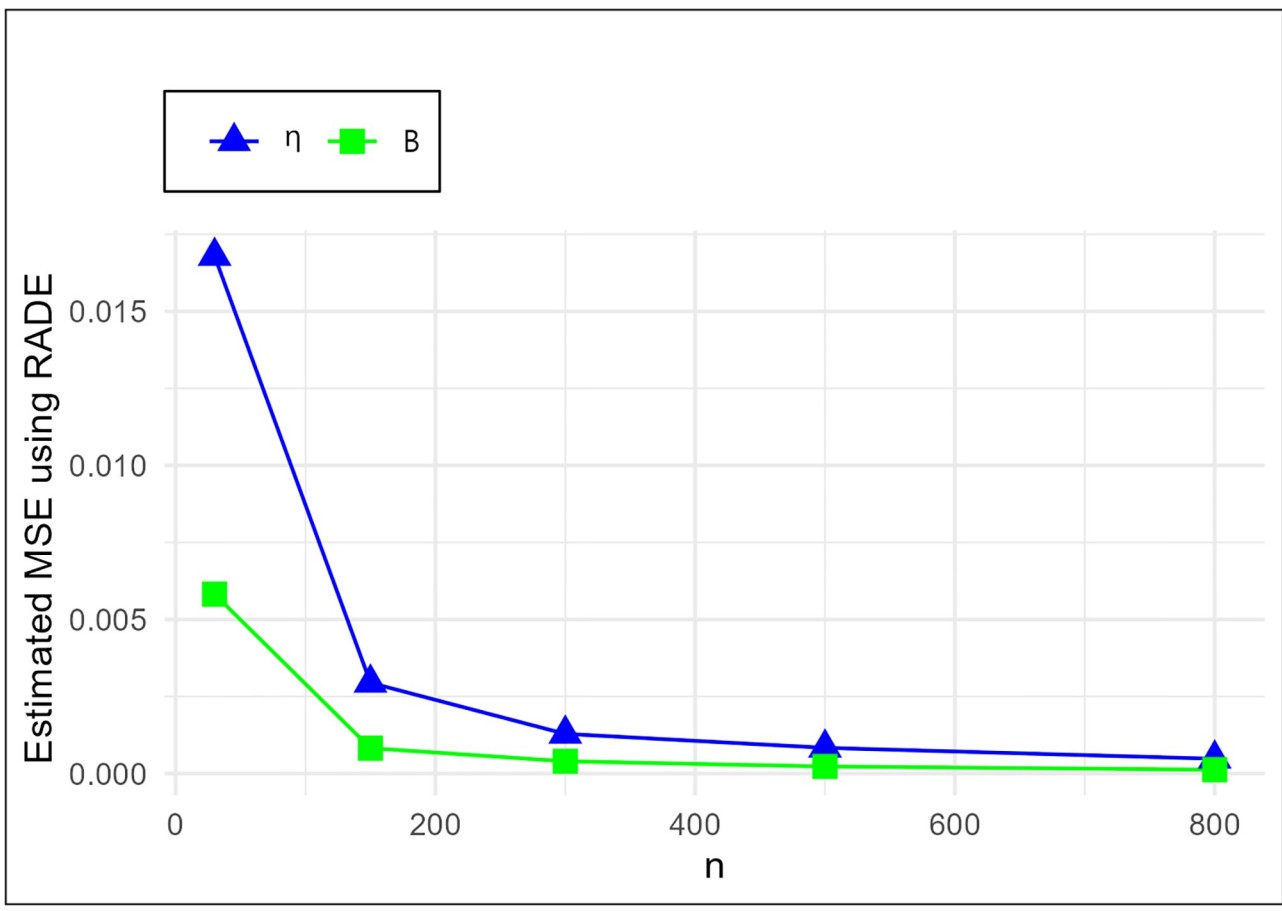

**Fig 16. MSE for RADE,WLSE and LADE schemes in** Table 4.

## 3.2 Moments and moment generating function

Suppose that the MKTL distribution applies to the random variable $X$. The $w^{th}$ moments of $X$ can be calculated by inserting (1) and (2) in (5) as follows:

$$\mu_w^/ = \int_0^1 x^w f(x; \eta, \beta) dx = 2\eta \sum_{i,j=0}^{\infty} \vartheta_{i,j} \int_0^1 x^{w+\eta\beta(i+1)+\eta j \ - \ 1}(1-x^\eta)(2-x^\eta)^{\beta(i+1)+j-1} dx. \quad (9)$$

By employing the binomial expansion to the previous Eq (9) as follows:

$$\mu_w^/ = \sum_{i,j=0}^{\infty} \sum_{k=0}^{\beta(i+1)+j-1} \varepsilon_{i,j} \int_0^1 x^{w+\eta[\beta(i+1)+j+k]-1}(1-x^\eta) dx,$$

where $\varepsilon_{i,j} = (-1)^k \eta 2^{\beta(i+1)+j-k} \vartheta_{i,j} \begin{pmatrix} \beta(i+1)+j-1 \\ k \end{pmatrix}$.

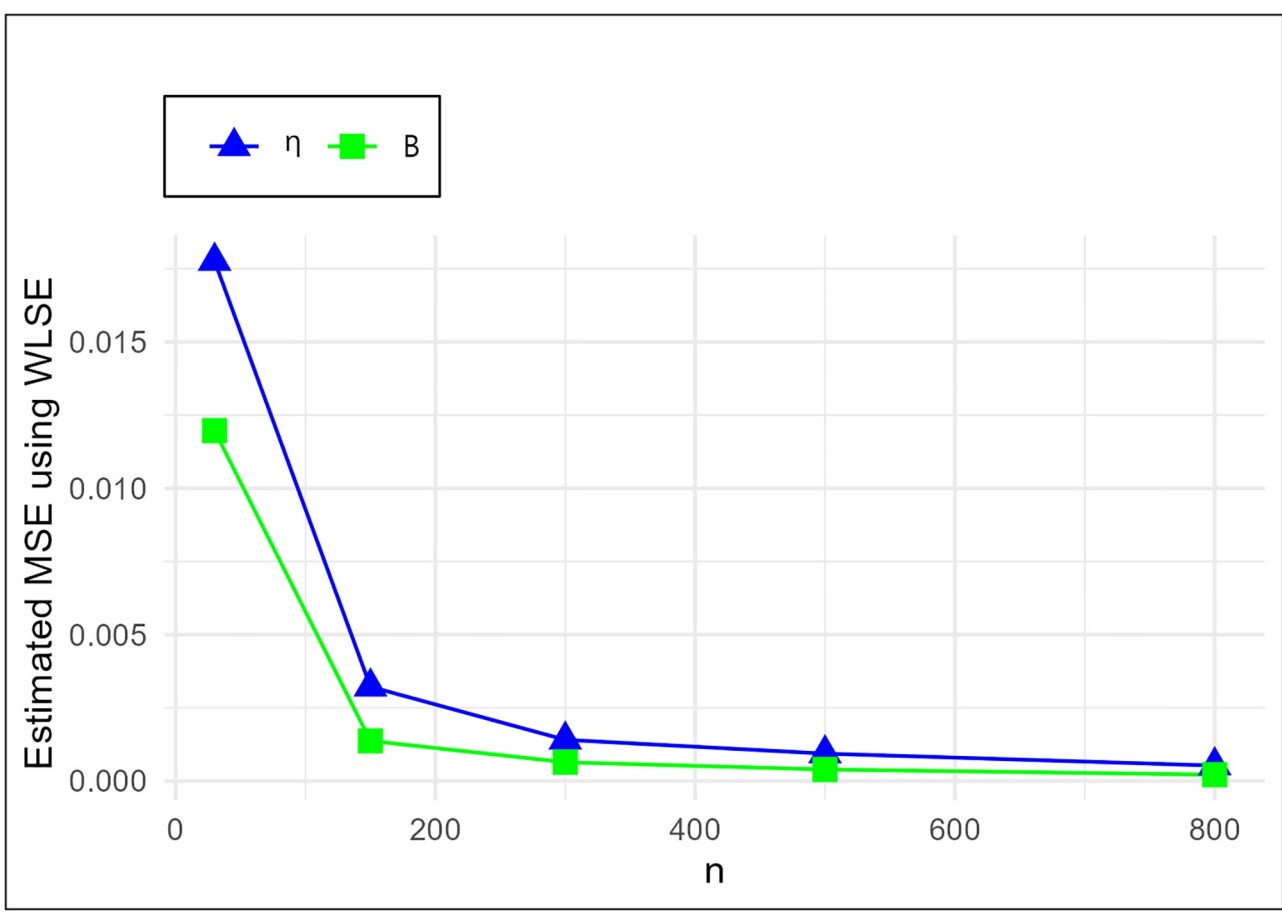

**Fig 17. MSE for WLSE schemes in Table 4.**

Then, the $w^{th}$ moments of the MKTL distribution is given by

$$\mu_w^/ = \sum_{i,j=0}^{\infty} \sum_{k=0}^{\beta(i+1)+j-1} \varepsilon_{i,j} \left[ \frac{1}{w + \eta[\beta(i+1)+j+k]} - \frac{1}{w + \eta[\beta(i+1)+j+k+1]} \right]. \quad (10)$$

The $p^{th}$ incomplete moments of $X$ can be calculated as below:

$$\Xi_p(t) = \int_0^t x^w f(x; \eta, \beta) dx = \sum_{i,j=0}^{\infty} \sum_{k=0}^{\beta(i+1)+j-1} \varepsilon_{i,j} \int_0^t x^{w+\eta[\beta(i+1)+j+k]-1} (1 - x^\eta) dx.$$

Then,

$$\Xi_p(t) = \sum_{i,j=0}^{\infty} \sum_{k=0}^{\beta(i+1)+j-1} \varepsilon_{i,j} \left[ \frac{t^{w+\eta[\beta(i+1)+j+k]}}{w + \eta[\beta(i+1)+j+k]} - \frac{t^{w+\eta[\beta(i+1)+j+k+1]}}{w + \eta[\beta(i+1)+j+k+1]} \right].$$

Table 2 show the numerical values of the moments $\mu_1'$, $\mu_2'$, $\mu_3'$ and $\mu_4'$ also the numerical values of variance ($\sigma^2$), standard deviation ($\sigma$), coefficient of skewness (CS), coefficient of kurtosis (CK) and coefficient of variation (CV) associated with the MKTL distribution.

Figs 7–10 shows the 3D plots of mean, variance, CS, and CK for the MKTL distribution.

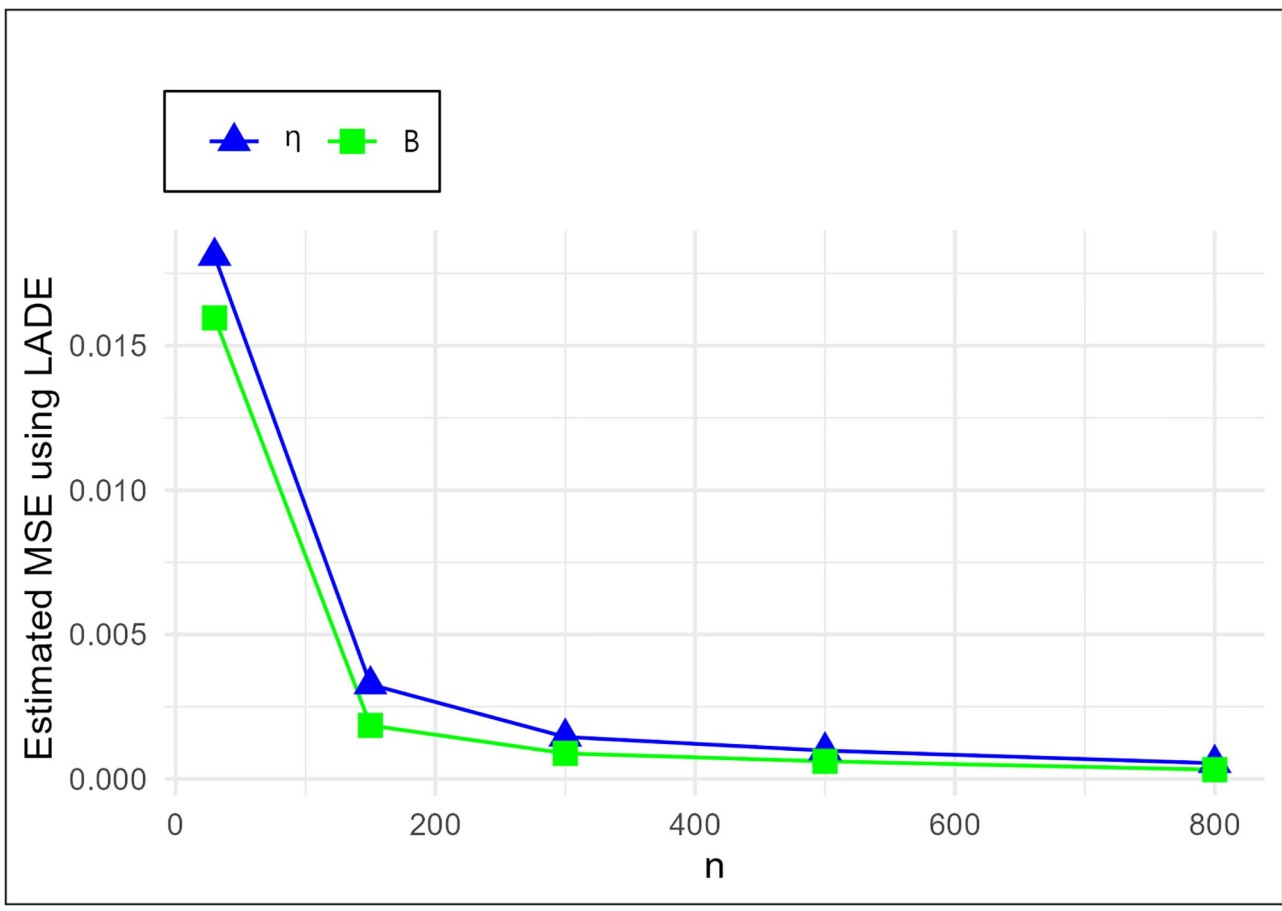

**Fig 18. MSE for LADE schemes in Table 4.**

### 3.3 Order statistics

Suppose that $X_1, X_2, \ldots, X_s$ are $s$ random samples from the MKTL distribution with CDF (6) and PDF (7). Let $X_{(1)}, X_{(2)}, \ldots, X_{(s)}$ are the corresponding order statistics. The PDF of the $m$th order statistics is computed as below:

$$f_{X_{(m)}}(x) = \frac{s!}{(m-1)!(s-m)!} f(x; \beta, \eta)[F(x; \beta, \eta)]^{m-1}[1 - F(x; \beta, \eta)]^{s-m}. \tag{11}$$

Inserting (6) and (7) in (11), we have the PDF of $X_{(m)}$ of order statistics for the MKTL distribution as follows:

$$f_{X_{(m)}}(x) = \frac{2\beta\eta s!}{(m-1)!(s-m)!} \frac{x^{\eta-1}(1-x)(2-x)^{\eta-1}(1-(1-x)^2)^{\eta\beta-\eta}}{[1-(1-(1-x)^2)^{\eta}]^{\beta+1}}$$
$$\times [1 - e^{-[(1-(1-x)^2)^{-\eta}-1]^{-\beta}}]^{m-1} e^{-(s-m+1)[(1-(1-x)^2)^{-\eta}-1]^{-\beta}}. \tag{12}$$

By Putting $m = 1$ and $s$ in (12), we get the smallest order statistics and the largest order statistics

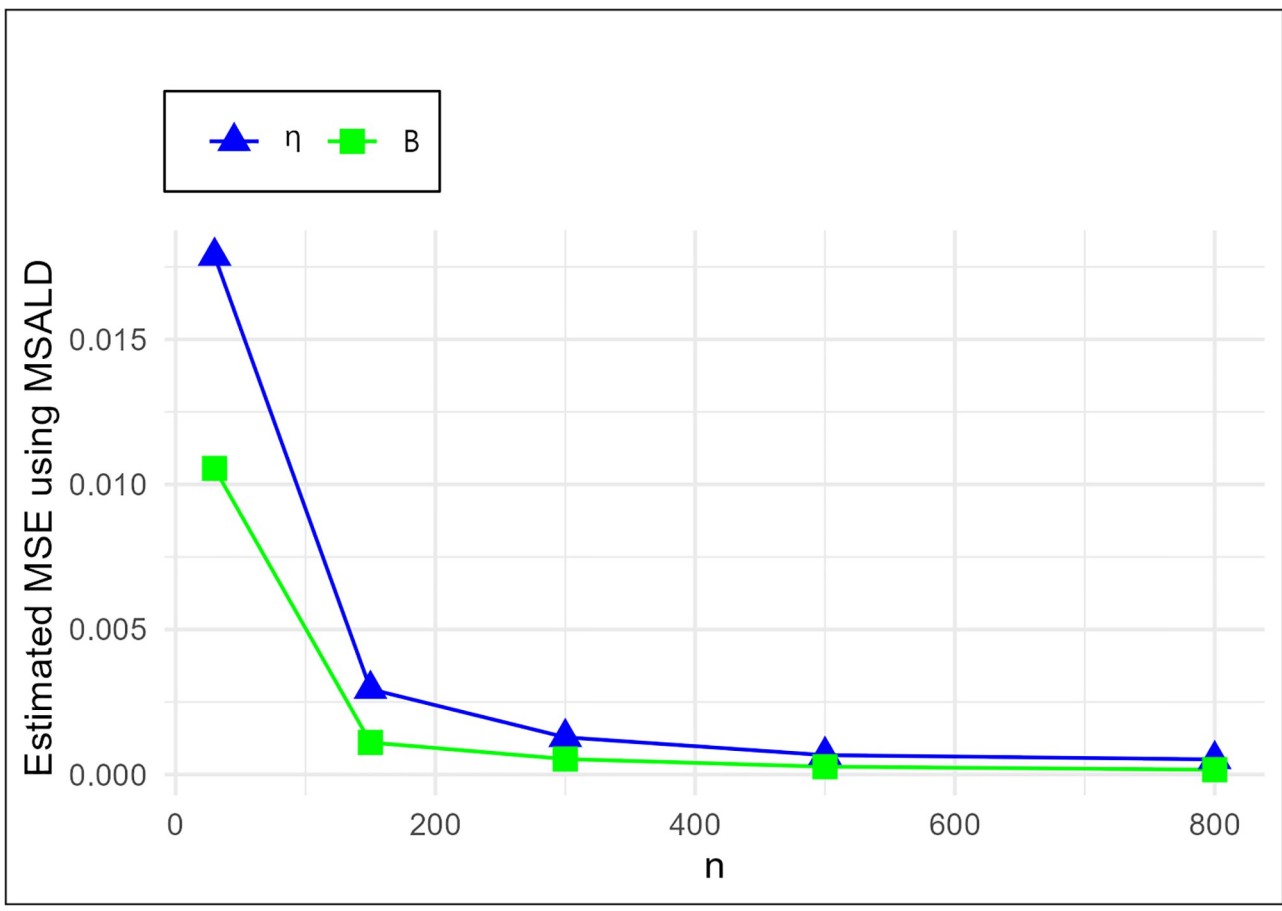

**Fig 19. MSE for MSALD schemes in Table 4.**

for the MKTL distribution as below:

$$f_{X_{(1)}}(x) = \frac{2\beta\eta s x^{\eta-1}(1-x)(2-x)^{\eta-1}\left(1-(1-x)^2\right)^{\eta\beta-\eta}}{\left[1-\left(1-(1-x)^2\right)^{\eta}\right]^{\beta+1}} e^{-s\left[\left(1-(1-x)^2\right)^{-\eta}-1\right]^{-\beta}}, \qquad (13)$$

and

$$f_{X_{(s)}}(x) = \frac{2\beta\eta s x^{\eta-1}(1-x)(2-x)^{\eta-1}\left(1-(1-x)^2\right)^{\eta\beta-\eta}}{\left[1-\left(1-(1-x)^2\right)^{\eta}\right]^{\beta+1}}$$

$$\times \left[1-e^{-\left[\left(1-(1-x)^2\right)^{-\eta}-1\right]^{-\beta}}\right]^{s-1} e^{-\left[\left(1-(1-x)^2\right)^{-\eta}-1\right]^{-\beta}}. \qquad (14)$$

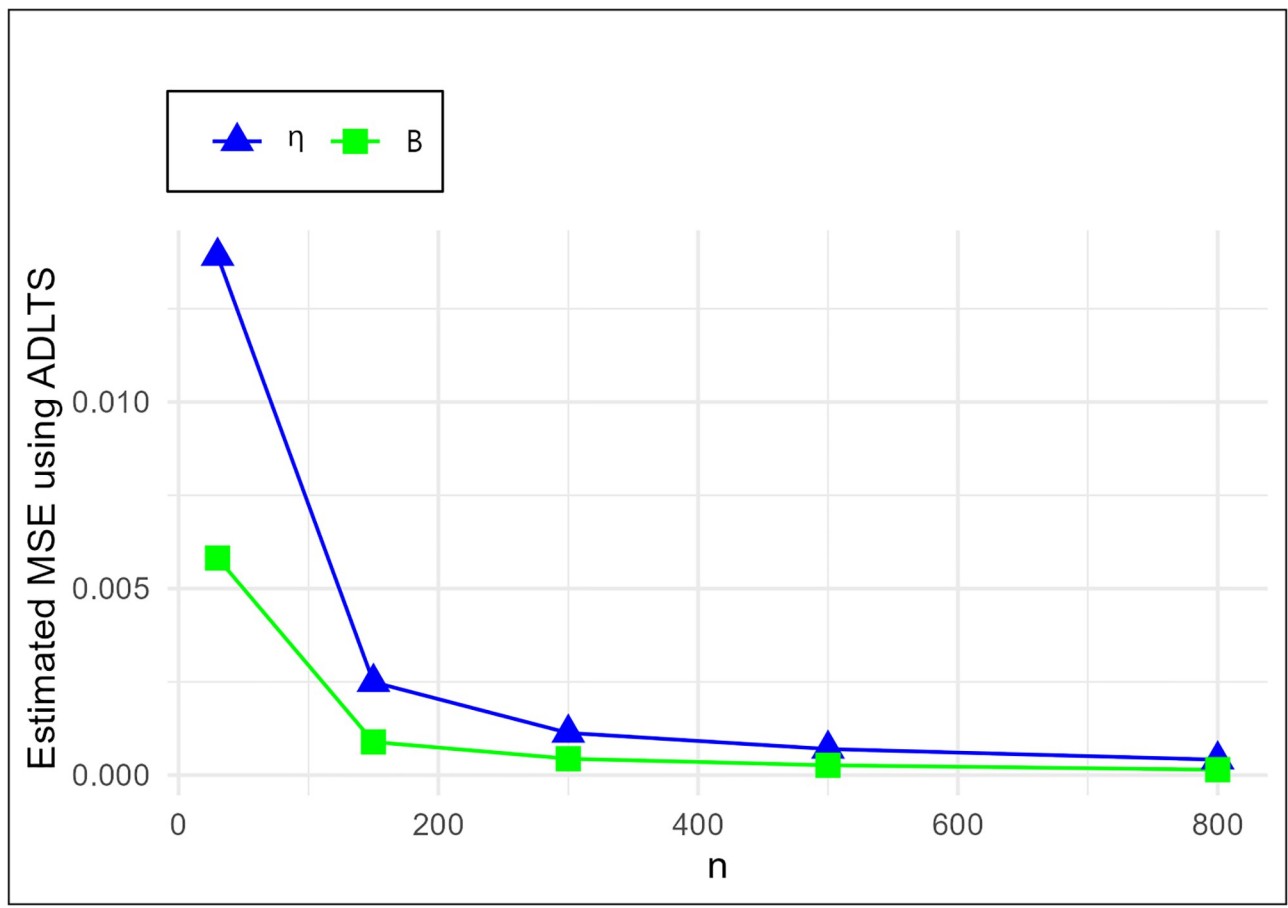

**Fig 20. MSE for ADLTS schemes in Table 4.**

## 4 Estimation methods

### 4.1 Method of maximum likelihood

Maximum likelihood estimation (MLE) see ([26, 27]), maximizes the log-likelihood function to estimate $\eta$ and $\beta$, the log-likelihood function is,

$$l = n\log(2) + n\log(\eta) + n\log(\beta) + (\eta - 1)\sum_{i=1}^{n}\log(x_i) + \sum_{i=1}^{n}\log(1 - x_i)$$

$$+ (\eta - 1)\sum_{i=1}^{n}\log(2 - x_i) + (\eta\beta - \eta)\sum_{i=1}^{n}\log(1 - (1 - x_i)^2)$$

$$- (\beta + 1)\sum_{i=1}^{n}\log\left[1 - (1 - (1 - x_i)^2)^\eta\right] - \sum_{i=1}^{n}\left[(1 - (1 - x_i)^2)^{-\eta} - 1\right]^{-\beta}.$$

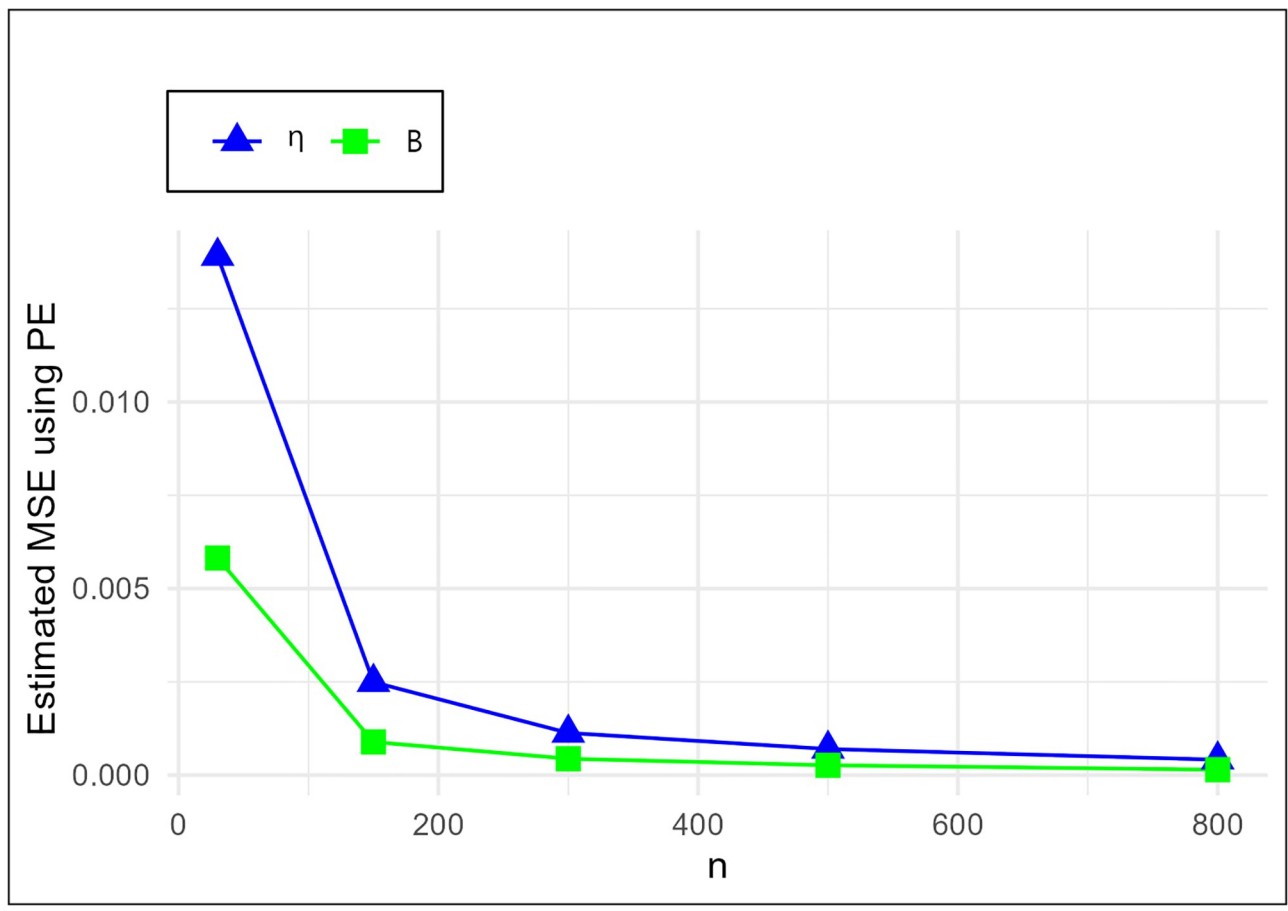

**Fig 21. MSE for PE schemes in Table 4.**

The partial derivatives of $l$ are,

$$\frac{\partial l}{\partial \eta} = \frac{n}{\eta} + \sum_{i=1}^{n} \log(x_i) + \sum_{i=1}^{n} \log(2 - x_i) + (\beta - 1)\sum_{i=1}^{n} \log(1 - (1 - x_i)^2)$$

$$+ (\beta + 1)\frac{(1 - (1 - x_i)^2)^{\eta} \log(1 - (1 - x_i)^2)}{1 - (1 - (1 - x_i)^2)^{\eta}}$$

$$- \beta\sum_{i=1}^{n}[(1 - (1 - x_i)^2)^{-\eta} - 1]^{-\beta-1}(1 - (1 - x_i)^2)^{-\eta}\log(1 - (1 - x_i)^2),$$

and

$$\frac{\partial l}{\partial \beta} = \frac{n}{\beta} + \eta\sum_{i=1}^{n} \log(1 - (1 - x_i)^2) - \sum_{i=1}^{n} \log\left[1 - (1 - (1 - x_i)^2)^{\eta}\right]$$

$$+ \sum_{i=1}^{n}[(1 - (1 - x_i)^2)^{-\eta} - 1]^{-\beta}\log\left[(1 - (1 - x_i)^2)^{-\eta} - 1\right].$$

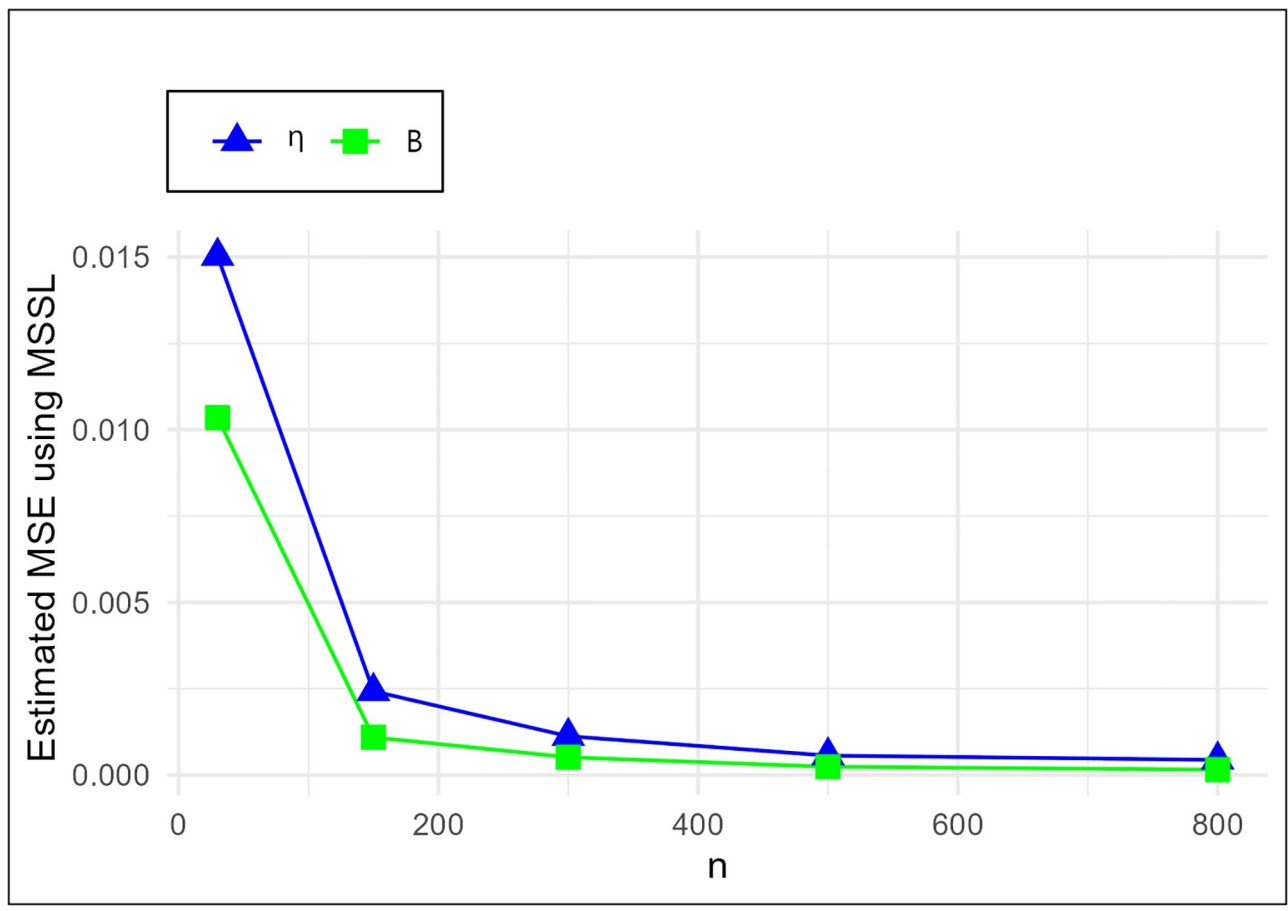

**Fig 22. MSE for MSSL schemes in Table 4.**

To estimate $\eta$ and $\beta$ it is needed to solve the two simultaneous equations $\frac{\partial l}{\partial \eta} = 0$ and $\frac{\partial l}{\partial \beta} = 0$ and that require numerical techniques and using R software.

## 4.2 Method of Anderson-Darling

It was introduced by [28], to estimate $\eta$ and $\beta$ we minimize its function which is

$$A(\eta, \beta) = -n - \frac{1}{n}\sum_{i=1}^{n}(2i-1)[\log F(x_i) + \log S(x_{n-i-1})].$$

The partial derivatives of $A(\eta, \beta)$ are,

$$\frac{\partial A(\eta, \beta)}{\partial \eta} = -\frac{1}{n}\sum_{i=1}^{n}(2i-1)\left[\frac{F_{\eta_i}}{F_i} - \frac{F_{\eta_{n+i-1}}}{1 - F_{n+i-1}}\right],$$

and

$$\frac{\partial A(\eta, \beta)}{\partial \beta} = -\frac{1}{n}\sum_{i=1}^{n}(2i-1)\left[\frac{F_{\beta_i}}{F_i} - \frac{F_{\beta_{n+i-1}}}{1 - F_{n+i-1}}\right],$$

**Table 4. Simulation results at $\eta = 0.4$ and $\beta = 0.4$.**

| n | | 30 | | 150 | | 300 | | 500 | | 800 | |
|---|---|---|---|---|---|---|---|---|---|---|---|
| Estimate | | Mean | MSE | Mean | MSE | Mean | MSE | Mean | MSE | Mean | MSE |
| MLE | $\eta$ | 0.41409 | 0.00926 | 0.40639 | 0.00172 | 0.40917 | 0.00087 | 0.40714 | 0.00049 | 0.40683 | 0.00034 |
| | $\beta$ | 0.42741 | 0.00564 | 0.40729 | 0.00073 | 0.40485 | 0.00035 | 0.40260 | 0.00019 | 0.40345 | 0.00011 |
| ADE | $\eta$ | 0.41762 | 0.01391 | 0.40481 | 0.00249 | 0.40576 | 0.00113 | 0.40307 | 0.00070 | 0.40300 | 0.00041 |
| | $\beta$ | 0.41363 | 0.00582 | 0.40326 | 0.00089 | 0.40246 | 0.00044 | 0.40071 | 0.00026 | 0.40164 | 0.00015 |
| CME | $\eta$ | 0.41524 | 0.01776 | 0.40451 | 0.00322 | 0.40513 | 0.00141 | 0.40245 | 0.00093 | 0.40264 | 0.00053 |
| | $\beta$ | 0.43669 | 0.01197 | 0.40646 | 0.00137 | 0.40357 | 0.00064 | 0.40114 | 0.00039 | 0.40135 | 0.00021 |
| MPS | $\eta$ | 0.40681 | 0.00990 | 0.40393 | 0.00173 | 0.40776 | 0.00085 | 0.40617 | 0.00049 | 0.40625 | 0.00033 |
| | $\beta$ | 0.47101 | 0.01183 | 0.41784 | 0.00105 | 0.41079 | 0.00047 | 0.40653 | 0.00024 | 0.40604 | 0.00014 |
| LSE | $\eta$ | 0.41524 | 0.01776 | 0.40451 | 0.00322 | 0.40513 | 0.00141 | 0.40245 | 0.00093 | 0.40264 | 0.00053 |
| | $\beta$ | 0.43669 | 0.01197 | 0.40646 | 0.00137 | 0.40357 | 0.00064 | 0.40114 | 0.00039 | 0.40135 | 0.00021 |
| RADE | $\eta$ | 0.42229 | 0.01679 | 0.40571 | 0.00294 | 0.40571 | 0.00129 | 0.40282 | 0.00083 | 0.40271 | 0.00048 |
| | $\beta$ | 0.41759 | 0.00582 | 0.40364 | 0.00082 | 0.40207 | 0.00040 | 0.40012 | 0.00023 | 0.40123 | 0.00012 |
| WLSE | $\eta$ | 0.41524 | 0.01776 | 0.40451 | 0.00322 | 0.40513 | 0.00141 | 0.40245 | 0.00093 | 0.40264 | 0.00053 |
| | $\beta$ | 0.43669 | 0.01197 | 0.40646 | 0.00137 | 0.40357 | 0.00064 | 0.40114 | 0.00039 | 0.40135 | 0.00021 |
| LADE | $\eta$ | 0.41158 | 0.01810 | 0.40363 | 0.00327 | 0.40476 | 0.00145 | 0.40210 | 0.00098 | 0.40257 | 0.00054 |
| | $\beta$ | 0.43566 | 0.01596 | 0.40729 | 0.00185 | 0.40516 | 0.00089 | 0.40327 | 0.00061 | 0.40295 | 0.00032 |
| MSALD | $\eta$ | 0.41851 | 0.01787 | 0.40516 | 0.00294 | 0.40462 | 0.00128 | 0.40229 | 0.00067 | 0.40240 | 0.00052 |
| | $\beta$ | 0.42011 | 0.01055 | 0.40279 | 0.00110 | 0.40106 | 0.00053 | 0.40024 | 0.00027 | 0.40093 | 0.00017 |
| ADLTS | $\eta$ | 0.41762 | 0.01391 | 0.40481 | 0.00249 | 0.40576 | 0.00113 | 0.40307 | 0.00070 | 0.40300 | 0.00041 |
| | $\beta$ | 0.41363 | 0.00582 | 0.40326 | 0.00089 | 0.40246 | 0.00044 | 0.40071 | 0.00026 | 0.40164 | 0.00015 |
| PE | $\eta$ | 0.41762 | 0.01391 | 0.40481 | 0.00249 | 0.40576 | 0.00113 | 0.40307 | 0.00070 | 0.40300 | 0.00041 |
| | $\beta$ | 0.41363 | 0.00582 | 0.40326 | 0.00089 | 0.40246 | 0.00044 | 0.40071 | 0.00026 | 0.40164 | 0.00015 |
| MSSL | $\eta$ | 0.40783 | 0.01503 | 0.40134 | 0.00243 | 0.40504 | 0.00113 | 0.40382 | 0.00056 | 0.40257 | 0.00044 |
| | $\beta$ | 0.44418 | 0.01035 | 0.40744 | 0.00109 | 0.40507 | 0.00052 | 0.40129 | 0.00024 | 0.40247 | 0.00016 |

where

$$F_i = F(x_i; \eta, \beta),\tag{15}$$

$$F_{\eta_i} = \frac{\beta(1 - (1 - x)^2)^{-\eta} \log(1 - (1 - x)^2)}{[(1 - (1 - x)^2)^{-\eta} - 1]^{\beta+1}} e^{-[(1-(1-x)^2)^{-\eta}-1]^{-\beta}}\tag{16}$$

and

$$F_{\beta_i} = -\frac{\log[(1 - (1 - x)^2)^{-\eta} - 1]}{[(1 - (1 - x)^2)^{-\eta} - 1]^{\beta}} e^{-[(1-(1-x)^2)^{-\eta}-1]^{-\beta}}\tag{17}$$

## 4.3 Method of Cramér_von_Mises

This method minimizing its function, see [29]

$$C(x_i) = \frac{1}{12n} + \sum_{i=1}^{n}\left[F(x_i) - \frac{2i-1}{2n}\right]^2$$

**Table 5. Simulation results at $\eta = 0.4$ and $\beta = 0.9$.**

| n | | 30 | | 150 | | 300 | | 500 | | 800 | |
|---|---|---|---|---|---|---|---|---|---|---|---|
| Estimate | | Mean | MSE | Mean | MSE | Mean | MSE | Mean | MSE | Mean | MSE |
| MLE | $\eta$ | 0.40196 | 0.00326 | 0.40003 | 0.00066 | 0.39916 | 0.00030 | 0.40085 | 0.00018 | 0.39996 | 0.00011 |
| | $\beta$ | 0.87491 | 0.00649 | 0.90398 | 0.00216 | 0.90560 | 0.00137 | 0.90374 | 0.00092 | 0.90216 | 0.00056 |
| ADE | $\eta$ | 0.40435 | 0.00369 | 0.40039 | 0.00073 | 0.39922 | 0.00034 | 0.40084 | 0.00021 | 0.39996 | 0.00013 |
| | $\beta$ | 0.84906 | 0.01036 | 0.89488 | 0.00276 | 0.90108 | 0.00160 | 0.90076 | 0.00111 | 0.90012 | 0.00069 |
| CME | $\eta$ | 0.40318 | 0.00413 | 0.40007 | 0.00081 | 0.39897 | 0.00039 | 0.40068 | 0.00023 | 0.39988 | 0.00014 |
| | $\beta$ | 0.84431 | 0.01151 | 0.89406 | 0.00338 | 0.90152 | 0.00193 | 0.90186 | 0.00145 | 0.90079 | 0.00090 |
| MPS | $\eta$ | 0.39870 | 0.00336 | 0.39928 | 0.00066 | 0.39877 | 0.00030 | 0.40062 | 0.00018 | 0.39982 | 0.00011 |
| | $\beta$ | 0.90381 | 0.00419 | 0.92052 | 0.00233 | 0.91746 | 0.00161 | 0.91230 | 0.00108 | 0.90795 | 0.00063 |
| LSE | $\eta$ | 0.40318 | 0.00413 | 0.40007 | 0.00081 | 0.39897 | 0.00039 | 0.40068 | 0.00023 | 0.39988 | 0.00014 |
| | $\beta$ | 0.84431 | 0.01151 | 0.89406 | 0.00338 | 0.90152 | 0.00193 | 0.90186 | 0.00145 | 0.90079 | 0.00090 |
| RADE | $\eta$ | 0.40578 | 0.00397 | 0.40061 | 0.00077 | 0.39928 | 0.00037 | 0.40088 | 0.00022 | 0.39998 | 0.00013 |
| | $\beta$ | 0.85271 | 0.00928 | 0.89549 | 0.00270 | 0.90130 | 0.00172 | 0.90064 | 0.00116 | 0.90026 | 0.00074 |
| WLSE | $\eta$ | 0.40318 | 0.00413 | 0.40007 | 0.00081 | 0.39897 | 0.00039 | 0.40068 | 0.00023 | 0.39988 | 0.00014 |
| | $\beta$ | 0.84431 | 0.01151 | 0.89406 | 0.00338 | 0.90152 | 0.00193 | 0.90186 | 0.00145 | 0.90079 | 0.00090 |
| LADE | $\eta$ | 0.40243 | 0.00435 | 0.39980 | 0.00086 | 0.39880 | 0.00042 | 0.40059 | 0.00025 | 0.39980 | 0.00015 |
| | $\beta$ | 0.82711 | 0.01613 | 0.88830 | 0.00396 | 0.90022 | 0.00227 | 0.90122 | 0.00162 | 0.90115 | 0.00106 |
| MSALD | $\eta$ | 0.40761 | 0.00601 | 0.40047 | 0.00097 | 0.39880 | 0.00042 | 0.40105 | 0.00026 | 0.39975 | 0.00016 |
| | $\beta$ | 0.84038 | 0.01403 | 0.89110 | 0.00347 | 0.89846 | 0.00196 | 0.89837 | 0.00129 | 0.89887 | 0.00079 |
| ADLTS | $\eta$ | 0.40435 | 0.00369 | 0.40039 | 0.00073 | 0.39922 | 0.00034 | 0.40084 | 0.00021 | 0.39996 | 0.00013 |
| | $\beta$ | 0.84906 | 0.01036 | 0.89488 | 0.00276 | 0.90108 | 0.00160 | 0.90076 | 0.00111 | 0.90012 | 0.00069 |
| PE | $\eta$ | 0.40435 | 0.00369 | 0.40039 | 0.00073 | 0.39922 | 0.00034 | 0.40084 | 0.00021 | 0.39996 | 0.00013 |
| | $\beta$ | 0.84906 | 0.01036 | 0.89488 | 0.00276 | 0.90108 | 0.00160 | 0.90076 | 0.00111 | 0.90012 | 0.00069 |
| MSSL | $\eta$ | 0.40160 | 0.00481 | 0.40042 | 0.00086 | 0.39857 | 0.00038 | 0.40079 | 0.00024 | 0.40010 | 0.00014 |
| | $\beta$ | 0.85931 | 0.00954 | 0.89992 | 0.00290 | 0.90349 | 0.00179 | 0.90226 | 0.00123 | 0.90123 | 0.00078 |

Finding partial derivative with respect to $\eta$ and $\beta$, using Eqs (15), (16) and (17),

$$\frac{\partial C(x_i)}{\partial \eta} = 2 \sum_{i=1}^{n} F_{\eta_i} \left[ F_i - \frac{2i-1}{2n} \right]$$

$$\frac{\partial C(x_i)}{\partial \beta} = 2 \sum_{i=1}^{n} F_{\beta_i} \left[ F_i - \frac{2i-1}{2n} \right],$$

then solving the two simultaneous equations $\frac{\partial C(x_i)}{\partial \eta} = 0$ and $\frac{\partial C(x_i)}{\partial \beta} = 0$ this requires numerical techniques and using R software.

## 4.4 Method of maximum product of spacings

It minizes the MPS function to estimate $\eta$ and $\beta$, for more about MPS see [30],

$$\delta(x_i) = \frac{1}{n+1} \sum_{i=1}^{n+1} \log \Lambda_i(x_i),$$

where $\Lambda_i(x_i) = F(x_i) - F(x_{i-1})$, $F(x_0) = 0$ and $F(x_{n+1}) = 1$, using Eqs (15), (16) and (17), the

**Table 6. Simulation results at $\eta = 0.9$ and $\beta = 0.4$.**

| n | | 30 | | 150 | | 300 | | 500 | | 800 | |
|---|---|---|---|---|---|---|---|---|---|---|---|
| Estimate | | Mean | MSE | Mean | MSE | Mean | MSE | Mean | MSE | Mean | MSE |
| MLE | $\eta$ | 0.80619 | 0.02412 | 0.88074 | 0.00502 | 0.89063 | 0.00307 | 0.89749 | 0.00217 | 0.89950 | 0.00137 |
| | $\beta$ | 0.42353 | 0.00533 | 0.40532 | 0.00071 | 0.40324 | 0.00036 | 0.40224 | 0.00022 | 0.40105 | 0.00013 |
| ADE | $\eta$ | 0.79036 | 0.03006 | 0.86994 | 0.00697 | 0.88435 | 0.00395 | 0.89720 | 0.00283 | 0.89962 | 0.00192 |
| | $\beta$ | 0.40783 | 0.00550 | 0.40199 | 0.00092 | 0.40140 | 0.00049 | 0.40050 | 0.00030 | 0.39979 | 0.00018 |
| CME | $\eta$ | 0.77099 | 0.03960 | 0.85888 | 0.00910 | 0.87906 | 0.00505 | 0.89419 | 0.00343 | 0.89829 | 0.00239 |
| | $\beta$ | 0.42937 | 0.01083 | 0.40536 | 0.00146 | 0.40340 | 0.00074 | 0.40114 | 0.00044 | 0.40039 | 0.00025 |
| MPS | $\eta$ | 0.79685 | 0.02692 | 0.87818 | 0.00527 | 0.88906 | 0.00316 | 0.89639 | 0.00219 | 0.89879 | 0.00139 |
| | $\beta$ | 0.46522 | 0.01072 | 0.41621 | 0.00101 | 0.40935 | 0.00046 | 0.40624 | 0.00027 | 0.40370 | 0.00015 |
| LSE | $\eta$ | 0.77099 | 0.03960 | 0.85888 | 0.00910 | 0.87906 | 0.00505 | 0.89419 | 0.00343 | 0.89829 | 0.00239 |
| | $\beta$ | 0.42937 | 0.01083 | 0.40536 | 0.00146 | 0.40340 | 0.00074 | 0.40114 | 0.00044 | 0.40039 | 0.00025 |
| RADE | $\eta$ | 0.78639 | 0.03238 | 0.86382 | 0.00782 | 0.88156 | 0.00440 | 0.89583 | 0.00313 | 0.89925 | 0.00217 |
| | $\beta$ | 0.41376 | 0.00553 | 0.40269 | 0.00084 | 0.40215 | 0.00044 | 0.40105 | 0.00028 | 0.40003 | 0.00016 |
| WLSE | $\eta$ | 0.77099 | 0.03960 | 0.85888 | 0.00910 | 0.87906 | 0.00505 | 0.89419 | 0.00343 | 0.89829 | 0.00239 |
| | $\beta$ | 0.42937 | 0.01083 | 0.40536 | 0.00146 | 0.40340 | 0.00074 | 0.40114 | 0.00044 | 0.40039 | 0.00025 |
| LADE | $\eta$ | 0.76372 | 0.04329 | 0.85608 | 0.00981 | 0.87829 | 0.00533 | 0.89328 | 0.00356 | 0.89746 | 0.00245 |
| | $\beta$ | 0.42558 | 0.01405 | 0.40600 | 0.00207 | 0.40273 | 0.00101 | 0.40091 | 0.00059 | 0.40028 | 0.00035 |
| MSALD | $\eta$ | 0.76596 | 0.03871 | 0.86660 | 0.00755 | 0.88391 | 0.00412 | 0.89326 | 0.00276 | 0.89971 | 0.00197 |
| | $\beta$ | 0.41598 | 0.00927 | 0.40193 | 0.00115 | 0.40055 | 0.00051 | 0.40086 | 0.00033 | 0.39916 | 0.00021 |
| ADLTS | $\eta$ | 0.79036 | 0.03006 | 0.86994 | 0.00697 | 0.88435 | 0.00395 | 0.89720 | 0.00283 | 0.89962 | 0.00192 |
| | $\beta$ | 0.40783 | 0.00550 | 0.40199 | 0.00092 | 0.40140 | 0.00049 | 0.40050 | 0.00030 | 0.39979 | 0.00018 |
| PE | $\eta$ | 0.79036 | 0.03006 | 0.86994 | 0.00697 | 0.88435 | 0.00395 | 0.89720 | 0.00283 | 0.89962 | 0.00192 |
| | $\beta$ | 0.40783 | 0.00550 | 0.40199 | 0.00092 | 0.40140 | 0.00049 | 0.40050 | 0.00030 | 0.39979 | 0.00018 |
| MSSL | $\eta$ | 0.76641 | 0.03717 | 0.86815 | 0.00699 | 0.88586 | 0.00399 | 0.89618 | 0.00251 | 0.89982 | 0.00178 |
| | $\beta$ | 0.43802 | 0.00984 | 0.40680 | 0.00106 | 0.40311 | 0.00051 | 0.40199 | 0.00030 | 0.40039 | 0.00018 |

partial derivatives with respect to the parameters $\eta$ and $\beta$ are,

$$\frac{\partial \delta(x_i)}{\partial \eta} = \frac{1}{n+1} \sum_{i=1}^{n+1} \frac{F_{\eta_i} - F_{\eta_{i-1}}}{F_i - F_{i-1}}$$

$$\frac{\partial \delta(x_i)}{\partial \beta} = \frac{1}{n+1} \sum_{i=1}^{n+1} \frac{F_{\beta_i} - F_{\beta_{i-1}}}{F_i - F_{i-1}},$$

by solving the equations $\frac{\partial \delta(x_i)}{\partial \eta} = 0$ and $\frac{\partial \delta(x_i)}{\partial \beta} = 0$ we get the estimates of $\eta$ and $\beta$, that requires using R software.

## 4.5 Methods of least squares

By minimizing its function see [31], its function is

$$V(x_i) = \sum_{i=1}^{n} \left[ F(x_i) - \frac{i}{n+1} \right]^2,$$

using the Eqs (15), (16) and (17), the partial derivatives with respect to the parameters $\eta$ and $\beta$

**Table 7. Simulation results at $\eta = 0.9$ and $\beta = 0.9$.**

| n | | 30 | | 150 | | 300 | | 500 | | 800 | |
|---|---|---|---|---|---|---|---|---|---|---|---|
| Estimate | | Mean | MSE | Mean | MSE | Mean | MSE | Mean | MSE | Mean | MSE |
| MLE | $\eta$ | 0.85919 | 0.00938 | 0.89377 | 0.00253 | 0.90117 | 0.00142 | 0.90089 | 0.00095 | 0.89944 | 0.00060 |
| | $\beta$ | 0.87102 | 0.00709 | 0.90208 | 0.00226 | 0.90552 | 0.00138 | 0.90393 | 0.00085 | 0.90293 | 0.00057 |
| ADE | $\eta$ | 0.85633 | 0.00994 | 0.89441 | 0.00280 | 0.90112 | 0.00162 | 0.90122 | 0.00107 | 0.89945 | 0.00069 |
| | $\beta$ | 0.84996 | 0.01080 | 0.89314 | 0.00273 | 0.90111 | 0.00157 | 0.90125 | 0.00104 | 0.90127 | 0.00070 |
| CME | $\eta$ | 0.84795 | 0.01150 | 0.89256 | 0.00306 | 0.90005 | 0.00181 | 0.90095 | 0.00119 | 0.89917 | 0.00077 |
| | $\beta$ | 0.84738 | 0.01208 | 0.89209 | 0.00326 | 0.90152 | 0.00191 | 0.90277 | 0.00133 | 0.90252 | 0.00092 |
| MPS | $\eta$ | 0.85506 | 0.01012 | 0.89218 | 0.00257 | 0.90035 | 0.00143 | 0.90037 | 0.00095 | 0.89910 | 0.00060 |
| | $\beta$ | 0.89832 | 0.00494 | 0.91974 | 0.00248 | 0.91764 | 0.00164 | 0.91263 | 0.00102 | 0.90877 | 0.00065 |
| LSE | $\eta$ | 0.84795 | 0.01150 | 0.89256 | 0.00306 | 0.90005 | 0.00181 | 0.90095 | 0.00119 | 0.89917 | 0.00077 |
| | $\beta$ | 0.84738 | 0.01208 | 0.89209 | 0.00326 | 0.90152 | 0.00191 | 0.90277 | 0.00133 | 0.90252 | 0.00092 |
| RADE | $\eta$ | 0.85364 | 0.01022 | 0.89376 | 0.00289 | 0.90119 | 0.00173 | 0.90130 | 0.00113 | 0.89948 | 0.00073 |
| | $\beta$ | 0.84828 | 0.01103 | 0.89443 | 0.00285 | 0.90141 | 0.00164 | 0.90266 | 0.00111 | 0.90128 | 0.00072 |
| WLSE | $\eta$ | 0.84795 | 0.01150 | 0.89256 | 0.00306 | 0.90005 | 0.00181 | 0.90095 | 0.00119 | 0.89917 | 0.00077 |
| | $\beta$ | 0.84738 | 0.01208 | 0.89209 | 0.00326 | 0.90152 | 0.00191 | 0.90277 | 0.00133 | 0.90252 | 0.00092 |
| LADE | $\eta$ | 0.84500 | 0.01245 | 0.89162 | 0.00327 | 0.89930 | 0.00191 | 0.90077 | 0.00127 | 0.89908 | 0.00083 |
| | $\beta$ | 0.83600 | 0.01436 | 0.88750 | 0.00371 | 0.89935 | 0.00228 | 0.90054 | 0.00166 | 0.90229 | 0.00111 |
| MSALD | $\eta$ | 0.83456 | 0.01477 | 0.88821 | 0.00343 | 0.90046 | 0.00199 | 0.90071 | 0.00131 | 0.90008 | 0.00087 |
| | $\beta$ | 0.83534 | 0.01451 | 0.88740 | 0.00332 | 0.90157 | 0.00198 | 0.90012 | 0.00133 | 0.89983 | 0.00086 |
| ADLTS | $\eta$ | 0.85633 | 0.00994 | 0.89441 | 0.00280 | 0.90112 | 0.00162 | 0.90122 | 0.00107 | 0.89945 | 0.00069 |
| | $\beta$ | 0.84996 | 0.01080 | 0.89314 | 0.00273 | 0.90111 | 0.00157 | 0.90125 | 0.00104 | 0.90127 | 0.00070 |
| PE | $\eta$ | 0.85633 | 0.00994 | 0.89441 | 0.00280 | 0.90112 | 0.00162 | 0.90122 | 0.00107 | 0.89945 | 0.00069 |
| | $\beta$ | 0.84996 | 0.01080 | 0.89314 | 0.00273 | 0.90111 | 0.00157 | 0.90125 | 0.00104 | 0.90127 | 0.00070 |
| MSSL | $\eta$ | 0.83606 | 0.01445 | 0.88984 | 0.00329 | 0.90049 | 0.00178 | 0.89902 | 0.00114 | 0.89935 | 0.00077 |
| | $\beta$ | 0.85590 | 0.01129 | 0.89788 | 0.00289 | 0.90577 | 0.00178 | 0.90377 | 0.00119 | 0.90147 | 0.00072 |

are,

$$\frac{\partial V(x_i)}{\partial \eta} = 2\sum_{i=1}^{n} F_{\eta_i}\left[F_i - \frac{i}{n+1}\right]$$

$$\frac{\partial V(x_i)}{\partial \beta} = 2\sum_{i=1}^{n} F_{\beta_i}\left[F_i - \frac{i}{n+1}\right]$$

by solving the two simultaneous equations $\frac{\partial V(x_i)}{\partial \eta} = 0$ and $\frac{\partial V(x_i)}{\partial \beta}$ we get the estimates of $\eta$ and $\beta$, which needs numerical techniques and using R software.

## 4.6 Methods of right_tail Anderson_Darling

RADE was provided by [28], to estimate $\eta$ and $\beta$ we minimize its function which is

$$R(x_i) = \frac{n}{2} - 2\sum_{i=1}^{n} F(x_i) - \frac{1}{n}\sum_{i=1}^{n}(2i-1)\log S(xi),$$

**Table 8. Simulation results at $\eta = 0.9$ and $\beta = 1.4$.**

| n | | 30 | | 150 | | 300 | | 500 | | 800 | |
|---|---|---|---|---|---|---|---|---|---|---|---|
| Estimate | | Mean | MSE | Mean | MSE | Mean | MSE | Mean | MSE | Mean | MSE |
| MLE | $\eta$ | 0.89825 | 0.00692 | 0.90012 | 0.00134 | 0.89987 | 0.00072 | 0.89942 | 0.00047 | 0.89972 | 0.00028 |
| | $\beta$ | 1.47739 | 0.05144 | 1.41258 | 0.00710 | 1.40877 | 0.00337 | 1.40714 | 0.00229 | 1.40373 | 0.00144 |
| ADE | $\eta$ | 0.90041 | 0.00717 | 0.90082 | 0.00145 | 0.90017 | 0.00077 | 0.89959 | 0.00051 | 0.89981 | 0.00031 |
| | $\beta$ | 1.42580 | 0.04882 | 1.40321 | 0.00872 | 1.40338 | 0.00421 | 1.40307 | 0.00266 | 1.40056 | 0.00174 |
| CME | $\eta$ | 0.89686 | 0.00784 | 0.90037 | 0.00160 | 0.90001 | 0.00085 | 0.89947 | 0.00056 | 0.89968 | 0.00034 |
| | $\beta$ | 1.47338 | 0.07456 | 1.41333 | 0.01218 | 1.40762 | 0.00593 | 1.40573 | 0.00349 | 1.40248 | 0.00228 |
| MPS | $\eta$ | 0.89230 | 0.00688 | 0.89886 | 0.00135 | 0.89918 | 0.00072 | 0.89901 | 0.00047 | 0.89947 | 0.00028 |
| | $\beta$ | 1.61321 | 0.10499 | 1.44948 | 0.01012 | 1.42970 | 0.00436 | 1.42058 | 0.00274 | 1.41269 | 0.00163 |
| LSE | $\eta$ | 0.89686 | 0.00784 | 0.90037 | 0.00160 | 0.90001 | 0.00085 | 0.89947 | 0.00056 | 0.89968 | 0.00034 |
| | $\beta$ | 1.47338 | 0.07456 | 1.41333 | 0.01218 | 1.40762 | 0.00593 | 1.40573 | 0.00349 | 1.40248 | 0.00228 |
| RADE | $\eta$ | 0.90126 | 0.00745 | 0.90112 | 0.00152 | 0.90033 | 0.00081 | 0.89968 | 0.00053 | 0.89984 | 0.00033 |
| | $\beta$ | 1.44067 | 0.06092 | 1.40789 | 0.00977 | 1.40550 | 0.00452 | 1.40436 | 0.00291 | 1.40103 | 0.00186 |
| WLSE | $\eta$ | 0.89692 | 0.00784 | 0.90037 | 0.00160 | 0.90001 | 0.00085 | 0.89947 | 0.00056 | 0.89968 | 0.00034 |
| | $\beta$ | 1.47294 | 0.07437 | 1.41333 | 0.01218 | 1.40762 | 0.00593 | 1.40573 | 0.00349 | 1.40248 | 0.00228 |
| LADE | $\eta$ | 0.89543 | 0.00842 | 0.90025 | 0.00171 | 0.89998 | 0.00090 | 0.89943 | 0.00060 | 0.89964 | 0.00037 |
| | $\beta$ | 1.46516 | 0.08109 | 1.40850 | 0.01365 | 1.40548 | 0.00645 | 1.40459 | 0.00404 | 1.40196 | 0.00263 |
| MSALD | $\eta$ | 0.90247 | 0.01086 | 0.90129 | 0.00214 | 0.89981 | 0.00104 | 0.90001 | 0.00066 | 0.89984 | 0.00039 |
| | $\beta$ | 1.46156 | 0.08575 | 1.40591 | 0.01209 | 1.40349 | 0.00539 | 1.40073 | 0.00326 | 1.39866 | 0.00208 |
| ADLTS | $\eta$ | 0.90041 | 0.00717 | 0.90082 | 0.00145 | 0.90017 | 0.00077 | 0.89959 | 0.00051 | 0.89981 | 0.00031 |
| | $\beta$ | 1.42580 | 0.04882 | 1.40321 | 0.00872 | 1.40338 | 0.00421 | 1.40307 | 0.00266 | 1.40056 | 0.00174 |
| PE | $\eta$ | 0.90041 | 0.00717 | 0.90082 | 0.00145 | 0.90017 | 0.00077 | 0.89959 | 0.00051 | 0.89981 | 0.00031 |
| | $\beta$ | 1.42580 | 0.04882 | 1.40321 | 0.00872 | 1.40338 | 0.00421 | 1.40307 | 0.00266 | 1.40056 | 0.00174 |
| MSSL | $\eta$ | 0.89771 | 0.00964 | 0.89879 | 0.00179 | 0.89915 | 0.00089 | 0.89978 | 0.00058 | 0.89951 | 0.00036 |
| | $\beta$ | 1.50804 | 0.08582 | 1.41697 | 0.01036 | 1.41178 | 0.00492 | 1.40618 | 0.00308 | 1.40211 | 0.00187 |

to find the partial derivative we use using Eqs (15), (16) and (17).

$$\frac{\partial R(x_i)}{\partial \eta} = -2\sum_{i=1}^{n} F_{\eta_i} + \frac{1}{n}\sum_{i=1}^{n}(2i-1)\frac{F_{\eta_i}}{1-F_i}$$

$$\frac{\partial R(x_i)}{\partial \beta} = -2\sum_{i=1}^{n} F_{\beta_i} + \frac{1}{n}\sum_{i=1}^{n}(2i-1)\frac{F_{\beta_i}}{1-F_i}.$$

by solving the two simultaneous equations $\frac{\partial R(x_i)}{\partial \eta} = 0$ and $\frac{\partial R(x_i)}{\partial \beta} = 0$ no closed form here and numerical methods are applied depending on R software.

## 4.7 Methods of weighted least squares

WLSE (see [31]) minimizes its function which is:

$$W(x_i) = \sum_{i=1}^{n} \frac{(n+1)^2(n+2)}{i(n-i+1)}\left[F(x_i) - \frac{i}{n+1}\right]^2.$$

**Table 9. Simulation results at $\eta = 1.4$ and $\beta = 0.9$.**

| n | | 30 | | 150 | | 300 | | 500 | | 800 | |
|---|---|---|---|---|---|---|---|---|---|---|---|
| Estimate | | Mean | MSE | Mean | MSE | Mean | MSE | Mean | MSE | Mean | MSE |
| MLE | $\eta$ | 1.40636 | 0.03671 | 1.40244 | 0.00713 | 1.39962 | 0.00375 | 1.40121 | 0.00218 | 1.40092 | 0.00134 |
| | $\beta$ | 0.93988 | 0.01881 | 0.91171 | 0.00310 | 0.90450 | 0.00140 | 0.90476 | 0.00102 | 0.90225 | 0.00054 |
| ADE | $\eta$ | 1.41387 | 0.04120 | 1.40453 | 0.00821 | 1.39966 | 0.00421 | 1.40303 | 0.00250 | 1.40093 | 0.00154 |
| | $\beta$ | 0.90881 | 0.01828 | 0.90426 | 0.00369 | 0.90084 | 0.00174 | 0.90104 | 0.00123 | 0.89985 | 0.00064 |
| CME | $\eta$ | 1.40757 | 0.04601 | 1.40374 | 0.00928 | 1.39891 | 0.00474 | 1.40313 | 0.00285 | 1.40077 | 0.00173 |
| | $\beta$ | 0.94511 | 0.03177 | 0.91003 | 0.00522 | 0.90410 | 0.00245 | 0.90272 | 0.00168 | 0.90087 | 0.00086 |
| MPS | $\eta$ | 1.39406 | 0.03720 | 1.40033 | 0.00719 | 1.39816 | 0.00376 | 1.40040 | 0.00218 | 1.40036 | 0.00135 |
| | $\beta$ | 1.03042 | 0.03937 | 0.93552 | 0.00450 | 0.91794 | 0.00176 | 0.91344 | 0.00122 | 0.90803 | 0.00061 |
| LSE | $\eta$ | 1.40757 | 0.04601 | 1.40374 | 0.00928 | 1.39891 | 0.00474 | 1.40313 | 0.00285 | 1.40077 | 0.00173 |
| | $\beta$ | 0.94511 | 0.03177 | 0.91003 | 0.00522 | 0.90410 | 0.00245 | 0.90272 | 0.00168 | 0.90087 | 0.00086 |
| RADE | $\eta$ | 1.41745 | 0.04406 | 1.40549 | 0.00879 | 1.39995 | 0.00447 | 1.40354 | 0.00268 | 1.40108 | 0.00164 |
| | $\beta$ | 0.91853 | 0.02054 | 0.90679 | 0.00398 | 0.90159 | 0.00189 | 0.90243 | 0.00128 | 0.90022 | 0.00068 |
| WLSE | $\eta$ | 1.40757 | 0.04601 | 1.40374 | 0.00928 | 1.39891 | 0.00474 | 1.40313 | 0.00285 | 1.40077 | 0.00173 |
| | $\beta$ | 0.94511 | 0.03177 | 0.91003 | 0.00522 | 0.90410 | 0.00245 | 0.90272 | 0.00168 | 0.90087 | 0.00086 |
| LADE | $\eta$ | 1.40569 | 0.04910 | 1.40312 | 0.00999 | 1.39851 | 0.00507 | 1.40313 | 0.00306 | 1.40065 | 0.00185 |
| | $\beta$ | 0.93447 | 0.03783 | 0.90848 | 0.00617 | 0.90352 | 0.00281 | 0.90143 | 0.00198 | 0.90062 | 0.00103 |
| MSALD | $\eta$ | 1.42366 | 0.06634 | 1.40489 | 0.01161 | 1.39990 | 0.00547 | 1.40222 | 0.00327 | 1.40273 | 0.00198 |
| | $\beta$ | 0.92807 | 0.03330 | 0.90659 | 0.00478 | 0.89973 | 0.00207 | 0.90183 | 0.00142 | 0.89930 | 0.00078 |
| ADLTS | $\eta$ | 1.41387 | 0.04120 | 1.40453 | 0.00821 | 1.39966 | 0.00421 | 1.40303 | 0.00250 | 1.40093 | 0.00154 |
| | $\beta$ | 0.90881 | 0.01828 | 0.90426 | 0.00369 | 0.90084 | 0.00174 | 0.90104 | 0.00123 | 0.89985 | 0.00064 |
| PE | $\eta$ | 1.41387 | 0.04120 | 1.40453 | 0.00821 | 1.39966 | 0.00421 | 1.40303 | 0.00250 | 1.40093 | 0.00154 |
| | $\beta$ | 0.90881 | 0.01828 | 0.90426 | 0.00369 | 0.90084 | 0.00174 | 0.90104 | 0.00123 | 0.89985 | 0.00064 |
| MSSL | $\eta$ | 1.39971 | 0.05237 | 1.40219 | 0.00977 | 1.39883 | 0.00489 | 1.40130 | 0.00283 | 1.40200 | 0.00180 |
| | $\beta$ | 0.96783 | 0.03286 | 0.91484 | 0.00451 | 0.90483 | 0.00191 | 0.90382 | 0.00133 | 0.90144 | 0.00074 |

using Eqs (15), (16) and (17), the partial derivatives using $\eta$ and $\beta$ are,

$$\frac{\partial W(x_i)}{\partial \eta} = 2 \sum_{i=1}^{n} \frac{(n+1)^2 (n+2) F_{\eta_i}}{i(n-i+1)} \left[ F_i - \frac{i}{n+1} \right]$$

$$\frac{\partial W(x_i)}{\partial \beta} = 2 \sum_{i=1}^{n} \frac{(n+1)^2 (n+2) F_{\beta_i}}{i(n-i+1)} \left[ F_i - \frac{i}{n+1} \right]$$

by solving the two simultaneous equations $\frac{\partial W(x_i)}{\partial \eta} = 0$ and $\frac{\partial W(x_i)}{\partial \beta} = 0$ we estimate $\eta$ and $\beta$, numerical methods are applied depending on R software.

## 4.8 Methods of left tail Anderson Darling

LADE (see [32]) minimizes its function which is,

$$L(x_i) = -\frac{3}{2} n + 2 \sum_{i=1}^{n} F(x_i) - \frac{1}{n} \sum_{i=1}^{n} (2i - 1) \log F(x_i).$$

**Table 10. Simulation results at $\eta$ = 1.9 and $\beta$ = 1.4.**

| n | | 30 | | 150 | | 300 | | 500 | | 800 | |
|---|---|---|---|---|---|---|---|---|---|---|---|
| Estimate | | Mean | MSE | Mean | MSE | Mean | MSE | Mean | MSE | Mean | MSE |
| MLE | $\eta$ | 1.90784 | 0.03151 | 1.89354 | 0.00617 | 1.89840 | 0.00318 | 1.90029 | 0.00186 | 1.90109 | 0.00114 |
| | $\beta$ | 1.47612 | 0.05280 | 1.41453 | 0.00726 | 1.40670 | 0.00352 | 1.40555 | 0.00205 | 1.40371 | 0.00137 |
| ADE | $\eta$ | 1.91285 | 0.03362 | 1.89380 | 0.00674 | 1.89943 | 0.00340 | 1.90066 | 0.00202 | 1.90163 | 0.00122 |
| | $\beta$ | 1.42774 | 0.05223 | 1.40382 | 0.00820 | 1.40087 | 0.00433 | 1.40143 | 0.00251 | 1.39932 | 0.00165 |
| CME | $\eta$ | 1.90659 | 0.03660 | 1.89231 | 0.00741 | 1.89917 | 0.00371 | 1.90027 | 0.00225 | 1.90167 | 0.00134 |
| | $\beta$ | 1.47560 | 0.08316 | 1.41226 | 0.01124 | 1.40567 | 0.00594 | 1.40434 | 0.00346 | 1.40049 | 0.00220 |
| MPS | $\eta$ | 1.89504 | 0.03163 | 1.89086 | 0.00626 | 1.89708 | 0.00318 | 1.89941 | 0.00186 | 1.90054 | 0.00114 |
| | $\beta$ | 1.61620 | 0.11015 | 1.45070 | 0.01011 | 1.42718 | 0.00437 | 1.41909 | 0.00245 | 1.41259 | 0.00155 |
| LSE | $\eta$ | 1.90659 | 0.03660 | 1.89231 | 0.00741 | 1.89917 | 0.00371 | 1.90027 | 0.00225 | 1.90167 | 0.00134 |
| | $\beta$ | 1.47560 | 0.08316 | 1.41226 | 0.01124 | 1.40567 | 0.00594 | 1.40434 | 0.00346 | 1.40049 | 0.00220 |
| RADE | $\eta$ | 1.91480 | 0.03501 | 1.89402 | 0.00705 | 1.89982 | 0.00355 | 1.90081 | 0.00212 | 1.90184 | 0.00128 |
| | $\beta$ | 1.44158 | 0.06053 | 1.40548 | 0.00955 | 1.40364 | 0.00483 | 1.40238 | 0.00276 | 1.40067 | 0.00182 |
| WLSE | $\eta$ | 1.90659 | 0.03660 | 1.89231 | 0.00741 | 1.89917 | 0.00371 | 1.90027 | 0.00225 | 1.90167 | 0.00134 |
| | $\beta$ | 1.47560 | 0.08316 | 1.41226 | 0.01124 | 1.40567 | 0.00594 | 1.40434 | 0.00346 | 1.40049 | 0.00220 |
| LADE | $\eta$ | 1.90458 | 0.03886 | 1.89159 | 0.00797 | 1.89922 | 0.00398 | 1.90021 | 0.00240 | 1.90163 | 0.00142 |
| | $\beta$ | 1.46172 | 0.08270 | 1.41128 | 0.01167 | 1.40237 | 0.00643 | 1.40321 | 0.00377 | 1.39955 | 0.00243 |
| MSALD | $\eta$ | 1.92361 | 0.05453 | 1.89558 | 0.00857 | 1.90121 | 0.00460 | 1.89994 | 0.00264 | 1.90131 | 0.00155 |
| | $\beta$ | 1.46305 | 0.09024 | 1.40465 | 0.00993 | 1.40114 | 0.00506 | 1.40107 | 0.00303 | 1.39883 | 0.00199 |
| ADLTS | $\eta$ | 1.91285 | 0.03362 | 1.89380 | 0.00674 | 1.89943 | 0.00340 | 1.90066 | 0.00202 | 1.90163 | 0.00122 |
| | $\beta$ | 1.42774 | 0.05223 | 1.40382 | 0.00820 | 1.40087 | 0.00433 | 1.40143 | 0.00251 | 1.39932 | 0.00165 |
| PE | $\eta$ | 1.91285 | 0.03362 | 1.89380 | 0.00674 | 1.89943 | 0.00340 | 1.90066 | 0.00202 | 1.90163 | 0.00122 |
| | $\beta$ | 1.42774 | 0.05223 | 1.40382 | 0.00820 | 1.40087 | 0.00433 | 1.40143 | 0.00251 | 1.39932 | 0.00165 |
| MSSL | $\eta$ | 1.90142 | 0.04337 | 1.89230 | 0.00799 | 1.89796 | 0.00409 | 1.90008 | 0.00238 | 1.90129 | 0.00143 |
| | $\beta$ | 1.51438 | 0.08824 | 1.41745 | 0.01030 | 1.40734 | 0.00495 | 1.40539 | 0.00273 | 1.40167 | 0.00185 |

with help of Eqs (15), (16) and (17), differentiating with respect to the parameters $\eta$ and $\beta$,

$$\frac{\partial L(x_i)}{\partial \eta} = 2\sum_{i=1}^{n} F_{\eta_i} - \frac{1}{n}\sum_{i=1}^{n}(2i-1)\frac{F_{\eta_i}}{F_i}.$$

$$\frac{\partial L(x_i)}{\partial \beta} = 2\sum_{i=1}^{n} F_{\beta_i} - \frac{1}{n}\sum_{i=1}^{n}(2i-1)\frac{F_{\beta_i}}{F_i},$$

by solving the two simultaneous equations $\frac{\partial L(x_i)}{\partial \eta} = 0$ and $\frac{\partial L(x_i)}{\partial \beta} = 0$ we estimate $\eta$ and $\beta$, numerical methods are applied depending on R software.

## 4.9 Minimum spacing absolute-log distance

MSALD minimizes its function

$$\Upsilon(x_i) = \sum_{i=1}^{n+1} |\log \Lambda_i - \log \frac{1}{n+1}|.$$

where $\Lambda_i = F(x_i) - F(x_{i-1})$ Same steps done before to estimate $\eta$ and $\beta$ numerical methods are applied depending on R software.

**Table 11. Total ranks of tables from Tables 4 to 10.**

| Parameters | n | MLE | ADE | CME | MPS | LSE | RADE | WLSE | LADE | MSALD | ADLTS | PE | MSSL |
|---|---|---|---|---|---|---|---|---|---|---|---|---|---|
| $\eta = 0.4$ and $\beta = 0.4$ | 30 | 2 | 7 | 19 | 10 | 19 | 12 | 19 | 24 | 18 | 7 | 7 | 12 |
| | 150 | 2 | 9 | 20 | 8 | 20 | 10 | 20 | 24 | 15 | 9 | 9 | 10 |
| | 300 | 3 | 9 | 20 | 7 | 20 | 10 | 20 | 24 | 15 | 9 | 9 | 10 |
| | 500 | 3 | 12 | 20 | 4 | 20 | 10 | 20 | 24 | 12 | 12 | 12 | 7 |
| | 800 | 3 | 9 | 20 | 4 | 20 | 9 | 20 | 24 | 16 | 9 | 9 | 13 |
| $\eta = 0.4$ and $\beta = 0.9$ | 30 | 3 | 10 | 17 | 3 | 17 | 9 | 17 | 22 | 23 | 10 | 10 | 15 |
| | 150 | 2 | 9 | 17 | 4 | 17 | 9 | 17 | 23 | 23 | 9 | 9 | 17 |
| | 300 | 2 | 7 | 18 | 7 | 18 | 12 | 18 | 24 | 22 | 7 | 7 | 14 |
| | 500 | 2 | 8 | 18 | 4 | 18 | 12 | 18 | 23 | 20 | 8 | 8 | 17 |
| | 800 | 2 | 8 | 19 | 4 | 19 | 12 | 19 | 23 | 20 | 8 | 8 | 14 |
| $\eta = 0.9$ and $\beta = 0.4$ | 30 | 2 | 7 | 20 | 10 | 20 | 11 | 20 | 24 | 14 | 7 | 7 | 14 |
| | 150 | 2 | 8 | 20 | 8 | 20 | 10 | 20 | 24 | 15 | 8 | 8 | 13 |
| | 300 | 2 | 9 | 20 | 5 | 20 | 10 | 20 | 24 | 14 | 9 | 9 | 14 |
| | 500 | 2 | 11 | 20 | 4 | 20 | 11 | 20 | 24 | 12 | 11 | 11 | 10 |
| | 800 | 2 | 10 | 20 | 4 | 20 | 11 | 20 | 24 | 15 | 10 | 10 | 10 |
| $\eta = 0.9$ and $\beta = 0.9$ | 30 | 3 | 7 | 17 | 6 | 17 | 12 | 17 | 21 | 24 | 7 | 7 | 18 |
| | 150 | 2 | 8 | 17 | 4 | 17 | 12 | 17 | 22 | 23 | 8 | 8 | 18 |
| | 300 | 2 | 7 | 18 | 7 | 18 | 12 | 18 | 23 | 23 | 7 | 7 | 14 |
| | 500 | 2 | 8 | 19 | 4 | 19 | 12 | 19 | 23 | 20 | 8 | 8 | 14 |
| | 800 | 2 | 8 | 18 | 4 | 18 | 12 | 18 | 23 | 20 | 8 | 8 | 17 |
| $\eta = 0.9$ and $\beta = 1.4$ | 30 | 6 | 6 | 15 | 13 | 15 | 11 | 15 | 19 | 22 | 6 | 6 | 22 |
| | 150 | 2 | 7 | 18 | 8 | 18 | 11 | 18 | 22 | 20 | 7 | 7 | 18 |
| | 300 | 2 | 7 | 18 | 7 | 18 | 12 | 18 | 23 | 20 | 7 | 7 | 17 |
| | 500 | 2 | 7 | 18 | 7 | 18 | 12 | 18 | 23 | 20 | 7 | 7 | 17 |
| | 800 | 2 | 8 | 18 | 4 | 18 | 12 | 18 | 23 | 20 | 8 | 8 | 17 |
| $\eta = 1.4$ and $\beta = 0.9$ | 30 | 5 | 6 | 15 | 14 | 15 | 11 | 15 | 21 | 22 | 6 | 6 | 20 |
| | 150 | 2 | 7 | 18 | 8 | 18 | 11 | 18 | 23 | 20 | 7 | 7 | 17 |
| | 300 | 2 | 7 | 18 | 7 | 18 | 12 | 18 | 23 | 20 | 7 | 7 | 17 |
| | 500 | 2 | 8 | 19 | 4 | 19 | 12 | 19 | 23 | 20 | 8 | 8 | 14 |
| | 800 | 2 | 8 | 18 | 4 | 18 | 12 | 18 | 23 | 20 | 8 | 8 | 17 |
| $\eta = 1.9$ and $\beta = 1.4$ | 30 | 5 | 6 | 16 | 14 | 16 | 11 | 16 | 16 | 23 | 6 | 6 | 21 |
| | 150 | 2 | 7 | 18 | 9 | 18 | 11 | 18 | 22 | 18 | 7 | 7 | 19 |
| | 300 | 3 | 7 | 18 | 6 | 18 | 12 | 18 | 22 | 20 | 7 | 7 | 18 |
| | 500 | 2 | 8 | 18 | 4 | 18 | 13 | 18 | 23 | 20 | 8 | 8 | 16 |
| | 800 | 3 | 8 | 18 | 3 | 18 | 12 | 18 | 22 | 20 | 8 | 8 | 18 |
| $\sum$ *Ranks* | | 87 | 278 | 640 | 223 | 640 | 393 | 640 | 795 | 669 | 278 | 278 | 539 |
| Overall Ranks | | 1 | 4 | 9 | 2 | 9 | 6 | 9 | 12 | 11 | 4 | 4 | 7 |

**Table 12. Some descriptive analysis of all data sets.**

| | n | Mean | Median | Variance | Skewness | Kurtosis | Range | Minimum | Maximum |
|---|---|---|---|---|---|---|---|---|---|
| data1 | 50 | 0.1632 | 0.1600 | 0.0066 | 0.0746 | -0.7374 | 0.3000 | 0.0200 | 0.3200 |
| data2 | 15 | 0.0402 | 0.0510 | 0.0008 | 0.1207 | -1.4363 | 0.0788 | 0.0041 | 0.0829 |
| data3 | 15 | 0.0598 | 0.0490 | 0.0008 | -0.1207 | -1.4363 | 0.0788 | 0.0171 | 0.0959 |

**Table 13. MLEs and SEs for the first data set.**

| Distributions | $\hat{\eta}$ | $\hat{\beta}$ | $\hat{\theta}$ | $SE(\hat{\eta})$ | $SE(\hat{\beta})$ | $SE(\hat{\theta})$ |
|---|---|---|---|---|---|---|
| MKTL | 0.6552 | 1.9321 | | 0.0356 | 0.2074 | |
| ITL | 44.4100 | | | 6.2805 | | |
| TL | 0.7248 | | | 0.1025 | | |
| Kw | 2.0775 | 33.1423 | | 0.2549 | 13.9256 | |
| TPL | 39.1200 | 2.1546 | | 15.0128 | 0.2403 | |
| B | 2.6829 | 13.8691 | | 0.5072 | 2.8287 | |
| TW | 36.2662 | 2.1195 | | 14.4750 | 0.2463 | |
| TITL | 44.4213 | | | 6.2836 | | |
| UEP | 1.7694 | 0.2203 | 0.9172 | 0.2018 | 2.0379 | 15.0132 |

**Table 14. Measures of fitting for the first data set.**

| Distributions | Lnl | AIC | BIC | CAIC | HQIC | KS | W | A |
|---|---|---|---|---|---|---|---|---|
| | | | | | | (Pv_KS) | (Pv_W) | (Pv_A) |
| MKTL | -114.625 | -110.625 | -106.801 | -110.37 | -109.169 | 0.0906 | 0.0733 | 0.4458 |
| | | | | | | (0.8066) | (0.7334) | (0.8017) |
| ITL | -107.632 | -105.632 | -103.72 | -105.549 | -104.904 | 0.1708 | 0.2346 | 1.1931 |
| | | | | | | (0.1082) | (0.2096) | (0.27) |
| TL | -56.8156 | -54.8156 | -52.9036 | -54.7323 | -54.0875 | 0.3623 | 1.6469 | 8.2635 |
| | | | | | | (>0.001) | (0.0001) | (0.0001) |
| Kw | -112.137 | -108.137 | -104.313 | -107.882 | -106.681 | 0.1103 | 0.1042 | 0.6776 |
| | | | | | | (0.5777) | (0.5663) | (0.5771) |
| TPL | -111.471 | -107.471 | -103.647 | -107.216 | -106.015 | 0.1101 | 0.1088 | 0.7357 |
| | | | | | | (0.5789) | (0.5453) | (0.5290) |
| B | -109.213 | -105.213 | -101.389 | -104.958 | -103.757 | 0.1415 | 0.1540 | 0.9125 |
| | | | | | | (0.2692) | (0.3785) | (0.4061) |
| TW | -111.784 | -107.784 | -103.96 | -107.528 | -106.327 | 0.1099 | 0.1066 | 0.7085 |
| | | | | | | (0.5813) | (0.5552) | (0.5510) |
| TITL | -107.633 | -105.633 | -103.721 | -105.549 | -104.905 | 0.1709 | 0.2352 | 1.1954 |
| | | | | | | (0.1079) | (0.2087) | (0.2691) |
| UEP | -77.2764 | -71.2764 | -65.5403 | -70.7546 | -69.092 | 0.0983 | 0.0838 | 0.5336 |
| | | | | | | (0.7194) | (0.6722) | (0.7120) |

**Table 15. MLEs and SEs for the second data set.**

| Distributions | $\hat{\eta}$ | $\hat{\beta}$ | $\hat{\theta}$ | $SE(\hat{\eta})$ | $SE(\hat{\beta})$ | $SE(\hat{\theta})$ |
|---|---|---|---|---|---|---|
| MKTL | 0.2890 | 2.7477 | | 0.0204 | 0.5692 | |
| ITL | 485.6900 | | | 125.4058 | | |
| TL | 0.3460 | | | 0.0893 | | |
| Kw | 1.4011 | 78.9277 | | 0.3138 | 74.3173 | |
| TPL | 86.6196 | 1.4266 | | 77.6232 | 0.3033 | |
| B | 1.5370 | 36.8077 | | 0.5119 | 14.3484 | |
| TW | 82.9288 | 1.4147 | | 76.0338 | 0.3080 | |
| TITL | 485.6900 | | | 125.4058 | | |
| UEP | 1.3668 | 0.0079 | 0.0887 | 0.2984 | 0.0054 | 0.0973 |

**Table 16. Measures of fitting for the second data set.**

| Distributions | Lnl | AIC | BIC | CAIC | HQIC | KS | W | A |
|---|---|---|---|---|---|---|---|---|
| | | | | | | (Pv_KS) | (Pv_W) | (Pv_A) |
| MKTL | -69.6850 | -65.6850 | -64.2689 | -64.6850 | -65.7001 | 0.2184 | 0.1056 | 0.5815 |
| | | | | | | (0.4123) | (0.5641) | (0.6627) |
| ITL | -66.0095 | -64.0095 | -63.3014 | -63.7018 | -64.0170 | 0.2274 | 0.2086 | 1.6328 |
| | | | | | | (0.3638) | (0.2527) | (0.1482) |
| TL | -44.4172 | -42.4172 | -41.7092 | -42.1096 | -42.4248 | 0.4708 | 0.6922 | 3.3673 |
| | | | | | | (0.0014) | (0.0121) | (0.0184) |
| Kw | -68.6785 | -64.6785 | -63.2624 | -63.6785 | -64.6936 | 0.2409 | 0.1185 | 0.6674 |
| | | | | | | (0.2983) | (0.5067) | (0.5839) |
| TPL | -68.5582 | -64.5582 | -63.1421 | -63.5582 | -64.5733 | 0.2417 | 0.1211 | 0.6869 |
| | | | | | | (0.2946) | (0.4959) | (0.5672) |
| B | -68.2194 | -64.2194 | -62.8033 | -63.2194 | -64.2345 | 0.2481 | 0.1161 | 0.6480 |
| | | | | | | (0.2668) | (0.5168) | (0.6010) |
| TW | -68.6161 | -64.6161 | -63.2000 | -63.6161 | -64.6312 | 0.2413 | 0.1199 | 0.6777 |
| | | | | | | (0.2964) | (0.5009) | (0.5750) |
| TITL | -66.0095 | -64.0095 | -63.3014 | -63.7018 | -64.0170 | 0.2274 | 0.2086 | 1.6328 |
| | | | | | | (0.3638) | (0.2527) | (0.1482) |
| UEP | -66.4233 | -60.4233 | -58.2991 | -58.2414 | -60.4459 | 0.2376 | 0.1167 | 0.6564 |
| | | | | | | (0.3138) | (0.5142) | (0.5936) |

## 4.10 Anderson Darling left tail second order

ADLTS (see [33]) minimizes its function,

$$LTS(x_i) = 2\sum_{i=1}^{n} \log F(x_i) + \frac{1}{n}\sum_{i=1}^{n} \frac{(2i-1)}{F(x_i)}.$$

by differentiating using the parameters $\eta$ and $\beta$,

$$\frac{\partial LTS(x_i)}{\partial \eta} = 2\sum_{i=1}^{n} \frac{F_{\eta_i}}{F_i} - \frac{1}{n}\sum_{i=1}^{n} \frac{(2i-1)F_{\eta_i}}{F_i^2}$$

**Table 17. MLEs and SEs for the third data set.**

| Distributions | $\hat{\eta}$ | $\hat{\beta}$ | $\hat{\theta}$ | $SE(\hat{\eta})$ | $SE(\hat{\beta})$ | $SE(\hat{\theta})$ |
|---|---|---|---|---|---|---|
| MKTL | 0.3441 | 3.9232 | | 0.0171 | 0.8173 | |
| ITL | 270.6017 | | | 69.8693 | | |
| TL | 0.4377 | | | 0.1130 | | |
| Kw | 2.4407 | 719.2125 | | 0.5278 | 982.3061 | |
| TPL | 732.8242 | 2.4472 | | 991.8768 | 0.5240 | |
| B | 3.8234 | 60.2533 | | 1.3398 | 22.4663 | |
| TW | 725.8930 | 2.4439 | | 988.8204 | 0.5269 | |
| TITL | 270.8100 | | | 69.9230 | | |
| UEP | 2.3069 | 0.0998 | 2.0681 | 0.4956 | 0.5065 | 24.2334 |

**Table 18. Measures of fitting for the third data set.**

| Distributions | Lnl | AIC | BIC | CAIC | HQIC | KS | W | A |
| --- | --- | --- | --- | --- | --- | --- | --- | --- |
| | | | | | | (Pv_KS) | (Pv_W) | (Pv_A) |
| MKTL | -67.3644 | -63.3644 | -61.9483 | -62.3644 | -63.3795 | 0.1869 | 0.0958 | 0.6235 |
| | | | | | | (0.6065) | (0.6121) | (0.6232) |
| ITL | -64.3267 | -62.3267 | -61.6187 | -62.0190 | -62.3343 | 0.2057 | 0.1070 | 0.6380 |
| | | | | | | (0.4867) | (0.5572) | (0.6100) |
| TL | -32.6896 | -30.6896 | -29.9816 | -30.3820 | -30.6972 | 0.5249 | 0.8986 | 4.2493 |
| | | | | | | (0.0002) | (0.0036) | (0.0068) |
| Kw | -66.7592 | -62.7592 | -61.3431 | -61.7592 | -62.7743 | 0.2002 | 0.1073 | 0.6497 |
| | | | | | | (0.5206) | (0.5558) | (0.5995) |
| TPL | -66.7467 | -62.7467 | -61.3306 | -61.7467 | -62.7618 | 0.2003 | 0.1076 | 0.6522 |
| | | | | | | (0.5199) | (0.5543) | (0.5973) |
| B | -65.6052 | -61.6052 | -60.1891 | -60.6052 | -61.6203 | 0.2099 | 0.1008 | 0.6471 |
| | | | | | | (0.4615) | (0.5872) | (0.6018) |
| TW | -66.7529 | -62.7529 | -61.3368 | -61.7529 | -62.768 | 0.2003 | 0.1075 | 0.6509 |
| | | | | | | (0.5201) | (0.5551) | (0.5985) |
| TITL | -64.3267 | -62.3267 | -61.6187 | -62.0190 | -62.3343 | 0.2060 | 0.1062 | 0.6339 |
| | | | | | | (0.4850) | (0.5613) | (0.6138) |
| UEP | -63.2437 | -57.2437 | -55.1196 | -55.0619 | -57.2664 | 0.1972 | 0.1071 | 0.6415 |
| | | | | | | (0.5400) | (0.5566) | (0.6069) |

$$\frac{\partial LTS(x_i)}{\partial \beta} = 2\sum_{i=1}^{n} \frac{F_{\beta_i}}{F_i} - \frac{1}{n}\sum_{i=1}^{n} \frac{(2i-1)F_{\beta_i}}{F_i^2},$$

by solving the two simultaneous equations $\frac{\partial LTS(x_i)}{\partial \eta} = 0$ and $\frac{\partial LTS(x_i)}{\partial \beta}$ we estimate $\eta$ and $\beta$, numerical methods are applied depending on R software.

## 4.11 Percentile estimation

The percentile estimation (PE) provided by [34, 35] to utilize on Weibull distribution and subsequently employed for alternative distributions, its function

$$PE(x_i) = \sum_{i=1}^{n} \left[ x_i - Q(\eta, \beta) \right]^2$$

to estimate $\eta$, and $\beta$ we need to minimize $PE(x_i)$ and repeating the procedures carried out on previous estimation methods.

## 4.12 Minimum spacing square-log distance

MSSL minimizes its function which is,

$$\delta_{(x_i)} = \sum_{i=1}^{n+1} \left( \log\left(\Lambda_i(x_i)\right) - \log\frac{1}{n+1} \right)^2,$$

where $\Lambda_i = F(x_i) - F(x_{i-1})$ repeating the procedures carried out on previous estimation methods.

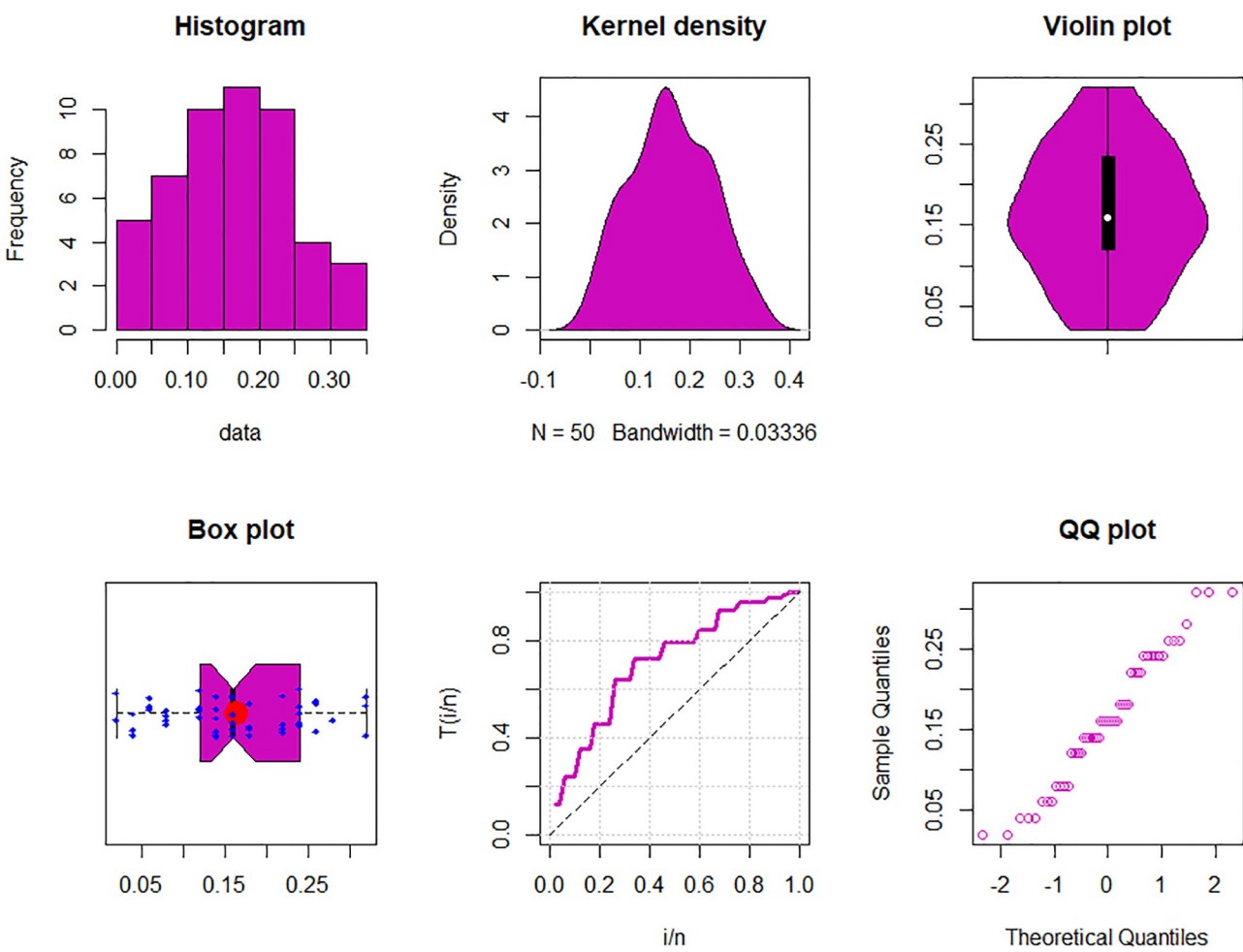

**Fig 23. Some basic non-parametric plots for the first dataset.**

## 5 Simulation

In simulation scenarios, numerical techniques are employed to compute estimates for $\eta$ and $\beta$, utilizing the R software to assess the average and mean square error (MSE) of these parameters. The estimation methods in Section 6 are employed for this purpose. To accomplish this, random samples of different sizes (n = 30, 150, 300, 500, 800) are drawn from the *MKTL* distribution, with each size replicated 1000 times. The initial values for $\eta$ and $\beta$ are set as shown in Table 3:

The MSE plots in Figs 11–22 associated with the twelve estimation approaches outlined in Tables 4–10 are given to illustrate their performance characteristics. Moreover, Table 11 displays the summation and overall ranks for MSE values across all tables, enabling a comparative analysis of the different estimation methodologies. Upon examination of the tables, graphs, and ranks, the following observations emerge:

- The MSE demonstrates a declining pattern with the increase in the variable *n* across all estimation techniques.

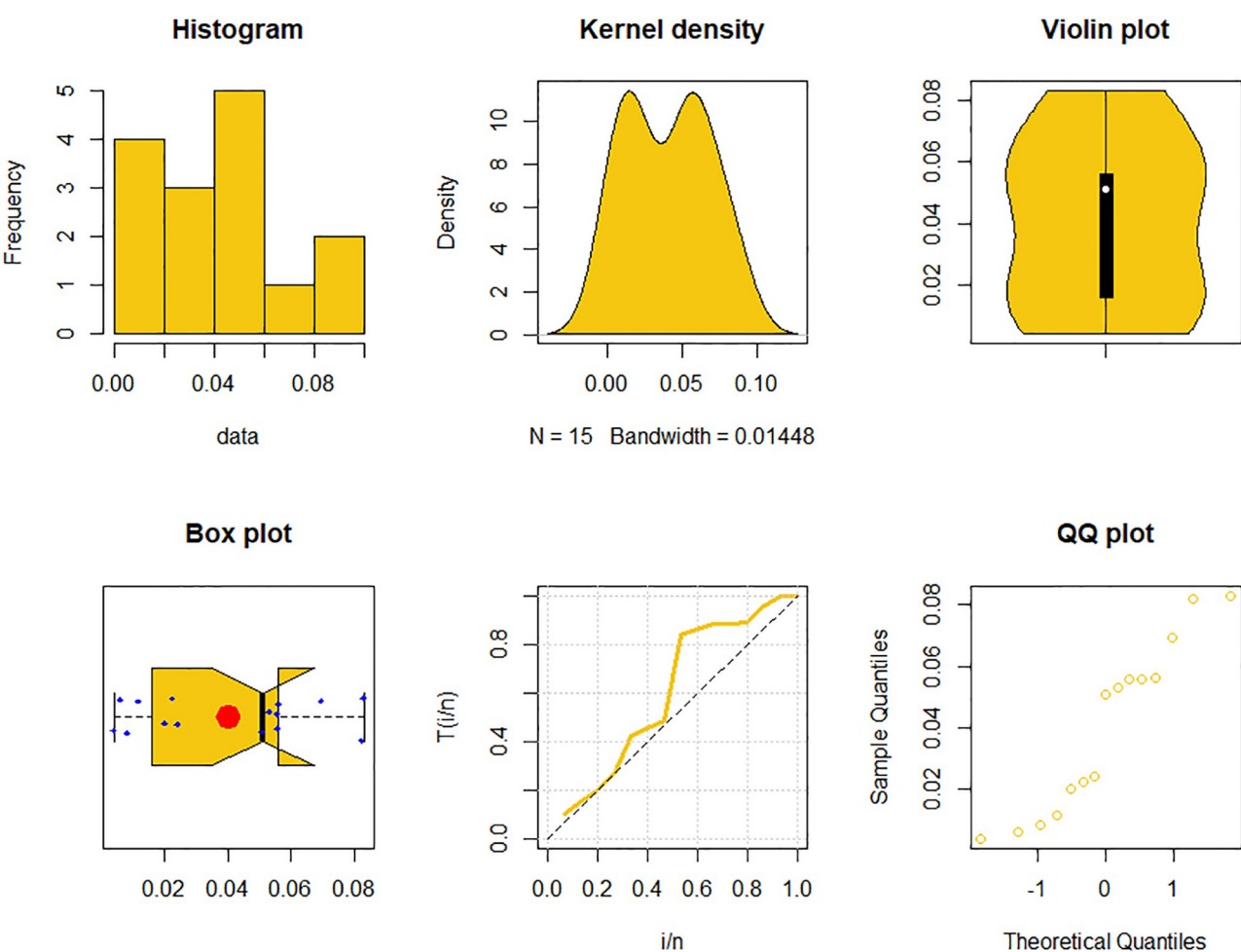

**Fig 24. Some basic non-parametric plots for the second dataset.**

- Figs 11–22 corroborate the MSE trends observed in Table 4, a pattern consistent across all tables.

- As *n* grows, the mean estimations for the parameters $(\eta, \beta)$ tends to converge towards their initial parameter values.

- The overall ranks displayed in Table 11 indicate that Maximum Likelihood Estimation (MLE) emerges as the superior method for parameter estimation.

## 6 Data analysis

The significance and promise of the MKTL distribution are demonstrated in this section through the use of three authentic data sets.

The first set of data shows the Burr measurements (in millimetres) for 50 Burr observations on iron sheets. The sheet thickness is 3.15 mm, and the hole diameter is 12 mm. was utilised by [36]. The provided data set consists of the following values: 0.32, 0.16, 0.22, 0.14, 0.16, 0.08, 0.14, 0.12, 0.06, 0.12, 0.08, 0.16, 0.04, 0.22, 0.12, 0.02, 0.06, 0.18, 0.16, 0.28, 0.04, 0.22, 0.26, 0.08,

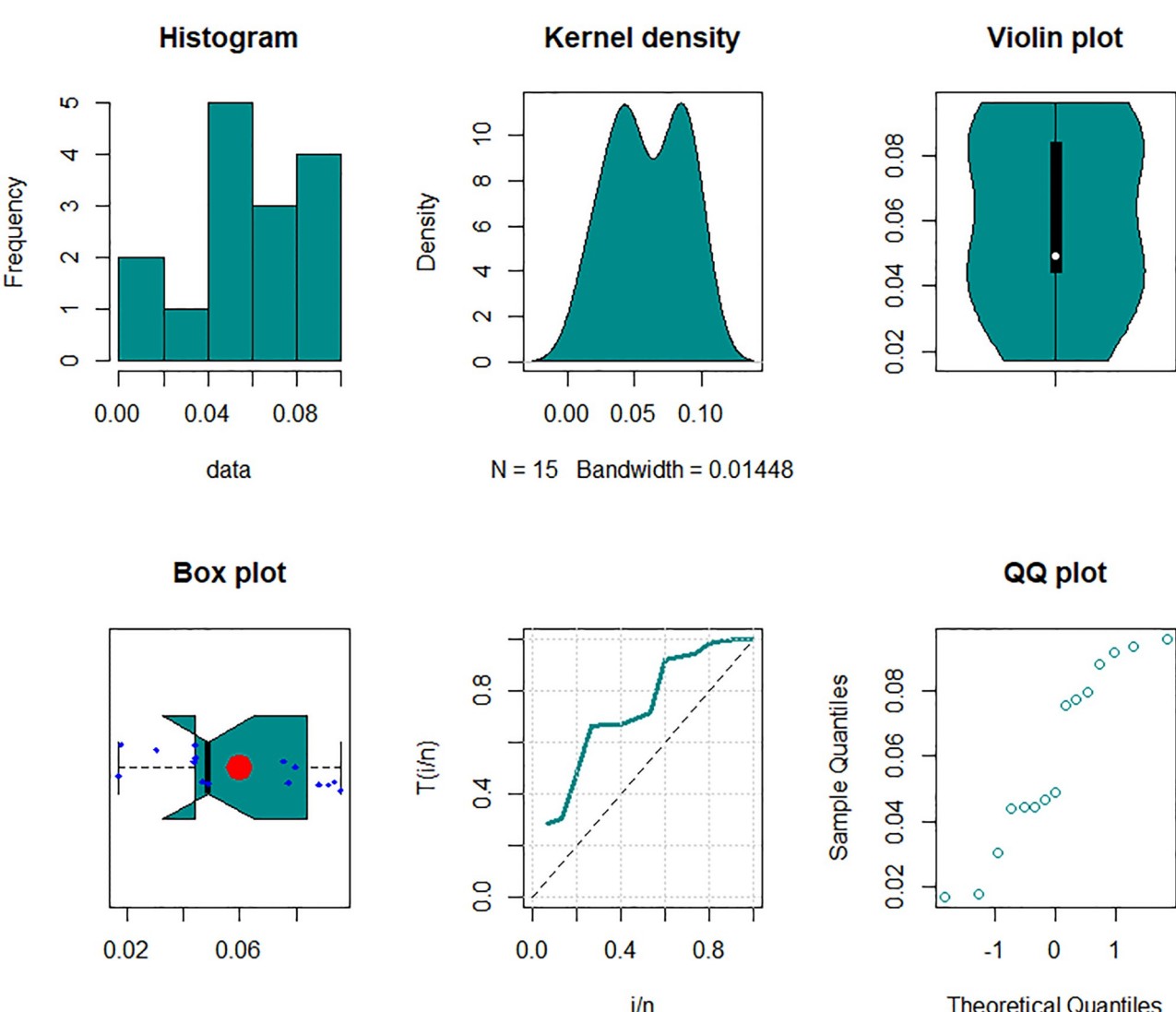

**Fig 25. Some basic non-parametric plots for the third dataset.**

0.14, 0.18, 0.08, 0.26, 0.18, 0.32, 0.24, 0.24, 0.24, 0.16, 0.16, 0.02, 0.18, 0.24, 0.14, 0.04, 0.14, 0.26, 0.14, 0.16, 0.32, 0.24, 0.06, 0.12, 0.22, 0.24.

The second set of data shows the permanent wilting point (PWP) observations from the first 100 daily soil moisture data points, represented by 15 data points of soil moisture deficit. is studied by [37], data are: 0.0468, 0.0774, 0.0443, 0.0938, 0.0882, 0.0171, 0.0917, 0.0305, 0.0798, 0.0757, 0.0444, 0.0959, 0.0179, 0.0439, 0.049.

The third data set presents 15 data points, which is examined by [37]. The following data relates to soil moisture, which is the reason for PWP: 0.0561, 0.0243, 0.0695, 0.0062, 0.0821, 0.0557, 0.0226, 0.0556, 0.0083, 0.0829, 0.0118, 0.051, 0.0041, 0.0202, 0.0532.

The descriptive analysis of all the data sets is reported in Table 12.

The MKTL distribution's goodness of fit is evaluated using these actual data sets. The suggested model is compared with inverse Topp Leone (ITL) [11], Kumaraswamy (Kw) [38],

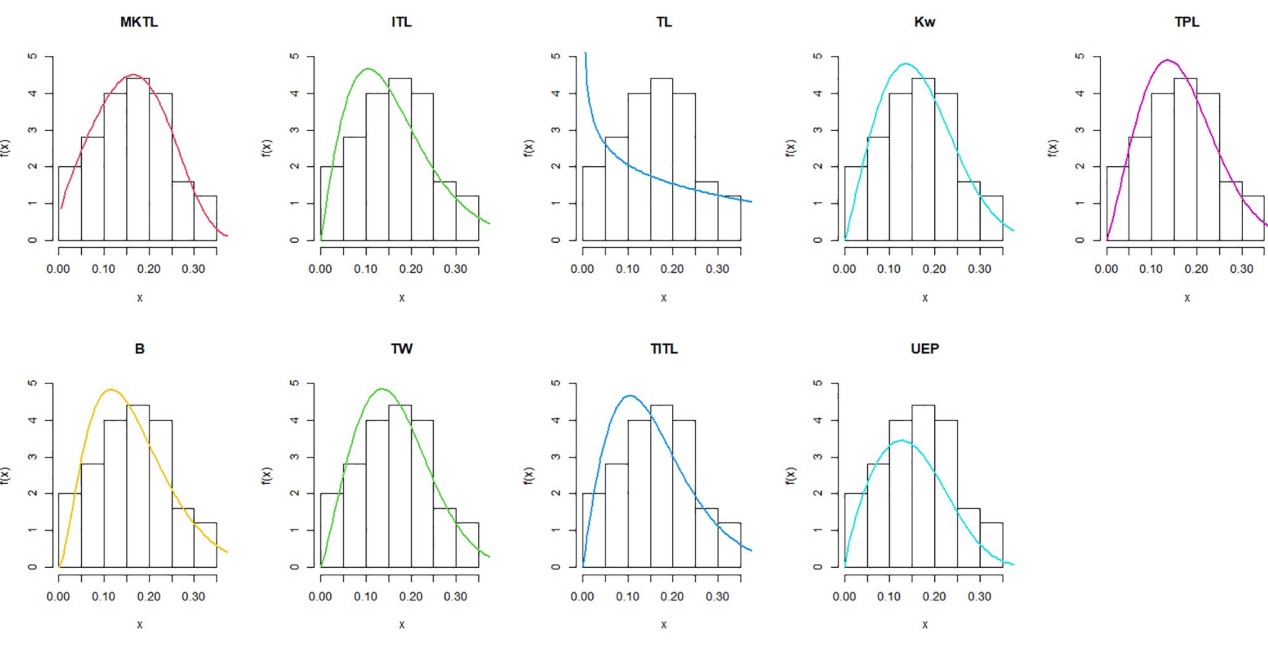

**Fig 26. Estimated pdf plots of data set 1.**

truncated power Lomax (TPL) [39], beta (B) [40], truncated Weibull (TW) [41], truncated inverse Topp Leone (TITL) [42], unit exponential Pareto (UEP) [43], and TL models.

The maximum likelihood estimators (MLEs) and standard errors (SEs) of the model parameters are computed. In order to assess the distribution models, various criteria are taken into account, including the Akaike information criterion (*AIC*), Bayesian information criterion (*BIC*), correct AIC (*CAIC*), Hannan-Quinn IC (*HQIC*), Kolmogorov-Smirnov (*KS*) test,

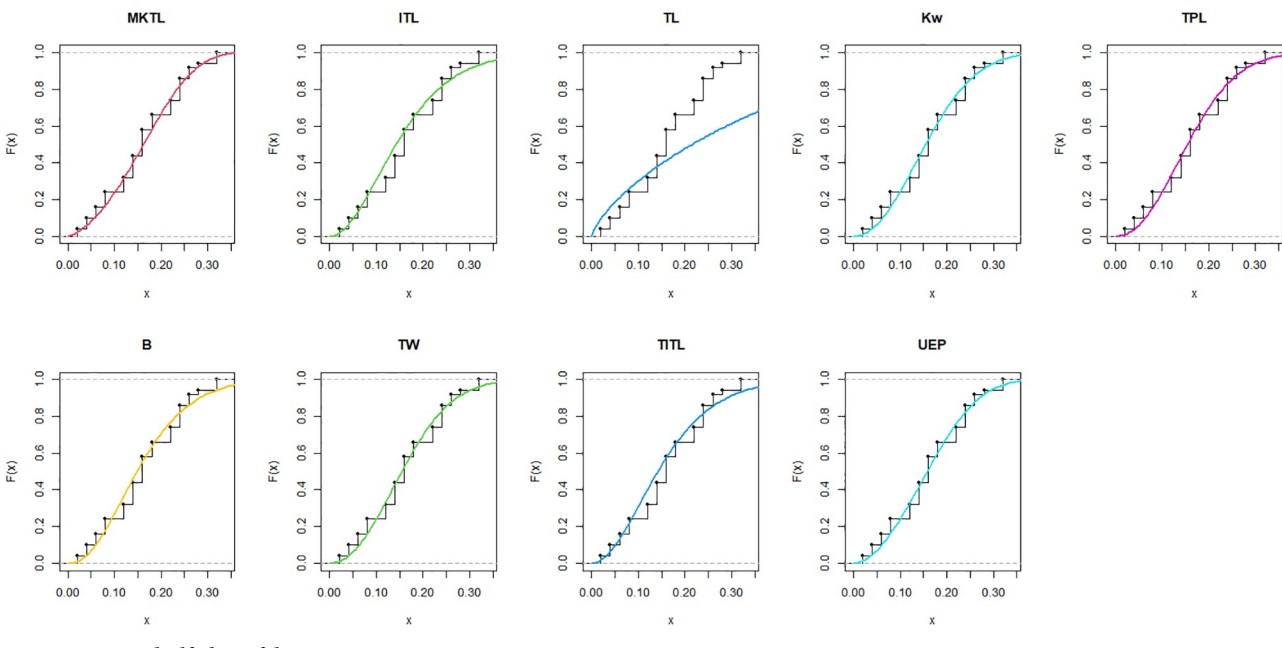

**Fig 27. Estimated cdf plots of data set 1.**

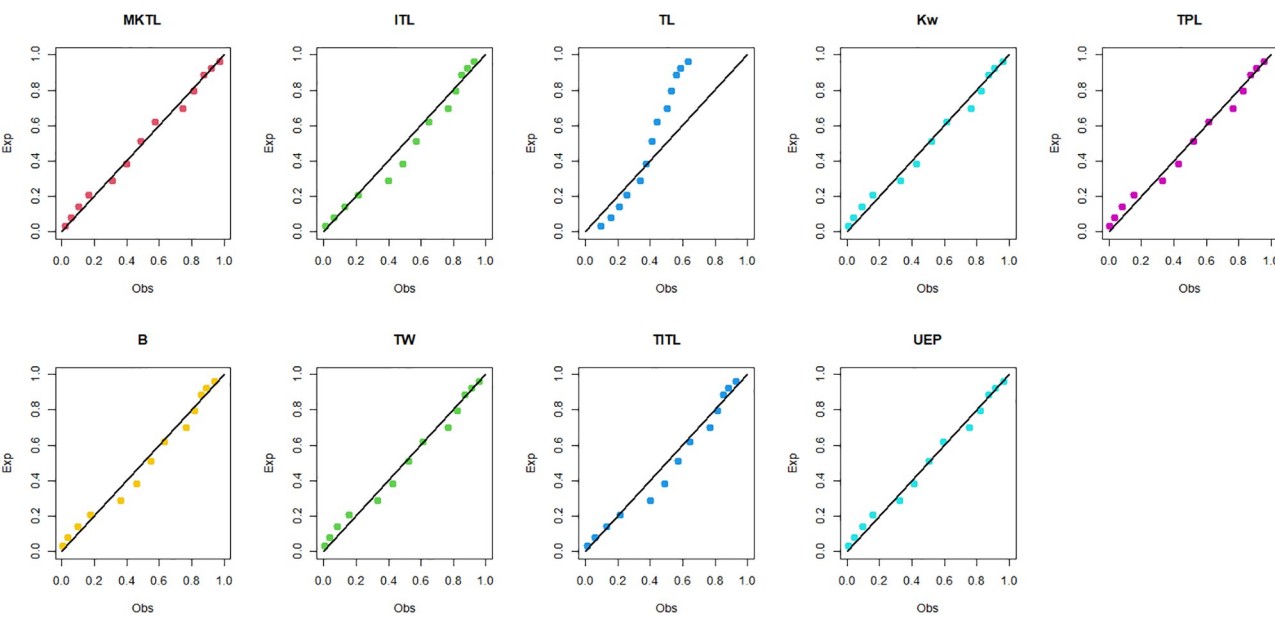

**Fig 28. Estimated PP plots of data set 1.**

p-value of *KS* (*PV_KS*), the Cramér-Von-Mises test (*W*), p-value of *W* (*PV_W*), the Anderson-Darling test (*A*) and p-value of *A* (*PV_A*). test. In contrast, the broader dissemination is associated with reduced values of *AIC*, *CAIC*, *BIC*, *HQIC*, *KS*, *W*, *A* and the highest magnitude of *PV_KS*, *PV_W* and *PV_A*. The MLEs of the competing models, along with their SEs and the values of *AIC*, *CAIC*, *BIC*, *HQIC*, *PV_KS*,*W*, *PV_W*, *A*, *PV_A* and *KS* for the proposed datasets, are shown in Tables 13–18. It was found that the MORW distribution, characterized by

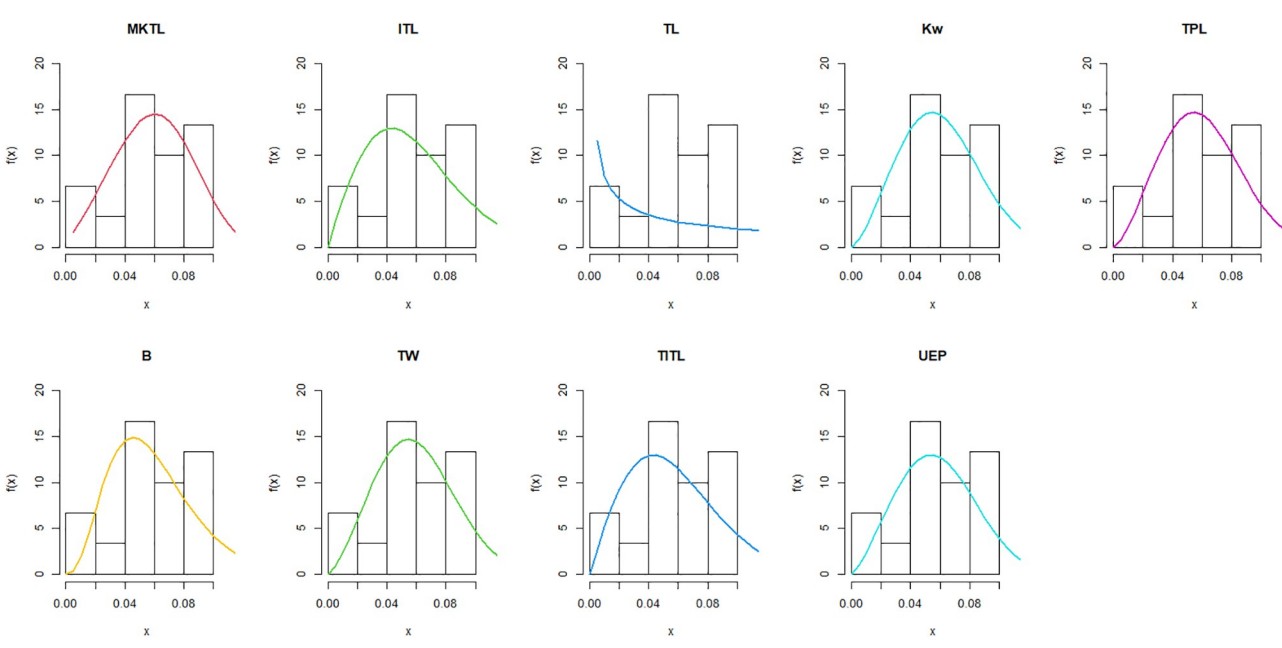

**Fig 29. Estimated pdf plots of data set 2.**

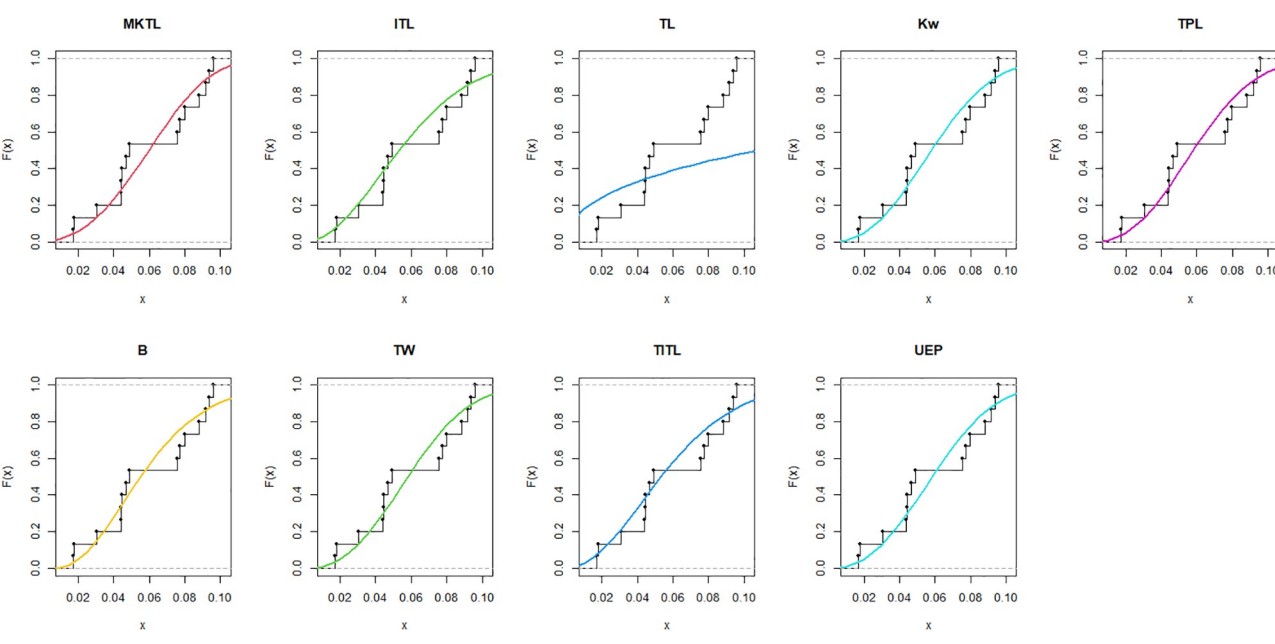

**Fig 30. Estimated cdf plots of data set 2.**

three parameters, has a better goodness of fit compared to alternative models. This distribution has the lowest values for *AIC*, *CAIC*, *BIC*, *HQIC*, *KS*, *W* and *A* and the highest value for *PV_KS*, *PV_W* and *PV_A* among the distributions considered in this analysis. among all fitted models. Figs 23–25 illustrates the original PDF shape using the non-parametric kernel density estimate method, and we can see from Figs 23–25 that the PDF has an asymmetrical shape.

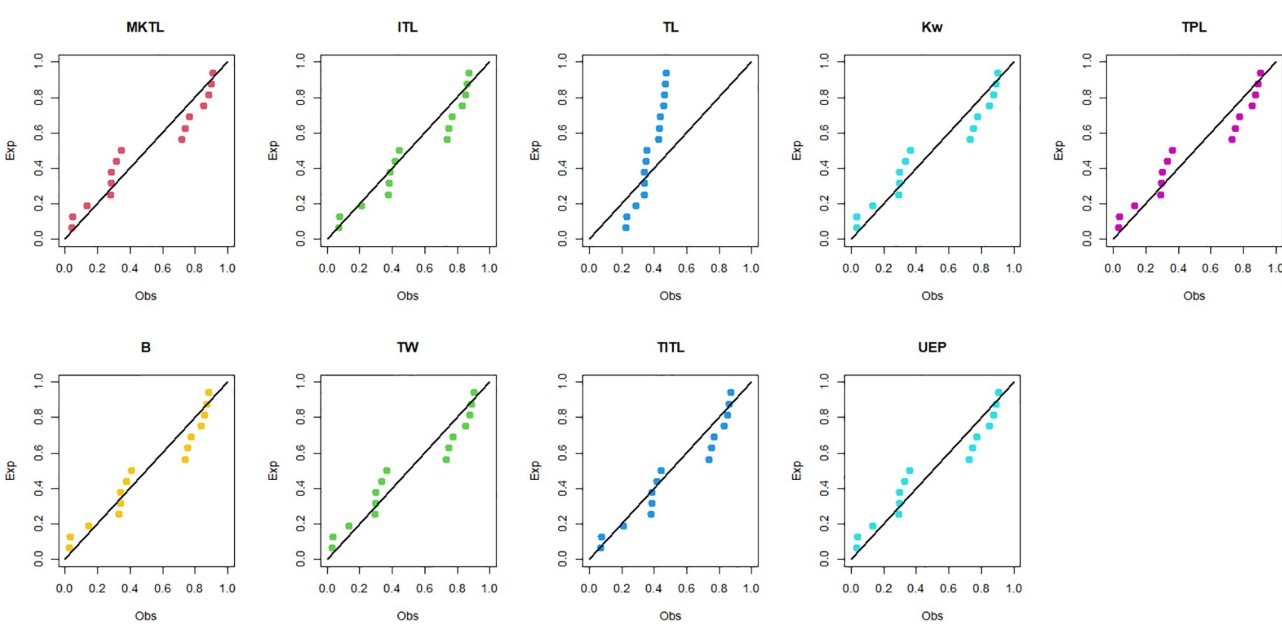

**Fig 31. Estimated PP plots of data set 2.**

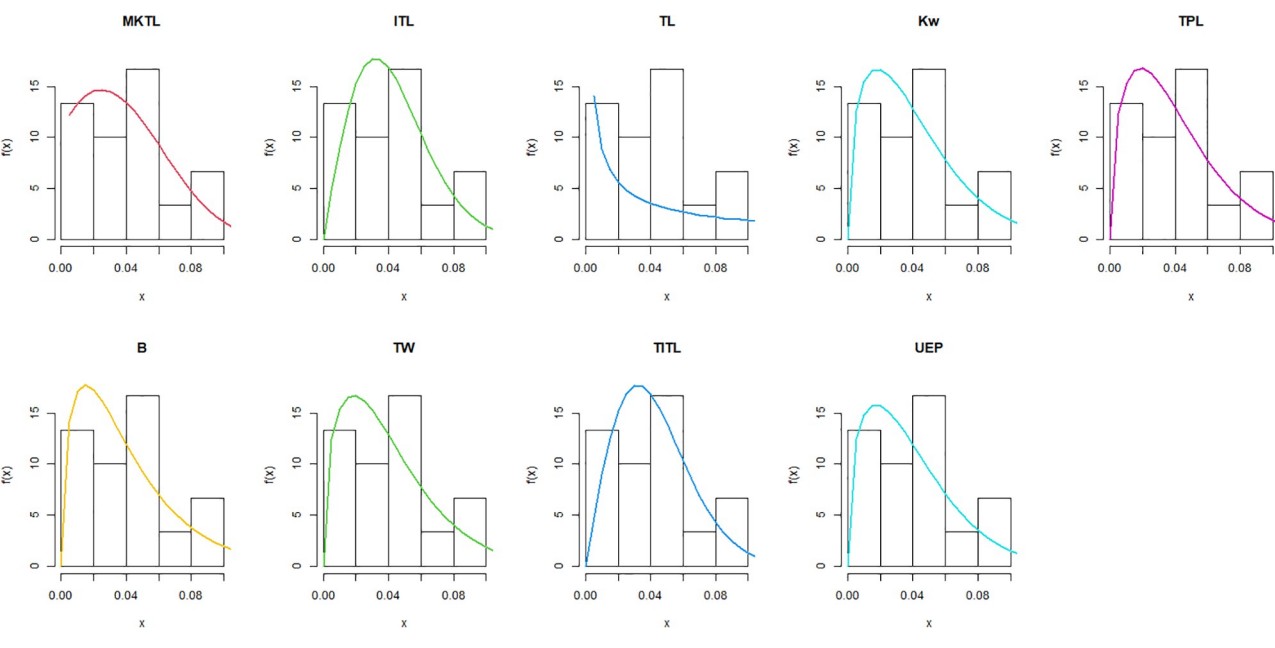

**Fig 32. Estimated pdf plots of data set 3.**

Moreover, the quantile-quantile (QQ) plot in the same figure is used to verify the normalcy criterion. The box plot can also be used to identify outliers. As a result, we may conclude that the first dataset contains outliers (Figs 23–25 shows the data as blue dots, while the red circle indicates the median). In addition, Figs 26–34 show the graphical representations of the estimated pdf, cdf and probability plots (PP) for the competing model applied to the given data sets.

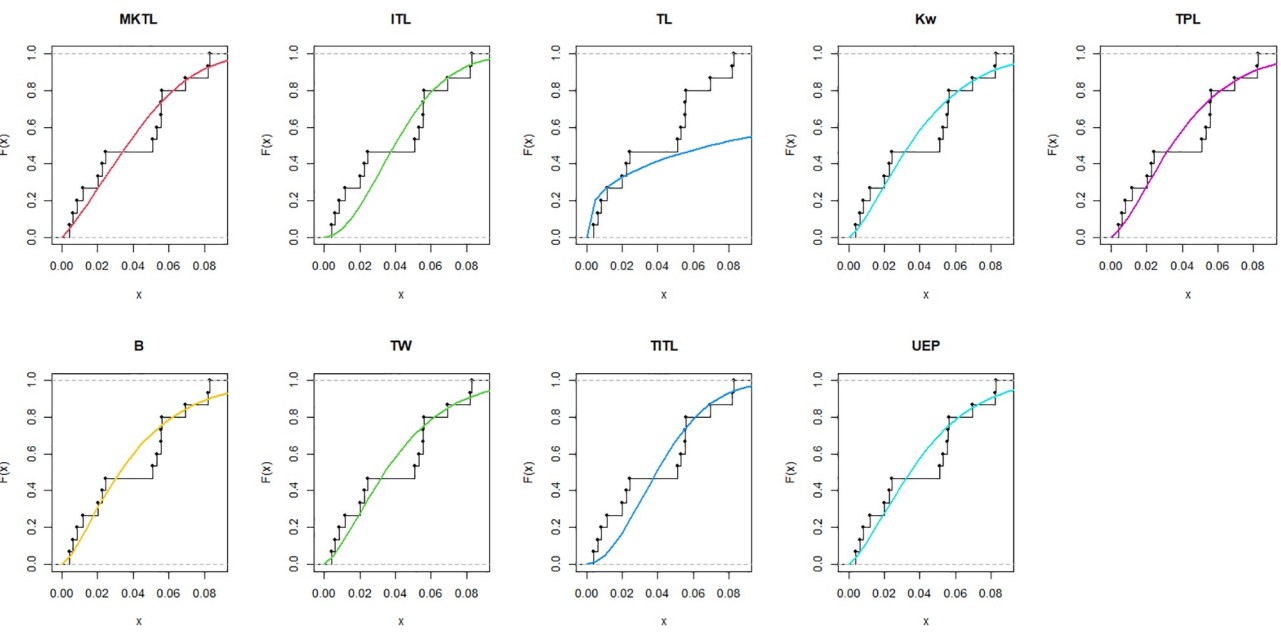

**Fig 33. Estimated cdf plots of data set 3.**

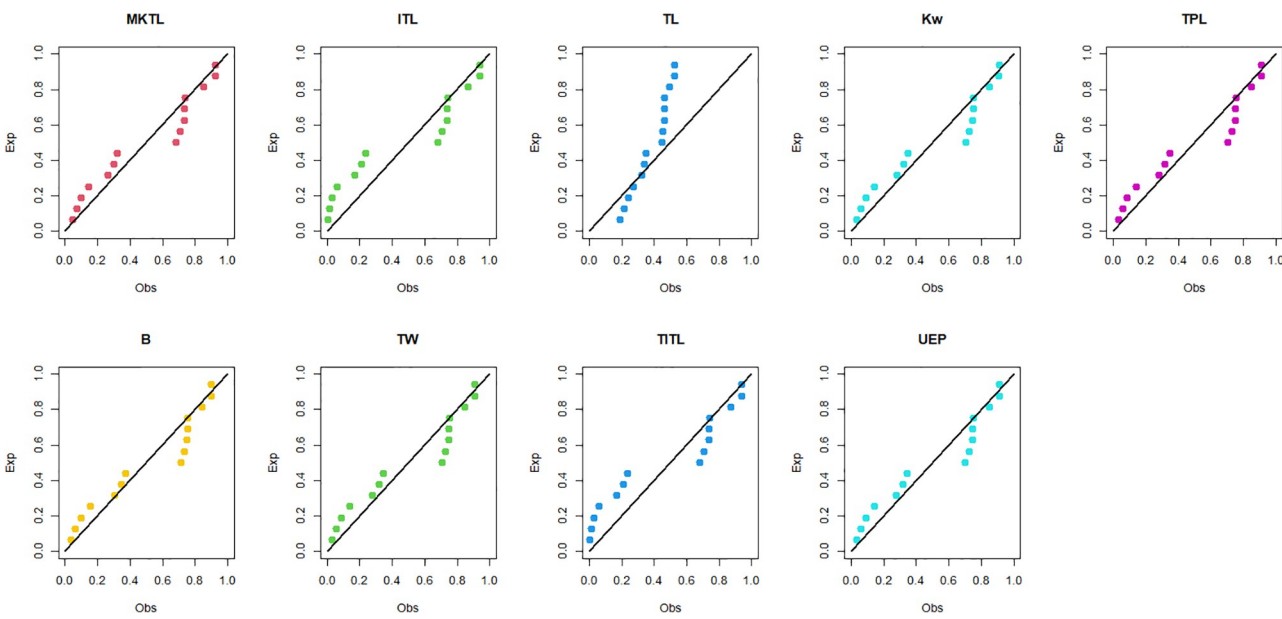

**Fig 34. Estimated PP plots of data set 3.**

## 7 Modified Kies Topp-Leone quantile regression

In modeling the relationship between an endogenous variable and a set of exogenous variables, the identification of an appropriate regression model is paramount for accurate statistical inference. Quantile regression model (QRM) is known to be robust when it comes to modeling such relationship, especially in situations when the response variable contains atypical points (outliers). In this section, we propose a new QRM based on the re-parameterization of the density function of the MKTL distribution in terms of its quantile function (qf). For more details on how to formulate QRM see [44, 45]. Given an endogenous variable $Y$ that has an MKTL distribution and $\tau \in (0, 1)$ is a quantile parameter, we formulate the re-parameterize density by first making $\eta$ the subject from the qf of the MKTL distribution. Thus,

$\eta = -\log\left(1 + (\log(1/(1-q)))^{\frac{-1}{\beta}}\right)/\log(1-(1-\tau)^2), q \in (0,1)$. The re-parameterize density function in terms of the quantile is therefore given by

$$f(y; \beta, \tau) = \frac{2\beta r(q) y^{r(q)-1}(1-y)(2-y)^{r(q)-1}(1-(1-y)^2)^{r(q)(\beta-1)} e^{-((1-(1-y)^2)^{-r(q)}-1)^{-\beta}}}{(1-(1-(1-y)^2)^{r(q)})^{\beta+1}}, \quad (18)$$

where $r(q) = -\log\left(1 + (\log(1/(1-q)))^{\frac{-1}{\beta}}\right)/\log(1-(1-\tau)^2)$. When $q = 0.10, 0.25, 0.50,$ 0.75, 0.90, 0.95 and 0.99, the density function of the $10^{th}, 25^{th}, 50^{th}, 75^{th}, 90^{th}, 95^{th}$ and $99^{th}$ percentiles are obtained respectively.

The MKTL QRM is attained by adopting a monotonically increasing and twice differentiable link function to define the relationship between the exogenous variables and the conditional quantiles. Thus, we have

$$h(\tau_i) = \mathbf{x}'_i \boldsymbol{\alpha},$$

where $h(\cdot)$ is the link function, $\tau_i$ is the $i^{th}$ quantile parameter, $\boldsymbol{\alpha} = (\alpha_0, \tau_1, \ldots, \alpha_k)'$ is the vector of parameters to be estimated and $\mathbf{x}'_i = (1, x_{i1}, x_{i2}, \ldots, x_{ik})$ are the unknown $i^{th}$ vector of

**Table 19. MKTL QRM simulation results for scenario I.**

| Parameter | n | AE | AB | RMSE | CP | LCL | UCL | AWCI |
|---|---|---|---|---|---|---|---|---|
| $\alpha_0 = 0.4$ | 50 | 0.4291 | 0.2260 | 0.2912 | 0.9330 | -0.1260 | 0.9841 | 1.1102 |
| | 100 | 0.4158 | 0.1655 | 0.2083 | 0.9340 | 0.0319 | 0.7996 | 0.7677 |
| | 250 | 0.4162 | 0.1011 | 0.1285 | 0.9310 | 0.1783 | 0.6541 | 0.4758 |
| | 450 | 0.4038 | 0.0718 | 0.0896 | 0.9590 | 0.2271 | 0.5804 | 0.3533 |
| | 850 | 0.4012 | 0.0530 | 0.0670 | 0.9400 | 0.2733 | 0.5291 | 0.2558 |
| | 1000 | 0.4007 | 0.0467 | 0.0582 | 0.9610 | 0.2825 | 0.5189 | 0.2363 |
| $\alpha_1 = -0.6$ | 50 | -0.5920 | 0.3607 | 0.4642 | 0.9400 | -1.4417 | 0.2576 | 1.6993 |
| | 100 | -0.6040 | 0.2432 | 0.3075 | 0.9420 | -1.1819 | -0.0260 | 1.1559 |
| | 250 | -0.6163 | 0.1534 | 0.1927 | 0.9440 | -0.9731 | -0.2594 | 0.7136 |
| | 450 | -0.5953 | 0.1074 | 0.1326 | 0.9600 | -0.8586 | -0.3320 | 0.5266 |
| | 850 | -0.5957 | 0.0773 | 0.0969 | 0.9570 | -0.7863 | -0.4051 | 0.3813 |
| | 1000 | -0.5987 | 0.0690 | 0.0864 | 0.9540 | -0.7746 | -0.4228 | 0.3519 |
| $\alpha_2 = 1.2$ | 50 | 1.2006 | 0.1940 | 0.2465 | 0.9500 | 0.7086 | 1.6925 | 0.9839 |
| | 100 | 1.1982 | 0.1427 | 0.1802 | 0.9430 | 0.8561 | 1.5403 | 0.6842 |
| | 250 | 1.1943 | 0.0896 | 0.1115 | 0.9410 | 0.9815 | 1.4071 | 0.4256 |
| | 450 | 1.1994 | 0.0644 | 0.0801 | 0.9530 | 1.0418 | 1.3569 | 0.3151 |
| | 850 | 1.1988 | 0.0466 | 0.0584 | 0.9560 | 1.0844 | 1.3131 | 0.2287 |
| | 1000 | 1.2018 | 0.0432 | 0.0533 | 0.9600 | 1.0963 | 1.3072 | 0.2108 |
| $\beta = 0.1$ | 50 | 0.1078 | 0.0107 | 0.0146 | 0.9420 | 0.0791 | 0.1366 | 0.0574 |
| | 100 | 0.1098 | 0.0109 | 0.0134 | 0.9160 | 0.0893 | 0.1303 | 0.0411 |
| | 250 | 0.1032 | 0.0055 | 0.0073 | 0.9190 | 0.0911 | 0.1153 | 0.0242 |
| | 450 | 0.1021 | 0.0041 | 0.0051 | 0.9280 | 0.0932 | 0.1110 | 0.0178 |
| | 850 | 0.1008 | 0.0028 | 0.0034 | 0.9420 | 0.0944 | 0.1072 | 0.0128 |
| | 1000 | 0.1008 | 0.0026 | 0.0032 | 0.9420 | 0.0949 | 0.1067 | 0.0118 |

exogenous variables. The MKTL median regression is attained when $q = 0.50$. The logit link function is utilized in this study to relate the exogenous variables to the conditional quantiles. Thus,

$$\tau_i = \frac{\exp(x_i'\alpha)}{1 + \exp(x_i'\alpha)}, i = 1, 2, \ldots, n.$$

The maximum likelihood estimation procedure is adopted to estimate the parameters of the QRM and the log-likelihood function for a sample of size $n$ is given by

$$\ell = \sum_{i=1}^{n} f(y_i; \beta, \tau_i). \tag{19}$$

The estimates of the QRM parameters are attained by maximizing the log-likelihood function.

### 7.1 Residual analysis

Assessing the suitability of the QRM before using it for any inference is very vital. The model assessment (or diagnostics) can be done by examining its residuals. How well the residuals behave will determine the adequacy of the model for a given data. We employed the Cox-Snell residuals (CSR) (see [46]) to determine how adequate the QRM is. The CSR is given by

$$r_i = -\log(1 - F(y_i; \hat{\beta}, \hat{\alpha})), i = 1, 2, \ldots, n,$$

**Table 20. MKTL QRM simulation results for scenario II.**

| Parameter | $n$ | AE | AB | RMSE | CP | LCL | UCL | AWCI |
|---|---|---|---|---|---|---|---|---|
| $\alpha_0 = 0.1$ | 50 | 0.0930 | 0.2249 | 0.2835 | 0.9310 | -0.4190 | 0.6050 | 1.0240 |
| | 100 | 0.1133 | 0.1471 | 0.1851 | 0.9430 | -0.2419 | 0.4684 | 0.7104 |
| | 250 | 0.0952 | 0.0934 | 0.1172 | 0.9500 | -0.1292 | 0.3196 | 0.4487 |
| | 450 | 0.0967 | 0.0716 | 0.0890 | 0.9430 | -0.0701 | 0.2635 | 0.3335 |
| | 850 | 0.0998 | 0.0510 | 0.0644 | 0.9380 | -0.0211 | 0.2207 | 0.2418 |
| | 1000 | 0.1016 | 0.0466 | 0.0583 | 0.9430 | -0.0097 | 0.2129 | 0.2226 |
| $\alpha_1 = 0.7$ | 50 | 0.7153 | 0.3390 | 0.4221 | 0.9250 | -0.0505 | 1.4810 | 1.5314 |
| | 100 | 0.6926 | 0.2209 | 0.2796 | 0.9410 | 0.1666 | 1.2186 | 1.0521 |
| | 250 | 0.6973 | 0.1351 | 0.1706 | 0.9520 | 0.3676 | 1.0269 | 0.6593 |
| | 450 | 0.7041 | 0.1010 | 0.1270 | 0.9480 | 0.4598 | 0.9484 | 0.4886 |
| | 850 | 0.7013 | 0.0757 | 0.0953 | 0.9320 | 0.5240 | 0.8787 | 0.3547 |
| | 1000 | 0.6993 | 0.0681 | 0.0857 | 0.9400 | 0.5362 | 0.8623 | 0.3261 |
| $\alpha_2 = 0.4$ | 50 | 0.3968 | 0.1838 | 0.2310 | 0.9390 | -0.0378 | 0.8313 | 0.8690 |
| | 100 | 0.4041 | 0.1262 | 0.1593 | 0.9330 | 0.1038 | 0.7043 | 0.6005 |
| | 250 | 0.4018 | 0.0780 | 0.0967 | 0.9510 | 0.2126 | 0.5911 | 0.3784 |
| | 450 | 0.4010 | 0.0563 | 0.0705 | 0.9530 | 0.2603 | 0.5417 | 0.2814 |
| | 850 | 0.4003 | 0.0424 | 0.0526 | 0.9480 | 0.2982 | 0.5024 | 0.2042 |
| | 1000 | 0.3997 | 0.0378 | 0.0475 | 0.9510 | 0.3057 | 0.4938 | 0.1881 |
| $\beta = 0.5$ | 50 | 0.5229 | 0.0562 | 0.0718 | 0.9230 | 0.4076 | 0.6381 | 0.2305 |
| | 100 | 0.5198 | 0.0415 | 0.0476 | 0.9630 | 0.4400 | 0.5997 | 0.1596 |
| | 250 | 0.5053 | 0.0200 | 0.0248 | 0.9580 | 0.4563 | 0.5543 | 0.0980 |
| | 450 | 0.5020 | 0.0146 | 0.0181 | 0.9760 | 0.4657 | 0.5382 | 0.0725 |
| | 850 | 0.5018 | 0.0107 | 0.0134 | 0.9450 | 0.4754 | 0.5281 | 0.0527 |
| | 1000 | 0.5029 | 0.0112 | 0.0139 | 0.9020 | 0.4786 | 0.5273 | 0.0487 |

where $F(y_i; \hat{\beta}, \hat{\boldsymbol{\alpha}})$ is the re-parameterized cumulative distribution function of the MKTL distribution. If the QRM provides an adequate fit to the given data, the CSR are expected to follow the standard exponential distribution.

## 7.2 Simulation experiment for MKTL QRM

The performance of the maximum likelihood estimation procedure is appraised in this subsection to deduce how well it estimates the parameters of the QRM. Monte Carlo simulation experiments are implemented using 1000 replications with three different scenarios of parameter combinations with sample sizes $n$ = 50, 100, 250, 450, 850 and 1000. We scrutinize how well the estimates behave by computing metrics such as the average estimate (AE), absolute bias (AB), root mean square error (RMSE), 95% confidence interval (CI) coverage probability (CP) and the average width of the CI (AWCI). In addition, we estimated the lower confidence limit (LCL) and upper confidence limit (UCL) of the parameter estimates. The three different set of parameters utilized in the simulation are: I: $\alpha_0 = 0.4$, $\alpha_1 = -0.6$, $\alpha_2 = 1.2$, $\beta = 0.1$, II: $\alpha_0 = 0.1$, $\alpha_1 = 0.7$, $\alpha_2 = 0.4$, $\beta = 0.5$ and III: $\alpha_0 = -0.2$, $\alpha_1 = 1.8$, $\alpha_2 = 0.2$, $\beta = 4.5$. Two exogenous variables were utilized in the simulation, $x_{i1}$ is generated from a standard uniform distribution and $x_{i2}$ is binary variable generated from Bernoulli distribution with probability 0.5. The exogenous variables were held fixed during the simulations. The observations for the response variable $y_i$ are attained via the inversion method. Thus,

$$y_i = 1 - (1 - (1 + (\log(1/(1 - u_i)))^{-1/\beta})^{-1/r(q)})^{1/2},$$

**Table 21. MKTL QRM simulation results for scenario III.**

| Parameter | n | AE | AB | RMSE | CP | LCL | UCL | AWCI |
|---|---|---|---|---|---|---|---|---|
| $\alpha_0 = -0.2$ | 50 | -0.2006 | 0.0335 | 0.0416 | 0.9300 | -0.2786 | -0.1226 | 0.1561 |
| | 100 | -0.1990 | 0.0236 | 0.0293 | 0.9490 | -0.2557 | -0.1424 | 0.1133 |
| | 250 | -0.2009 | 0.0155 | 0.0193 | 0.9420 | -0.2368 | -0.1650 | 0.0719 |
| | 450 | -0.2002 | 0.0109 | 0.0136 | 0.9500 | -0.2270 | -0.1735 | 0.0535 |
| | 850 | -0.1998 | 0.0082 | 0.0101 | 0.9470 | -0.2193 | -0.1803 | 0.0390 |
| | 1000 | -0.1999 | 0.0074 | 0.0093 | 0.9470 | -0.2179 | -0.1820 | 0.0360 |
| $\alpha_1 = 1.8$ | 50 | 1.8004 | 0.0461 | 0.0581 | 0.9400 | 1.6911 | 1.9098 | 0.2187 |
| | 100 | 1.8007 | 0.0328 | 0.0411 | 0.9420 | 1.7220 | 1.8794 | 0.1574 |
| | 250 | 1.8005 | 0.0216 | 0.0269 | 0.9380 | 1.7507 | 1.8503 | 0.0997 |
| | 450 | 1.8001 | 0.0149 | 0.0186 | 0.9560 | 1.7630 | 1.8371 | 0.0741 |
| | 850 | 1.7999 | 0.0111 | 0.0138 | 0.9470 | 1.7729 | 1.8268 | 0.0539 |
| | 1000 | 1.8002 | 0.0103 | 0.0131 | 0.9420 | 1.7754 | 1.8251 | 0.0497 |
| $\alpha_2 = 0.2$ | 50 | 0.2002 | 0.0264 | 0.0329 | 0.9330 | 0.1396 | 0.2609 | 0.1213 |
| | 100 | 0.1999 | 0.0180 | 0.0227 | 0.9440 | 0.1560 | 0.2438 | 0.0877 |
| | 250 | 0.2003 | 0.0112 | 0.0141 | 0.9510 | 0.1724 | 0.2281 | 0.0557 |
| | 450 | 0.1994 | 0.0085 | 0.0107 | 0.9430 | 0.1787 | 0.2201 | 0.0414 |
| | 850 | 0.2003 | 0.0064 | 0.0080 | 0.9460 | 0.1852 | 0.2154 | 0.0302 |
| | 1000 | 0.2000 | 0.0056 | 0.0070 | 0.9450 | 0.1861 | 0.2139 | 0.0278 |
| $\beta = 4.5$ | 50 | 4.8168 | 0.3974 | 0.5257 | 0.9820 | 3.7776 | 5.8561 | 2.0785 |
| | 100 | 4.6165 | 0.2716 | 0.3666 | 0.9570 | 3.9224 | 5.3106 | 1.3882 |
| | 250 | 4.5430 | 0.1872 | 0.2413 | 0.9230 | 4.1119 | 4.9741 | 0.8622 |
| | 450 | 4.5373 | 0.1221 | 0.1510 | 0.9780 | 4.2166 | 4.8580 | 0.6415 |
| | 850 | 4.5209 | 0.0951 | 0.1200 | 0.9500 | 4.2884 | 4.7535 | 0.4651 |
| | 1000 | 4.5157 | 0.0823 | 0.1026 | 0.9710 | 4.3016 | 4.7298 | 0.4282 |

where $u_i$ are observations from the standard uniform distribution. We used $q = 0.5$ to perform the simulation. The make-up of the regression model used in the simulation experiments is

$$\tau_i = \frac{\exp(\alpha_0 + \alpha_1 x_{i1} + \alpha_2 x_{i2})}{1 + \exp(\alpha_0 + \alpha_1 x_{i1} + \alpha_2 x_{i2})}.$$

The results in Tables 19–21 shows that the AEs are quite close to the true values and gets more closer as $n \to \infty$, the ABs and RMSEs decreases as $n \to \infty$, the CPs are quite high and closer to the nominal value of 0.95, and the AWCI gets narrower as $n$ increases. The estimates of the CI (LCL and UCL) gets tighter as the sample size become large. This is an affirmation that the estimates of the parameters are well behaved and the estimation approach adopted is able to estimate the parameters well.

## 7.3 MKTL QRM application

The appositeness of the MKTL QRM is exemplified in this section by exploring the effect of labour market insecurity (LMI) and homicide rate (HR) on educational attainment value (EAV) in OECD countries. The detail description of the data can be found in [47]. Mazucheli et al. [47] fitted the unit generalized half normal (UGHN) QRM to the data and unveiled the 0.1 conditional quantile as the best with AIC = −62.8264 and BIC = −56.2761. Also, [48] studied the relationship between the EAV, LMI and HR utilizing the beta regression (AIC = −59.6000, BIC = −53.0.490) and log-weighted exponential mean regression (AIC = −65.2580,

**Table 22. Parameter estimates for various quantiles and information criteria.**

| q | | $\hat{\alpha}_0$ | $\hat{\alpha}_1$ | $\hat{\alpha}_2$ | $\hat{\beta}$ | AIC | BIC |
|---|---|---|---|---|---|---|---|
| 0.01 | Estimate | 1.2867 | -0.2283 | -0.0574 | 0.5933 | -67.5460 | -60.9957 |
| | Standard error | 0.2916 | 0.0681 | 0.0274 | 0.0760 | | |
| | p-value | <0.0001 | 0.0008 | 0.0364 | <0.0001 | | |
| 0.10 | Estimate | 1.6475 | -0.1873 | -0.0495 | 0.5946 | -67.5118 | -60.9614 |
| | Standard error | 0.2538 | 0.0551 | 0.0222 | 0.0756 | | |
| | p-value | <0.0001 | 0.0007 | 0.0258 | <0.0001 | | |
| 0.25 | Estimate | 1.9163 | -0.1649 | -0.0456 | 0.5942 | -67.3156 | -60.7652 |
| | Standard error | 0.2332 | 0.0481 | 0.0197 | 0.0754 | | |
| | p-value | <0.0001 | 0.0006 | 0.0208 | <0.0001 | | |
| 0.50 | Estimate | 2.2769 | -0.1416 | -0.0419 | 0.5934 | -66.8989 | -60.3486 |
| | Standard error | 0.2134 | 0.0405 | 0.0175 | 0.0753 | | |
| | p-value | <0.0001 | 0.0005 | 0.0165 | <0.0001 | | |
| 0.75 | Estimate | 2.6727 | -0.1225 | -0.0391 | 0.5930 | -66.3645 | -59.8142 |
| | Standard error | 0.2076 | 0.0337 | 0.0159 | 0.0751 | | |
| | p-value | <0.0001 | 0.0003 | 0.0140 | <0.0001 | | |
| 0.90 | Estimate | 3.0198 | -0.1104 | -0.0373 | 0.5925 | -65.9342 | -59.3839 |
| | Standard error | 0.2196 | 0.0292 | 0.0150 | 0.0750 | | |
| | p-value | <0.0001 | 0.0002 | 0.0129 | <0.0001 | | |
| 0.95 | Estimate | 3.2148 | -0.1053 | -0.0366 | 0.5920 | -65.7288 | -59.1785 |
| | Standard error | 0.2324 | 0.0272 | 0.0146 | 0.0749 | | |
| | p-value | <0.0001 | 0.0001 | 0.0125 | <0.0001 | | |
| 0.99 | Estimate | 3.5499 | -0.0987 | -0.0356 | 0.5910 | -65.4420 | -58.8917 |
| | Standard error | 0.2618 | 0.0247 | 0.0142 | 0.0749 | | |
| | p-value | <0.0001 | <0.0001 | 0.0122 | <0.0001 | | |

BIC = −58.7070) models. Here, the regression structure

$$\tau_i = \frac{\exp(\alpha_0 + \alpha_1 \mathrm{LMI}_i + \alpha_2 \mathrm{HR}_i)}{1 + \exp(\alpha_0 + \alpha_1 \mathrm{LMI}_i + \alpha_2 \mathrm{HR}_i)},$$

is arrogated to investigate the relationship. Table 22 convey the parameter estimates, standard errors, p−values and information criteria for the different conditional quantiles. The estimated parameters are all significant and the 0.01 conditional quantile appears the best from the reported information criteria. Hence, the LMI and HR have significant effect on the EAV. The fitted conditional quantiles in this study outperforms the models fitted in [47, 48].

The diagnostic checks of the model residuals for the various conditional quantiles is carried out using the CSR. The probability-probability (P-P) plots in Fig 35 affirm the adequacy of the model as the CSR clutch along the diagonals.

The rate of change of the estimated coefficients across the various quantiles are shown in Fig 36. The estimate of $\alpha_0$ increase as the quantile level increases whiles estimates of $\alpha_1$ and $\alpha_2$ approaches zero as the quantile level increases.

## 8 Concluding remarks

In this paper, we introduce another appendage of the Topp-Leone distribution hinged on the modified Kies family of distributions. Some statistical properties of the contemporary

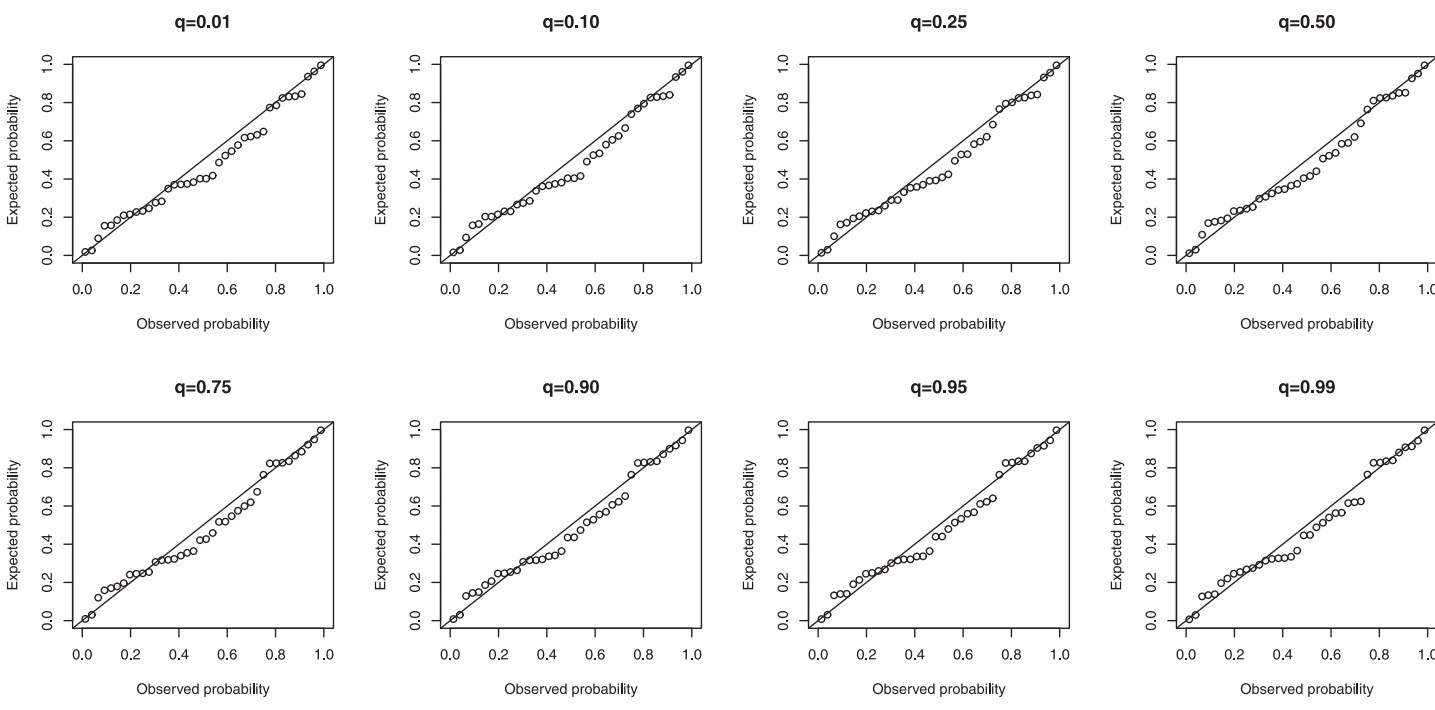

**Fig 35. P-P plots of CSR for the MKTL QRM.**

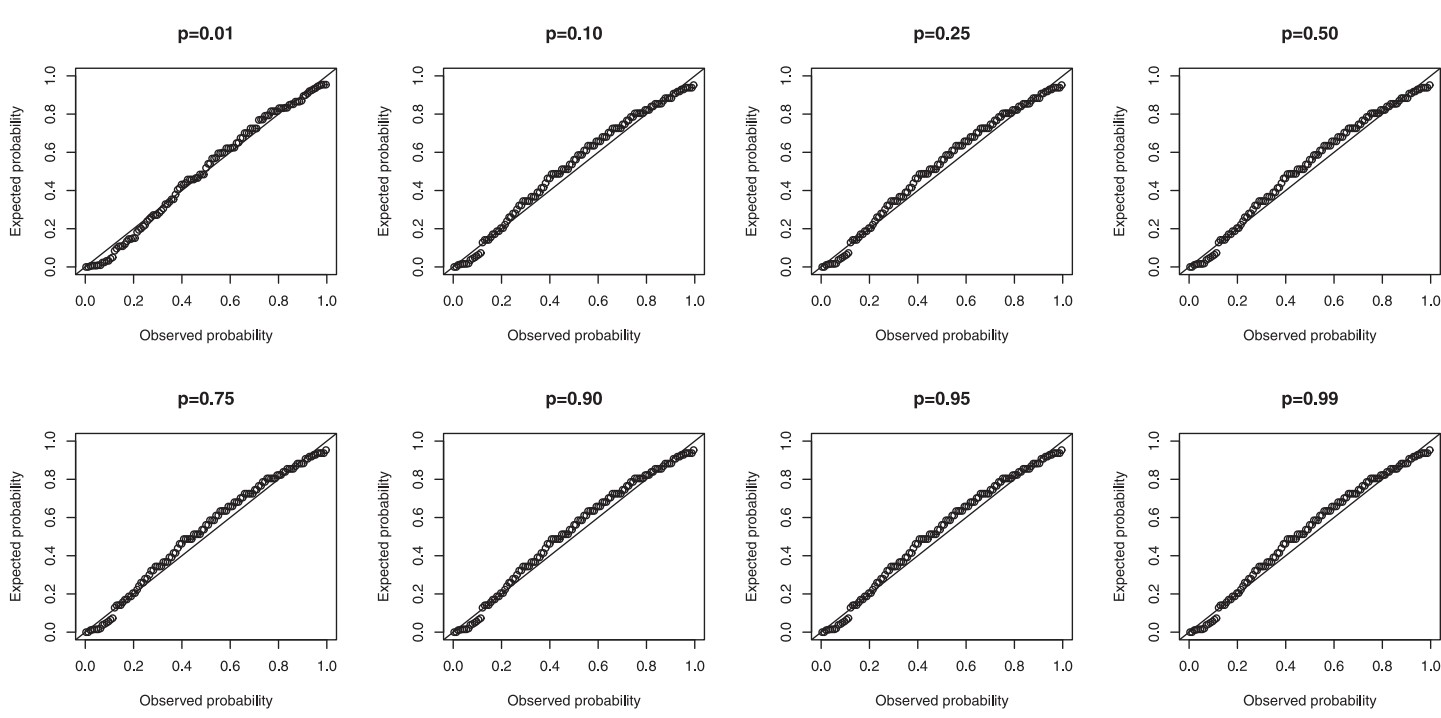

**Fig 36. Rate of change of parameter estimates for various quantiles.**

distribution are attained and twelve estimation methods utilized to estimate the parameters of the distribution. The findings of the simulation experiments affirm the maximum likelihood method as the superior for estimating the parameters of the distribution. The practicality of the new distribution is exemplified utilizing three data sets and the outcome suggest the MKTL distribution as the best when compared to other competitive distributions. The performance of the proposed quantile regression is assessed by exploring the effects of LMI and HR on EAV of OECD countries. The developed regression model provided a good fit to the given data and proved to be better than the UGHN QRM, beta and log-weighted exponential mean regression models.

## Supporting information

**S1 File.**
(ZIP)

**S2 File.**
(BST)

## Author Contributions

**Conceptualization:** Safar M. Alghamdi, Olayan Albalawi, Sanaa Mohammed Almarzouki, Vasili B. V. Nagarjuna, Suleman Nasiru, Mohammed Elgarhy.

**Formal analysis:** Safar M. Alghamdi, Olayan Albalawi, Sanaa Mohammed Almarzouki, Vasili B. V. Nagarjuna, Suleman Nasiru, Mohammed Elgarhy.

**Methodology:** Safar M. Alghamdi, Olayan Albalawi, Sanaa Mohammed Almarzouki, Vasili B. V. Nagarjuna, Suleman Nasiru, Mohammed Elgarhy.

**Software:** Safar M. Alghamdi, Olayan Albalawi, Sanaa Mohammed Almarzouki, Vasili B. V. Nagarjuna, Suleman Nasiru, Mohammed Elgarhy.

**Writing – original draft:** Safar M. Alghamdi, Olayan Albalawi, Sanaa Mohammed Almarzouki, Vasili B. V. Nagarjuna, Suleman Nasiru, Mohammed Elgarhy.

**Writing – review & editing:** Safar M. Alghamdi, Olayan Albalawi, Sanaa Mohammed Almarzouki, Vasili B. V. Nagarjuna, Suleman Nasiru, Mohammed Elgarhy.

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
