## [Decision Letter · Decision Letter 0]

29 May 2024

PONE-D-24-17754Different Estimation Methods of the Modified Kies Topp-Leone Model with Applications and Quantile RegressionPLOS ONE

Dear Dr. Nagarjuna,

Thank you for submitting your manuscript to PLOS ONE. After careful consideration, we feel that your manuscript will likely be suitable for publication if it is revised to address the points below.   Therefore, my decision is "Minor Revision".

We invite you to submit a revised version of the manuscript that addresses the points raised during the review process.

Please revise this paper.

We encourage you to submit your revision by Jul 13 2024 11:59PM. If you will need more time than this to complete your revisions, please reply to this message or contact the journal office at plosone@plos.org. Please include the following items when submitting your revised manuscript:A rebuttal letter that responds to each point raised by the academic editor and reviewer(s). You should upload this letter as a separate file labeled 'Response to Reviewers'.A marked-up copy of your manuscript that highlights changes made to the original version. You should upload this as a separate file labeled 'Revised Manuscript with Track Changes'.An unmarked version of your revised paper without tracked changes. You should upload this as a separate file labeled 'Manuscript'.If applicable, we recommend that you deposit your laboratory protocols in protocols.io to enhance the reproducibility of your results. Protocols.io assigns your protocol its own identifier (DOI) so that it can be cited independently in the future. For instructions see: https://journals.plos.org/plosone/s/submission-guidelines#loc-laboratory-protocols. Additionally, PLOS ONE offers an option for publishing peer-reviewed Lab Protocol articles, which describe protocols hosted on protocols.io. Read more information on sharing protocols at https://plos.org/protocols?utm_medium=editorial-email&utm_source=authorletters&utm_campaign=protocols.

We look forward to receiving your revised manuscript.

Kind regards,

Oluwafemi Samson Balogun, Ph.D.

Academic Editor

PLOS ONE

Journal Requirements:

"This research was funded by Taif University, Saudi Arabia, Project No. (TU-DSSP-2024-296)."

"The authors extended their appreciation to Taif University, Saudi Arabia, for supporting this work

through project number (TU-DSSP-2024-296)."

"This research was funded by Taif University, Saudi Arabia, Project No. (TU-DSSP-2024-296)."

6. We note that your Data Availability Statement is currently as follows: All relevant data are within the manuscript and its Supporting Information files.

Reviewers' comments:

Reviewer's Responses to Questions

**Comments to the Author**

1. Is the manuscript technically sound, and do the data support the conclusions?

Reviewer #1: Yes

Reviewer #2: Yes

2. Has the statistical analysis been performed appropriately and rigorously? 

Reviewer #1: Yes

Reviewer #2: Yes

3. Have the authors made all data underlying the findings in their manuscript fully available?

Reviewer #1: Yes

Reviewer #2: Yes

4. Is the manuscript presented in an intelligible fashion and written in standard English?

Reviewer #1: Yes

Reviewer #2: Yes

5. Review Comments to the Author

Reviewer #1: This paper introduces the modified Kies Topp-Leone (MKTL) distribution for modeling data on the (0, 1) or [0, 1] interval. The shapes of the density and hazard rate functions manifest desirable shapes, making the MKTL distribution suitable for modeling data with different characteristics at the unit interval. Twelve different estimation methods are utilized to estimate the distribution parameters, and Monte Carlo simulation experiments are executed to assess the performance of the methods. The simulation results suggest that the maximum likelihood method is the superior method. The usefulness of the new distribution is illustrated by utilizing three data sets. This work is quite good, scientifically sound and suitable for publication. Therefore it must be accepted for publication.

Reviewer #2: REVIEWER REPORT ON PONE-D-24-17754

TOPIC: DIFFERENT ESTIMATION METHODS OF THE MODIFIED KIES TOPPLEONE MODEL WITH APPLICATIONS AND QUANTILE REGRESSION

After carefully reviewing the manuscript of the authors, I can say they have done some good

work. However, I have the following comments which are MAJOR for the authors to address.

1. Authors did not define what Q1, Q2, Q3, BSK and MKUR measures represents in Table

1. Authors should also provide 3D plots for the BSK and MKUR measures.

2. In section 4.3, Cramer_von_Mises should be written as Cramér-von Mises.

3. The word burr in the second paragraph of Section 6 should be written as Burr.

4. The sum in the last column in Table 12 is not necessary. Authors should delete it.

5. In Table 17, what does NaN represents. Is it an indication that the optimization did not

converge for the MOKw distribution. Authors should crosscheck and address the issue.

6. Authors need to carefully proof read the manuscript and address all grammatical errors.

7- Add R codes with numerical value of lower and upper quintile in Apendex for figure 19: Rate of change of parameter estimates for various quantiles

6. PLOS authors have the option to publish the peer review history of their article (what does this mean?). If published, this will include your full peer review and any attached files.

Reviewer #1: **Yes: **Dr. Muhammad Suhail

Reviewer #2: No

---

## [Author Response · Author response to Decision Letter 0]

13 Jun 2024

Response to the reviewer’s comments 

PONE-D-24-17754

Title: Different Estimation Methods of the Modified Kies Topp-Leone Model with Applications and Quantile Regression

PLOS ONE

Dear Editors and Reviewers: 

 First, the authors would like to thank the Editor in Chief, Associate editor, Lead editor, Academic editor and Anonymous referees for spending their time on the manuscript carefully. The comments of the editors and reviewers are valuable. We have taken all the suggestions/comments positively and did our best to incorporate all these suggestions in the revised version. Our point wise responses to the reviewer’s comments/suggestions are given below. 

Response to Editor 

 Dear Sir/Mam, we appreciate your time in handling our paper and providing suggestions for improvement. We believe the quality of the revised version has considerably improved and hope that you find the revised manuscript satisfactory this time.

Reply to the Reviewer #1 

Reviewer #1: This paper introduces the modified Kies Topp-Leone (MKTL) distribution for modeling data on the (0, 1) or [0, 1] interval. The shapes of the density and hazard rate functions manifest desirable shapes, making the MKTL distribution suitable for modeling data with different characteristics at the unit interval. Twelve different estimation methods are utilized to estimate the distribution parameters, and Monte Carlo simulation experiments are executed to assess the performance of the methods. The simulation results suggest that the maximum likelihood method is the superior method. The usefulness of the new distribution is illustrated by utilizing three data sets. This work is quite good, scientifically sound and suitable for publication. Therefore it must be accepted for publication.

Answer: Many thanks for you positive report.

Reply to the Reviewer #2 

Reviewer #2: REVIEWER REPORT ON PONE-D-24-17754

TOPIC: DIFFERENT ESTIMATION METHODS OF THE MODIFIED KIES TOPPLEONE MODEL WITH APPLICATIONS AND QUANTILE REGRESSION

After carefully reviewing the manuscript of the authors, I can say they have done some good

work. However, I have the following comments which are MAJOR for the authors to address.

1. Authors did not define what Q1, Q2, Q3, BSK and MKUR measures represents in Table 1

 Answer: Many thanks for this nice comment. We defined it in the revised version.

2. Authors should also provide 3D plots for the BSK and MKUR measures.

 Answer: Many thanks for this nice comment. We added it in Figures 2 and 3.

3. In section 4.3, Cramer_von_Mises should be written as Cramér-von Mises.

Answer: Many thanks for this nice comment. We corrected it.

4. The word burr in the second paragraph of Section 6 should be written as Burr.

Answer: Many thanks for this nice comment. We corrected it.

5. The sum in the last column in Table 12 is not necessary. Authors should delete it.

 Answer: Many thanks for this nice comment. We deleted it.

6. In Table 17, what does NaN represents. Is it an indication that the optimization did not converge for the MOKw distribution. Authors should crosscheck and address the issue.

 Answer: Many thanks for this nice comment. We deleted it.

7. Authors need to carefully proof read the manuscript and address all grammatical errors.

Answer: Many thanks for this nice comment. We revised it.

8. Add R codes with numerical value of lower and upper quintile in Appendix for figure 19: Rate of change of parameter estimates for various quantiles

Answer: Many thanks for this nice comment. Please attach it

######################################### RATE OF CHANGE OF Parameter Plots#######################################

########### Plotting the rate of Change of Parameter Values Across various Quantiles #############

######## alpha0, alpha1, alpha2 and beta are the estimate of the parameters for various quantiles ###

##########lapha0, lapha1, lalpha2 and lbeta are the estimated lower confidence limit for the parameters across various quantiles##

#### uapha0, uapha1, ualpha2 and ubeta are the estimated upper confidence limit for the parameters across various quantiles##

q<-c(0.01,0.1,0.25,0.5,0.75,0.9,0.95,0.99) ######## quantiles 

alpha0<-c(1.28673937,1.64753913,1.91634265,2.27689722,2.67268642,3.0198273,3.21475942,3.54988938)##### estimates of alpha0 for respective quantiles 

lalpha0<-c(0.71526609,1.15010681,1.45934317,1.8586783,2.26585706,2.58945246,2.75918486,3.03678882)##### estimates of lalpha0 for respective quantiles 

ualpha0<-c(1.85821265,2.14497145,2.37334213,2.69511614,3.07951578,3.45020214,3.67033398,4.06298994)##### estimates of ualpha0 for respective quantiles 

alpha1<-c(-0.22829258,-0.1873418-0.164946,-0.14159633,-0.1224503,-0.11038132,-0.10525902,-0.09865827)##### estimates of alpha1 for respective quantiles 

lalpha1<-c(-0.36172742,-0.29541816,-0.25920828,-0.22090185,-0.1885807,-0.16765252,-0.15866706,-0.14704675)##### estimates of lalpha1 for respective quantiles 

ualpha1<-c(-0.09485774,-0.07926544,-0.07068372,-0.06229081,-0.0563199,-0.05311012,-0.05185098,-0.05026979)##### estimates of ualpha1 for respective quantiles 

alpha2<-c(-0.05742771,-0.04947872,-0.04561576,-0.04194493,-0.03910326,-0.03732533,-0.03655452,-0.03555266)##### estimates of alpha2 for respective quantiles 

lalpha2<-c(-0.11122383,-0.092977,-0.08429636,-0.07624885,-0.07029078,-0.06673905,-0.06525088,-0.06335722)##### estimates of lalpha2 for respective quantiles 

ualpha2<-c(-0.00363159,-0.00598044-0.00693516,-0.00764101,-0.00791574,-0.00791161,-0.00785816,-0.0077481)##### estimates of ualpha2 for respective quantiles 

beta<-c(0.59333507,0.59459697,0.5942175,0.59339415,0.59295771,0.59248447,0.59204514,0.59096904)##### estimates of beta for respective quantiles 

lbeta<-c(0.44437311,0.44632689,0.44637274,0.44588847,0.44579111,0.44553543,0.44515686,0.44409252)##### estimates of lbeta for respective quantiles 

ubeta<-c(0.74229703,0.74286705,0.74206226,0.74089983,0.74012431,0.73943351,0.73893342,0.73784556)##### estimates of ubeta for respective quantiles 

windows(height=10,width=20)

par(mfrow=c(2,2))

plot(q,alpha0,type="l",xlab="Quantile Level",ylab=expression(paste(hat(alpha[0]))),ylim=c(0,5))

lines(q,lalpha0,type="l",col="red")

lines(q,ualpha0,type="l",col="red")

legend("bottomright",legend=c("Point Estimate","95% CI."),col=c("black","darkgreen"),lty=1,text.font=1,cex=0.6,inset=c(0.1,0.1))

# Fill area between lines 

polygon(c(q, rev(q)), c(alpha0, rev(lalpha0)), col = "darkgreen")

polygon(c(q, rev(q)), c(alpha0, rev(ualpha0)), col = "darkgreen")

#################

plot(q,alpha1,type="l",xlab="Quantile Level",ylab=expression(paste(hat(alpha[1]))),ylim=c(-0.5,0))

lines(q,lalpha1,type="l",col="red")

lines(q,ualpha1,type="l",col="red")

legend("bottomleft",legend=c("Point Estimate","95% CI"),col=c("black","darkgreen"),lty=1,text.font=1,cex=0.6,inset=c(0.1,0.1))

polygon(c(q, rev(q)), c(alpha1, rev(lalpha1)), col = "darkgreen")

polygon(c(q, rev(q)), c(alpha1, rev(ualpha1)), col = "darkgreen")

##############################

plot(q,alpha2,type="l",xlab="Quantile Level",ylab=expression(paste(hat(alpha[2]))),ylim=c(-0.6,0.1))

lines(q,lalpha1,type="l",col="red")

lines(q,ualpha2,type="l",col="red")

legend("bottomright",legend=c("Point Estimate","95% CI"),col=c("black","darkgreen"),lty=1,text.font=1,cex=0.6,inset=c(0.1,0.1))

polygon(c(q, rev(q)), c(alpha2, rev(lalpha1)), col = "darkgreen")

polygon(c(q, rev(q)), c(alpha2, rev(ualpha2)), col = "darkgreen")

##################

plot(q,beta,type="l",xlab="Quantile Level",ylab=expression(paste(hat(beta))),ylim=c(0,0.9))

lines(q,lbeta,type="l",col="red")

lines(q,ubeta,type="l",col="red")

legend("bottomright",legend=c("Point Estimate","95% CI"),col=c("black","darkgreen"),lty=1,text.font=1,cex=0.6,inset=c(0.1,0.1))

polygon(c(q, rev(q)), c(beta, rev(lbeta)), col = "darkgreen")

polygon(c(q, rev(q)), c(beta, rev(ubeta)), col = "darkgreen")

---

## [Decision Letter · Decision Letter 1]

4 Jul 2024

Different Estimation Methods of the Modified Kies Topp-Leone Model with Applications and Quantile Regression

PONE-D-24-17754R1

Dear Dr. Mohammed Elgarhy,

We’re pleased to inform you that your manuscript has been judged scientifically suitable for publication and will be formally accepted for publication once it meets all outstanding technical requirements.

Kind regards,

Oluwafemi Samson Balogun, Ph.D.

Academic Editor

PLOS ONE

Additional Editor Comments (optional):

Reviewers' comments:

Reviewer's Responses to Questions

**Comments to the Author**

1. If the authors have adequately addressed your comments raised in a previous round of review and you feel that this manuscript is now acceptable for publication, you may indicate that here to bypass the “Comments to the Author” section, enter your conflict of interest statement in the “Confidential to Editor” section, and submit your "Accept" recommendation.

Reviewer #2: All comments have been addressed

2. Is the manuscript technically sound, and do the data support the conclusions?

Reviewer #2: Yes

3. Has the statistical analysis been performed appropriately and rigorously? 

Reviewer #2: Yes

4. Have the authors made all data underlying the findings in their manuscript fully available?

Reviewer #2: Yes

5. Is the manuscript presented in an intelligible fashion and written in standard English?

Reviewer #2: No

6. Review Comments to the Author

Reviewer #2: After careful reading, now the paper is in very good shape.

The revised version is deserved for possible publication.

7. PLOS authors have the option to publish the peer review history of their article (what does this mean?). If published, this will include your full peer review and any attached files.

Reviewer #2: No

---

## [Editor Report · Acceptance letter]

15 Jul 2024

PONE-D-24-17754R1 

PLOS ONE

Dear Dr. Elgarhy, 

I'm pleased to inform you that your manuscript has been deemed suitable for publication in PLOS ONE. Congratulations! Your manuscript is now being handed over to our production team.

Kind regards, 

on behalf of

Dr. Oluwafemi Samson Balogun 

Academic Editor

PLOS ONE